# Layer Collapse Can be Induced by Unstructured Pruning

**Zhu Liao**                                                                    *zhu.liao@telecom-paris.fr*
*LTCI, Télécom Paris, Institut Polytechnique de Paris*

**Victor Quétu**                                                          *victor.quetu@telecom-paris.fr*
*LTCI, Télécom Paris, Institut Polytechnique de Paris*

**Van-Tam Nguyen**                                                *van-tam.nguyen@telecom-paris.fr*
*LTCI, Télécom Paris, Institut Polytechnique de Paris*

**Enzo Tartaglione**                                          *enzo.tartaglione@telecom-paris.fr*
*LTCI, Télécom Paris, Institut Polytechnique de Paris*

**Reviewed on OpenReview:** *https://openreview.net/forum?id=rfDYZNZIZT*

## Abstract

Unstructured pruning is a popular compression method for efficiently reducing model parameters. However, while it effectively decreases the number of parameters, it is commonly believed that unstructured pruning cannot shorten the computational critical path, i.e., the maximum number of layers traversed during forward propagation.

In this paper, we study when and how unstructured pruning can yield structural effects. For rectifier-activated networks, we introduce the notion of neuron entropy, which quantifies the degree of nonlinearity utilization. We show that magnitude-based pruning naturally lowers this entropy, sometimes down to zero-entropy layers that become linearizable and can thus be removed. Building on this insight, we propose a method that leverages "unstructured" pruning to favor sparsity in low-entropy layers, enabling their complete removal. We validate the phenomenon across CNNs, Vision Transformers, and NLP models: unstructured pruning can induce effective layer removal with little or no performance degradation in over-parameterized networks. Our code is available at https://github.com/ZhuLIAO001/NEPENTHE.git.

## 1 Introduction

Artificial Intelligence has undergone a transformative evolution propelled by the advent of Deep Neural Networks (DNNs), which have emerged as instrumental in achieving state-of-the-art outcomes across pivotal computer vision domains, including semantic segmentation (Lu et al., 2022) and classification (Zhang et al., 2023; Arslan et al., 2022). Notably, the pervasive impact of DNNs extends beyond conventional computer vision tasks, showcasing absolute potential in realms such as natural language processing (Touvron et al., 2023), and multi-modal tasks (Sun et al., 2018).

While DNNs' performance has exhibited scalability concerning model and dataset size (Hestness et al., 2017), the inherent computational burden is one major downside. Notably, contemporary state-of-the-art models are characterized by millions (or even billions) of parameters, demanding billions (or trillions) of floating-point operations (FLOPs) for a single input prediction (Guo et al., 2022). The heavy hardware and energy demands of large networks hinder real-time and edge applications.

Over the past decade, the research landscape has witnessed the emergence of compression techniques as a crucial avenue to address the resource-intensive nature of DNNs. Intrinsically, there exists a link between the generalization capability of DNNs and the model's complexity: off-the-shelf architectures employed in

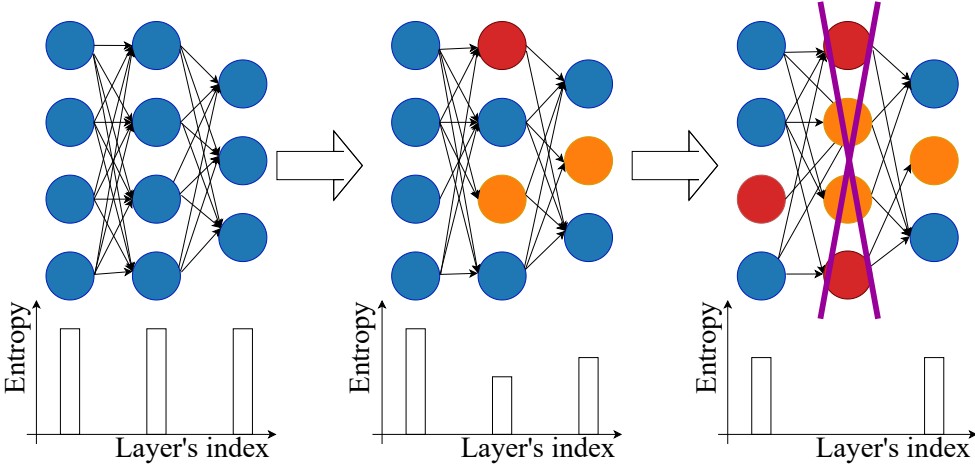

Figure 1: In this work, we show that the average neuron's entropy calculated at the layer scale reduces as we induce some sparsity in the model, and once it reaches zero, the layer becomes linearizable and thus can be removed.

downstream tasks are, in many cases, over-parametrized, representing a threat for generalization (Hestness et al., 2017). Removing redundant parameters improves both computation and generalization (Tartaglione et al., 2021; 2022). Recently, the most widely used compression technique that removes the greatest number of parameters is unstructured pruning (Han et al., 2015), which eliminates individual weights based on their magnitude without considering the network's structure.

Nevertheless, the impact of removing individual parameters or whole filters on recent computing resources, such as GPUs, is relatively marginal. Due to the parallelization of computations, the size of layers, whether larger or smaller, is primarily constrained by memory caching and core availability. The bottleneck in computation lies in the *critical path* that forward-propagation must traverse (Ali Mehmeti-Göpel & Disselhoff, 2023), a challenge that can be addressed by strategically removing layers.

It is commonly believed that unstructured pruning can only remove parameters but not shorten network depth. However, our theoretical and experimental evidence demonstrates that this perception is incomplete: unstructured pruning can induce layer collapse. In rectifier-activated networks, unstructured pruning reduces the neuron entropy and when a layer's average entropy drops to zero, the layer becomes linearizable and can be removed. Therefore, we design an unstructured entropy-weighted allocation pruning scheme aimed at driving the entropy value of the low-entropy layer to zero with minimal performance loss. We summarize, here below, our key messages and contributions.

- We propose an entropy measure at the single-neuron level, indicating how much a neuron relies on its linear component. By minimizing this entropy, the neuron can be effectively linearized, and when the average entropy across neurons approaches zero, the layer itself becomes entirely linear (Sec. 3.1).
- We theoretically show that "unstructured" pruning, in rectifier-activated layers, naturally reduces the layer's entropy (Sec. 3.2), and further demonstrate it empirically (Sec. 4.2).
- We propose NEPENTHE, a novel method that reduces a neural network's effective depth (Sec. 3.3) by performing an entropy-guided reallocation of the unstructured pruning budget across layers (Sec. 3.3), and validate its effectiveness across a variety of setups and popular architectures (Sec. 4.3).

## 2 Related Works

**Neural Network Pruning.** Neural network pruning attracts attention for improving efficiency and reducing overfitting. Its goal is to reduce a cumbersome network to a smaller one while maintaining accuracy

by removing irrelevant weights, filters, or other structures from neural networks. While *structured* pruning removes entire neurons, filters, or channels (Tartaglione et al., 2021; He & Xiao, 2023; Lin et al., 2020), *unstructured* pruning algorithms remove weights without explicitly considering the neural network's structure (Han et al., 2015). Magnitude-based pruning, where the importance score to prune parameters is based on their magnitude (Han et al., 2015; Louizos et al., 2018; Zhu & Gupta, 2017), and gradient-based pruning, where the ranking or the penalty term is a function of the gradient magnitude (or to higher order derivatives) (Lee et al., 2019; Tartaglione et al., 2022), are the main types of unstructured pruning approaches. Blalock et al. (2020) compared the effectiveness of these approaches and concluded that, in general, gradient-based methods are less accurate than magnitude-based methods. Moreover, Gale et al. (2019) showed that simple magnitude pruning approaches achieve comparable or better results than complex methods, making them a good trade-off between complexity and competitiveness. In addition to magnitude-based pruning, a recent method called Wanda (Sun et al., 2024) ranks each weight by the absolute product of its magnitude and its input feature, pruning those with the lowest scores. We verify that the phenomenon discussed in this paper also holds for Wanda, showing its generality across different pruning criteria. From a computational perspective, it is widely recognized that structured pruning offers greater advantages over unstructured methods in general-purpose hardware environments, both in terms of memory and computation, despite achieving significantly lower sparsity levels (Bragagnolo et al., 2021). In this work, we focus on bridging this gap: investigating when and how unstructured pruning can induce structural effects.

**Entropy-Guided Pruning.** Some works have already tried to propose entropy-based approaches to guide pruning. For convolutional neural networks, Luo & Wu (2017) put forward an iterative filter pruning strategy in which the importance of each filter is calculated by its entropy-based channel selection metric. To recover performance, the pruned model is then fine-tuned. Also for CNNs, Hur & Kang (2019) suggested an entropy-based method that determines dynamically during training the threshold by considering the average amount of information from the weights to the output. Moreover, Min et al. (2018) proposed a two-stage filter pruning framework, first intra-layer and then extra-layer. Given that the entropy is a measure of disorder, evidently, it identifies filters that mutually have low entropy: these can be considered redundant and can be removed from the model. These methods reduce width, not depth. EASIER (Quétu et al., 2024) proposed an entropy-based importance metric to collapse low-information layers and reduce network depth. By measuring per-layer neuron entropy, they identify layers whose activations lie almost entirely in a linear or inactive regime and remove them wholesale. Although this approach relies on an entropy-based metric, it performs structured layer removal rather than applying unstructured weight pruning to directly lower entropy values. Similarly, recent work (Lin et al., 2024) shows that certain self-attention blocks in Vision Transformers exhibit low feature entropy and can be merged into their subsequent MLPs without harming accuracy. In their approach, entropy is used as a post-hoc scoring metric: layers with lower entropy are identified as redundant and directly removed, but the method itself does not attempt to reduce entropy during training. By contrast, in our framework, unstructured pruning actively drives the entropy of neurons and layers downward, allowing them to naturally reach a linearizable state in which they can be safely removed. By prioritizing pruning connections in low-entropy layers, EGP (Liao et al., 2023) is also an unstructured entropy-guided pruning method that reduces DNN depth. Nevertheless, when EGP performs unstructured pruning, it classifies rectifier states into fewer categories than ours and does not account for the entropy of individual neurons.

**Neural Network Depth Reduction.** Towards neural network depth reduction, Structured Sparsity Learning (SSL) (Wen et al., 2016) uses group lasso regularization on the weights of each layer to determine which layers are less critical. This approach aims to reduce redundant layers with minor performance loss. However, SSL does not guarantee entire layer removal while maintaining model performance, as depth reduction is often achieved at the cost of accuracy degradation. Then, Chen & Zhao (2019) inspects the possibility of having a layer-wise pruning method based on feature representation, a-posteriori employing a retraining strategy that utilizes knowledge distillation. However, its effectiveness strongly depends on the retraining stage, and the additional knowledge distillation step increases training cost and limits scalability to larger models. Endorsing this, Dror et al. (2022) proposed a method that learns whether non-linear activations can be removed, allowing the folding of consecutive linear layers into one. More specifically, ReLU-activated layers are replaced with PReLU activations, showcasing a regularized slope. During post-training, the PReLUs almost linear are removed, and the layer can be folded with its subsequent one. Ali Mehmeti-

Göpel & Disselhoff (2023) proposes a similar channel-wise approach that enables reducing more non-linear units in the network while maintaining similar performance. While previous methods attempted to reduce network depth by constraining activations to remain either linear or non-linear, our approach reveals that unstructured pruning naturally leads the model to achieve this behavior without explicit enforcement. We demonstrate, both theoretically and empirically, that there exists the possibility of network layers "collapsing on their own" to reduce their depth, even with the classical unstructured pruning strategy. This finding provides a new perspective on network depth reduction.

## 3 Unstructured Pruning Induces Layer Collapse

In this section, we first introduce the notion of neuron entropy, which quantifies the degree of nonlinearity utilization (Sec. 3.1). Then, we show that unstructured pruning naturally minimizes the neuron's entropy (in rectifier-activated layers) (Sec. 3.2). This motivates our entropy-guided pruning approach, which allows a gradual layer removal. Finally, we propose our method NEPENTHE, which focuses on pruning connections in layers with low entropy to remove them entirely (Sec. 3.3).

### 3.1 Entropy for Rectifier Activations

Let us assume $\psi$ is the rectifier of the $l$-th layer, populated by $N_l$ neurons. We can monitor the output $y_{l,i}^{\boldsymbol{x}}$ of the $i$-th neuron from a given input $\boldsymbol{x}$ of the dataset $\mathcal{D}$ and write it as:

$$y_{l,i}^{\boldsymbol{x}} = \psi(z_{l,i}^{\boldsymbol{x}}), \tag{1}$$

where $z_{l,i}^{\boldsymbol{x}}$ is the output of the $i$-th neuron inside the $l$-th layer. From equation 1, we can define three possible "states" for the neuron:

$$s_{l,i}^{\boldsymbol{x}} = \begin{cases} +1 & \text{if } z_{l,i}^{\boldsymbol{x}} > 0 \\ -1 & \text{if } z_{l,i}^{\boldsymbol{x}} < 0 \\ 0 & \text{if } z_{l,i}^{\boldsymbol{x}} = 0. \end{cases} \tag{2}$$

More synthetically, for the output of the $i$-th neuron, we can easily identify in which of these states we are by simply applying the sign function to $z_{l,i}^{\boldsymbol{x}}$, obtaining $s_{l,i}^{\boldsymbol{x}} = \text{sign}(z_{l,i}^{\boldsymbol{x}})$. Informally, we can say that the neuron is in the *ON State* when $s_{l,i}^{\boldsymbol{x}} = +1$ (as it is typically the linear region) while it is in the *OFF State* when $s_{l,i}^{\boldsymbol{x}} = -1$ (given that $\lim_{x \to -\infty} \psi(x) = 0$).[1] The third State $s_{l,i}^{\boldsymbol{x}} = 0$ is a special case, as it can be either mapped as an ON or OFF State. From the average over a batch of outputs for the neuron, we can obtain the probability (in the frequentist sense) of the i-th neuron of being in either the ON or the OFF States. For instance, we can obtain the probability of the ON State as:

$$p(s_{l,i} = +1) = \begin{cases} \frac{1}{S_{l,i}} \sum_{j=1}^{\|\mathcal{D}\|_0} s_{l,i}^{\boldsymbol{x}_j} \Theta(s_{l,i}^{\boldsymbol{x}_j}) & \text{if } S_{l,i} \neq 0 \\ 0 & \text{otherwise}, \end{cases} \tag{3}$$

where

$$S_{l,i} = \sum_{j=1}^{\|\mathcal{D}\|_0} \left| s_{l,i}^{\boldsymbol{x}_j} \right| \tag{4}$$

counts how many times the ON and the OFF states are encountered, $\|\mathcal{D}\|_0$ is the number of the input samples, and $\Theta$ is the Heaviside function.[2] Evidently, we exclude the third state from this count as it can be associated with being either within ON or OFF. Given that we are either interested in the ON or the OFF States, we can then deduce that, when $S_{l,i} \neq 0$, $p(s_{l,i} = -1) = 1 - p(s_{l,i} = +1)$. Given this, we can calculate the entropy of the $i$-th neuron in the $l$-th layer as follows:

$$\mathcal{H}_{l,i} = - \sum_{s_{l,i} = \pm 1} p(s_{l,i}) \log_2 \left[ p(s_{l,i}) \right]. \tag{5}$$

---

[1] There are few exceptions, such as LeakyReLU. In these cases, although the activation doesn't converge to zero, we still call it the OFF state since the output's magnitude is lower for the same input magnitude.

[2] For convolutional layers, it is necessary to sum and average over the entire feature map generated per input.

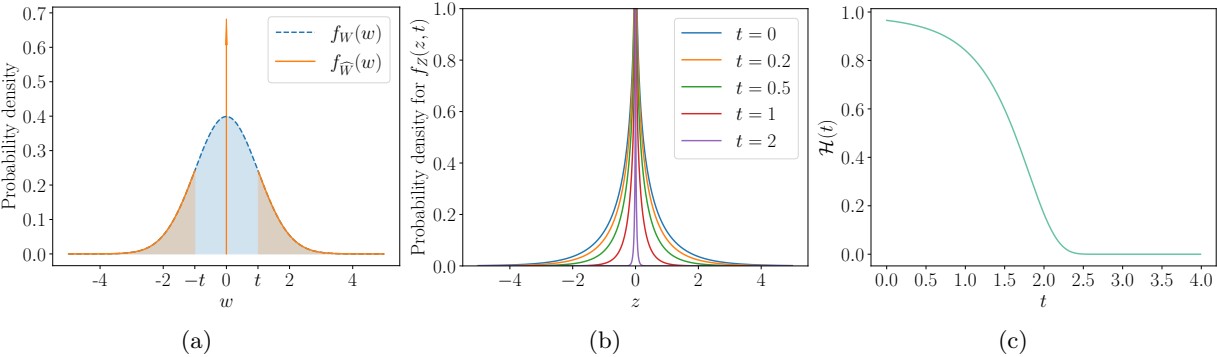

Figure 2: (a) Distribution of a layer's parameters with magnitude pruning at threshold $t$; (b) pre-activation distribution at varying $t$ under the assumption of independence and centering of the Gaussian distributed input and layer's parameters; (c) entropy of the rectifier-activated neuron's output as a function of $t$ , all in the large $N$ limit.

With the definition in equation 5, $\mathcal{H}_{l,i}$ can be zero in two possible cases:

- $s_{l,i} = -1 \ \forall j$. In this case, $z_{l,i} \leq 0 \ \forall j$. When employing a ReLU, the output of the $i$-th neuron is always 0, and in this specific case, the neuron can be simply pruned.

- $s_{l,i} = +1 \ \forall j$. In this case, $z_{l,i} \geq 0 \ \forall j$. When employing a ReLU, the output of the $i$-th neuron is always the same as its input, and the neuron can be absorbed by the following layer as there is no non-linearity between them anymore. For smooth rectifier variants such as GELU/SiLU, this process is an approximation where the error can be negligible. We formalize and bound this approximation error in Appendix A.

By averaging the entropy values for the total number of neurons $N_l$ inside the $l$-th layer, we can define the average layer entropy of the $l$-th layer as:

$$\widehat{\mathcal{H}}_l = \frac{1}{N_l} \sum_i \mathcal{H}_{l,i}. \tag{6}$$

We refer to $\widehat{\mathcal{H}}_l$ as the average first-order layer entropy of layer $l$, i.e., the mean neural entropy across its neurons. This quantity is a scalable proxy for layer linearizability and is not intended to estimate the joint entropy of the layer's activation-state vector.

Since we aim to minimize the depth of deep neural networks by eliminating zero-entropy layers, we would like to have $\widehat{\mathcal{H}}_l = 0$.

## 3.2 Unstructured Pruning Naturally Reduces the Entropy

Let us assume the input $\boldsymbol{x}$ for a given neuron is a sequence of random variables $X \sim \mathcal{N}(\mu_X, \sigma_X^2)$. Similarly, we can assume the $N$ parameters populating such neuron, for a large $N$ limit, follow as well a Gaussian distribution, and we model it as $W \sim \mathcal{N}(\mu_W, \sigma_W^2)$. These assumptions are empirically validated in Appendix D.5. Let us assume we apply a magnitude-based pruning mask to the neuron's parameters, where we apply some threshold $t$. As such, we obtain a modified distribution for the layer's parameters:

$$f_{\widehat{W}}(w,t) = \begin{cases} \dfrac{1}{\sigma_W \sqrt{2\pi}} \exp\left[ -\dfrac{1}{2}\left( \dfrac{w - \mu_W}{\sigma_W} \right)^2 \right] & |w| > t \\[4mm] \zeta(t)\delta(w) & |w| \leq t, \end{cases} \tag{7}$$

---

**Algorithm 1** Our proposed method NEPENTHE.

---

1: **function** NEPENTHE($\boldsymbol{w}^{\text{INIT}}$, $L$, $\mathcal{D}$, $\zeta$, $\theta$)
2:     $\boldsymbol{w} \leftarrow \text{Train}(\boldsymbol{w}^{\text{init}}, \mathcal{D}_{\text{train}})$
3:     dense_acc $\leftarrow$ Evaluate($\boldsymbol{w}$, $\mathcal{D}_{\text{val}}$)
4:     current_acc $\leftarrow$ dense_acc
5:     **while** current_acc $> \theta \cdot$ dense_acc **do**
6:         $\widehat{\mathcal{H}} \leftarrow \text{Entropy}(\boldsymbol{w}, L, \mathcal{D}_{\text{train}})$
7:         $\|\boldsymbol{w}\|_0^{\text{pruned}} \leftarrow \zeta \cdot \|\boldsymbol{w}\|_0$
8:         $\|\boldsymbol{w}_L\|_0^{\text{pruned}} \leftarrow \text{Weights\_to\_prune}(L, \widehat{\mathcal{H}}, \|\boldsymbol{w}\|_0^{\text{pruned}}, \mathcal{D}_{\text{train}})$
9:         $\boldsymbol{w} \leftarrow \text{Prune}(\|\boldsymbol{w}_L\|_0^{\text{pruned}})$
10:         $\boldsymbol{w} \leftarrow \text{Train}(\boldsymbol{w}, \mathcal{D}_{\text{train}})$
11:         current_acc $\leftarrow$ Evaluate($\boldsymbol{w}$, $\mathcal{D}_{\text{val}}$)
12:     **end while**
13:     **return** $\boldsymbol{w}$
14: **end function**

---

where

$$\zeta(t) = \frac{1}{2}\left[\text{erf}\left(\frac{t - \mu_W}{\sigma_W\sqrt{2}}\right) - \text{erf}\left(\frac{-t - \mu_W}{\sigma_W\sqrt{2}}\right)\right] \tag{8}$$

is the fraction of parameters pruned, or *pruning rate*, $\delta$ is the Dirac delta and erf is the error function. Fig. 2a displays an example of distribution when applying magnitude pruning having threshold $t$ against the original distribution. Following Craig (1936), we work with the standardised variables

$$\tilde{X} = \frac{X - \mu_X}{\sigma_X}, \qquad \tilde{W} = \frac{W - \mu_W}{\sigma_W},$$

so that $\tilde{X}, \tilde{W} \sim \mathcal{N}(0, 1)$, and with the normalised pre-activation

$$\tilde{Z} = \tilde{X}\tilde{W} = \frac{Z}{\sigma_X \sigma_W}.$$

For notational simplicity, we denote $\tilde{Z}$ again by $Z$ in what follows. Under the assumption of independent-centered distributions having a unitary variance, we can obtain the distribution for the pre-activation $z$ (resulting from the product of the weights and the input, modeled through the random variable $Z$), according to the result obtained by Craig (1936); Seijas-Macías & Oliveira (2012), follows

$$f_Z(z, t) = \frac{1}{\pi}K_0\left(\left|\frac{1}{q(t)} \cdot z\right|\right), \tag{9}$$

where

$$q(t) = 1 - \text{erf}\left(\frac{t}{\sqrt{2}}\right) \tag{10}$$

and $K_n$ is the n-th order modified Bessel function of the second kind. We can observe, from Fig. 2b, how $f_Z$ is affected by increasing the thresholding $t$. Now, let us assume the activation function of such a neuron is a rectifier function, and we are interested in observing what is the probability of the post-activation output being in the linear region: we are interested in measuring

$$p[Z > \varepsilon] = \frac{1}{\pi}\int_\epsilon^{+\infty} K_0\left(\left|\frac{1}{q(t)} \cdot z\right|\right) dz = \frac{1}{2}\left[1 - I\left(\frac{\epsilon}{q(t)}\right)\right], \tag{11}$$

where

$$I(x) = x[L_{-1}(x)K_0(x) + L_0(x)K_1(x)], \tag{12}$$

$L_n$ is the n-th order modified Struve function, and $\epsilon$ is a positive small value. From this, we can easily obtain the complementary probability $p[Z \leq \varepsilon]$ and for instance calculate the entropy between the two States.

---

**Algorithm 2** Function `Weights_to_prune`.

---

1: **function** WEIGHTS_TO_PRUNE($L$, $\widehat{\mathcal{H}}$, $\|\boldsymbol{w}\|_0^{\text{PRUNED}}$, $\mathcal{D}$)
2:     **for** $l \in L$ **do**
3:        $\mathcal{I}_l \leftarrow \frac{\|\boldsymbol{w}_l\|_1}{\|\boldsymbol{w}_l\|_0} \cdot \widehat{\mathcal{H}}_l$
4:        $\mathcal{R}_l \leftarrow \begin{cases} \frac{\sum_{j \in L} \mathcal{I}_{|}}{\mathcal{I}_l} & \text{if } \mathcal{I}_l \neq 0 \\ 0 & \text{otherwise.} \end{cases}$
5:        $\|\boldsymbol{w}_l\|_0^{\text{pruned}} \leftarrow \|\boldsymbol{w}\|_0^{\text{pruned}} \cdot \frac{\exp[\mathcal{R}_l]}{\sum_j \exp[\mathcal{R}(j)]}$
6:     **end for**
7:     **return** $\|\boldsymbol{w}_L\|_0^{\text{pruned}}$
8: **end function**

---

Fig. 2c displays the entropy as a function of the thresholding parameter $t$: as we observe, the entropy decreases given that the threshold increases: through unstructured pruning, the neuron's output entropy is naturally minimized when employing rectified activations, even in the oversimplified case here treated.

In the following, we will present how we are exploiting such a property of unstructured pruning towards layer entropy minimization.

### 3.3 NEPENTHE

Driven by the promising theoretical results presented in Sec. 3.1 and Sec. 3.2, we design here NEPENTHE (e**N**tropy-bas**E**d **P**runing as a n**E**ural **N**etwork dep**TH**'s r**E**ducer) that guides the unstructured pruning to lower the whole layer's entropy $\widehat{\mathcal{H}}_l$. Since we aim to increase the number of zero-entropy layers, intuitively more pruning should be applied to layers with lower entropy, as they are the best candidates to be removed. Concurrently, to minimize the impact on performance, only low-magnitude weights should be removed, as they are typically those providing the lowest contribution to the neural network's output (Han et al., 2015; Tartaglione et al., 2021). To reach these two objectives, we first define an intra-layer's pruning irrelevance score

$$\mathcal{I}_l = \frac{\|\boldsymbol{w}_l\|_1}{\|\boldsymbol{w}_l\|_0} \cdot \widehat{\mathcal{H}}_l, \tag{13}$$

where $\|\boldsymbol{w}_l\|_0$ is the current layer's parameters cardinality (hence, not accounting for the already pruned weights, if any) and $\|\boldsymbol{w}_l\|_1$ is the $\ell_1$ norm (sum of absolute values). This metric accounts for the average parameter's magnitude and the layer's entropy at the same time: layers with few parameters but high entropy are less prone to be removed than layers with more parameters but lower entropy (under the same parameter's norm constraint). Besides, the parameter's magnitude of neurons with zero entropy is not accounted for in the importance score calculation. Symmetrically, to remove parameters from layers having lower pruning irrelevance, we define the inter-layer's pruning relevance score $\mathcal{R}_l$ as:

$$\mathcal{R}_l = \begin{cases} \frac{1}{\mathcal{I}_l} \sum_{j \in L} \mathcal{I}_j & \text{if } \mathcal{I}_l \neq 0 \\ 0 & \text{otherwise.} \end{cases} \tag{14}$$

This measure is as large as the $l$-th layer's pruning irrelevance score is smaller compared to the other layer's. Noticeably, $\mathcal{R}_l \in [1; +\infty)$: to exactly establish how many parameters $\|\boldsymbol{w}_l\|_0^{\text{pruned}}$ should be removed inside each layer $l$ at a given pruning iteration, we have the *entropy-weighted pruned parameter budget*

$$\|\boldsymbol{w}_l\|_0^{\text{pruned}} = \|\boldsymbol{w}\|_0^{\text{pruned}} \cdot \frac{\exp[\mathcal{R}_l]}{\sum_j \exp[\mathcal{R}(j)]}. \tag{15}$$

In Alg. 1, we present a summary of NEPENTHE. Indeed, if a layer has an entropy equal to zero, then all of its neurons have an entropy equal to zero: $\widehat{\mathcal{H}}_l = 0 \Leftrightarrow \mathcal{H}_{l,i} = 0$ , $\forall i$. Hence, this layer doesn't necessarily need to have a rectifier: this layer can be removed entirely without the need for future pruning. Towards this end, we first train the neural network, represented by its weights at initialization $\boldsymbol{w}^{\text{init}}$, on the training set

$\mathcal{D}_{\text{train}}$ (line 2) and evaluate it on the validation set $\mathcal{D}_{\text{val}}$ (line 3). As defined in equation 6, we then calculate the entropy $\widehat{\mathcal{H}}$ on the training set $\mathcal{D}_{\text{train}}$ for each layer $l$ of the considered list of layers $L$ (line 6). This list is initialized to all the layers of the neural network having a rectifier activation (hence, the output layer is excluded).

Considering that $\zeta$ represents the percentage of parameters to remove at each pruning iteration and $\|\boldsymbol{w}\|_0$ the total weight parameters of the considered $L$ layers in the model, we can define the number of weight parameters to be pruned at each iteration $\|\boldsymbol{w}\|_0^{\text{pruned}}$ (line 7) as:

$$\|\boldsymbol{w}\|_0^{\text{pruned}} = \zeta \cdot \|\boldsymbol{w}\|_0. \tag{16}$$

To determine the parameters to prune in each layer, we define a function `Weights_to_prune`, as presented in Alg. 2. This function calculates the weights to remove for each layer and returns a list indicating the number of neurons that need to be removed from each layer, as discussed in Sec. 3.3. At this point, for each layer $l$, the neurons having non-zero entropy are first selected and then $\|\boldsymbol{w}_l\|_0^{\text{pruned}}$ non-zero weights having the lowest absolute magnitude are removed (line 9). The model is then retrained (line 10) and re-evaluated on the validation set $\mathcal{D}_{\text{val}}$ (line 11). The final model is obtained once the performance on the validation set drops below some relative threshold $\theta$.

## 4 Experiments

In this section, we empirically apply unstructured magnitude pruning across multiple architectures and datasets for traditional image classification and natural language processing setups to validate the mechanism: whether unstructured pruning can lower layer entropy and yield zero-entropy, linearizable layers with no performance degradation. We further analyze how pruning affects each layer's entropy and the model's loss landscape sharpness.

Then, we compare NEPENTHE with the iterative magnitude pruning (IMP) baseline from Han et al. (2015). Additionally, in image classification tasks, we compare our results with two other approaches: removing the layers having the lowest sum of weights/gradients. These baselines serve as naive structural criteria that directly measure layer importance without considering activation behavior, allowing us to isolate the specific contribution of entropy-based guidance. We further induce intra-layer sparsity using HRank (Lin et al., 2020), a filter pruning method that removes filters with low-rank feature maps. This comparison highlights the distinction between width-oriented compression and our depth-reduction strategy. In addition, we minimize the group lasso penalty for each layer following the approach of Ochiai et al. (2017). We also compare our method with EGP (Liao et al., 2023), which removes layers through an unstructured pruning process and represents the work most closely related to ours. Furthermore, we demonstrate that our approach can be combined with structured pruning, suggesting promising directions for future pruning method development.

### 4.1 Experimental setup

A variety of setups is covered by evaluating our method on three popular image classification models: ResNet-18 (He et al., 2016), MobileNet-V2 (Howard et al., 2017), and Swin-T (Liu et al., 2021), trained on five datasets: CIFAR-10 (Krizhevsky et al., 2009), Tiny-ImageNet (Le & Yang, 2015), and PACS, VLCS, and SVIRO from DomainBed (Gulrajani & Lopez-Paz, 2021), following the same training policies as Quétu & Tartaglione (2024) and Xu et al. (2021). Moreover, two natural language processing models: BERT (Kenton & Toutanova, 2019) and RoBERTa (Liu

Table 1: Trend in the bottom six layers' entropies for ResNet-18 trained on CIFAR-10.

| **Approach** | $\widehat{\mathcal{H}}_1$ | $\widehat{\mathcal{H}}_2$ | $\widehat{\mathcal{H}}_3$ | $\widehat{\mathcal{H}}_4$ | $\widehat{\mathcal{H}}_5$ | $\widehat{\mathcal{H}}_6$ | **top-1** |
|---|---|---|---|---|---|---|---|
| Dense | 0.647 | 0.680 | 0.728 | 0.785 | 0.791 | 0.797 | 91.66 |
| IMP (iter #1) | 0.585 | 0.650 | 0.699 | 0.725 | 0.767 | 0.778 | 92.29 |
| IMP (iter #2) | 0.506 | 0.580 | 0.647 | 0.654 | 0.700 | 0.722 | 92.25 |
| IMP (iter #3) | 0.256 | 0.623 | 0.658 | 0.672 | 0.682 | 0.737 | 92.46 |
| IMP (iter #4) | 0.192 | 0.660 | 0.667 | 0.676 | 0.698 | 0.763 | 92.27 |
| IMP (iter #5) | 0.136 | 0.589 | 0.648 | 0.727 | 0.728 | 0.791 | 92.44 |
| IMP (iter #6) | 0.093 | 0.447 | 0.640 | 0.650 | 0.764 | 0.765 | 91.89 |
| IMP (iter #7) | 0.055 | 0.335 | 0.487 | 0.592 | 0.640 | 0.775 | 91.66 |
| NEPENTHE | 0 | 0 | 0 | 0.014 | 0.121 | 0.942 | **92.55** |

Table 2: Test performance (top-1) and the number of removed layers (Rem.) for all the considered image classification setups. The results achieved by our method are in italics.

| Model | Approach | CIFAR-10 top-1 | CIFAR-10 Rem. | Tiny-ImageNet top-1 | Tiny-ImageNet Rem. | PACS top-1 | PACS Rem. | VLCS top-1 | VLCS Rem. | SVIRO top-1 | SVIRO Rem. |
|---|---|---|---|---|---|---|---|---|---|---|---|
| ResNet-18 | Dense | 91.66 | 0/17 | 41.44 | 0/17 | 94.70 | 0/17 | 80.89 | 0/17 | 99.93 | 0/17 |
| | Smallest weights | 10.00 | 1/17 | 0.5 | 1/17 | 16.20 | 1/17 | 46.13 | 1/17 | 35.55 | 1/17 |
| | Smallest gradients | 9.29 | 1/17 | 0.5 | 1/17 | 16.20 | 1/17 | 46.13 | 1/17 | 35.55 | 1/17 |
| | Hrank | 91.70 | 0/17 | | – | | – | | – | | – |
| | Group lasso | 92.11 | 1/17 | 38.92 | 0/17 | 81.20 | 0/17 | 67.85 | 0/17 | 99.57 | 0/17 |
| | IMP | 91.66 | 0/17 | 39.14 | 0/17 | 89.80 | 0/17 | 74.09 | 0/17 | 99.45 | 0/17 |
| | EGP | 92.18 | **3/17** | 39.50 | 4/17 | 84.30 | 2/17 | 74.28 | **2/17** | 98.66 | 5/17 |
| | *NEPENTHE* | **92.55** | **3/17** | **39.56** | **5/17** | **90.10** | **3/17** | **78.38** | **2/17** | **99.61** | **8/17** |
| MobileNet-V2 | Dense | 93.68 | 0/35 | 45.86 | 0/35 | 93.20 | 0/35 | 81.83 | 0/35 | 99.95 | 0/35 |
| | Smallest weights | 10.00 | 1/35 | 0.50 | 1/35 | 18.50 | 1/35 | 6.43 | 1/35 | 35.55 | 1/35 |
| | Smallest gradients | 10.00 | 1/35 | 46.62 | 1/35 | 16.20 | 1/35 | 46.13 | 1/35 | 35.55 | 1/35 |
| | Hrank | 91.73 | 0/35 | | – | | – | | – | | – |
| | Group lasso | 83.00 | 4/35 | 47.1 | 0/35 | 92.10 | 0/35 | 78.84 | 0/35 | 99.93 | 0/35 |
| | IMP | 92.50 | 0/35 | 45.24 | 0/35 | 91.40 | 0/35 | 79.43 | 0/35 | 99.95 | 0/35 |
| | EGP | 92.22 | 6/35 | 47.52 | 6/35 | 17.70 | **3/35** | 45.85 | **2/35** | 35.05 | **2/35** |
| | *NEPENTHE* | **93.26** | **7/35** | **47.92** | **12/35** | **92.20** | *1/35* | **80.06** | **2/35** | **99.98** | **2/35** |
| Swin-T | Dense | 91.54 | 0/12 | 75.60 | 0/12 | 97.10 | 0/12 | 86.58 | 0/12 | 99.95 | 0/12 |
| | Smallest weights | 89.22 | **2/12** | **75.12** | 1/12 | **96.10** | **2/12** | 84.62 | **1/12** | 99.70 | 4/12 |
| | Smallest gradients | 89.21 | **2/12** | 74.54 | 1/12 | 95.70 | **2/12** | 84.15 | **1/12** | 99.55 | 4/12 |
| | Hrank | 91.87 | 0/12 | | – | | – | | – | | – |
| | Group lasso | 91.68 | 0/12 | 71.30 | 0/12 | 94.30 | 0/12 | 84.81 | 0/12 | 99.69 | 0/12 |
| | IMP | 90.53 | 0/12 | 67.56 | 0/12 | 93.90 | 0/12 | 80.06 | 0/12 | 99.75 | 0/12 |
| | EGP | 92.01 | 1/12 | 71.48 | **1/12** | 93.50 | 1/12 | 82.95 | **1/12** | 99.64 | **5/12** |
| | *NEPENTHE* | **92.29** | **2/12** | *72.58* | **1/12** | *95.10* | **2/12** | **85.27** | **1/12** | **99.75** | **5/12** |

et al., 2019) are trained on three datasets: SST-2 (Socher et al., 2013), QNLI (Williams et al., 2018), and RTE (Bentivogli et al., 2009), following the training strategies of Peer et al. (2022). To verify the generality of the layer collapse phenomenon induced by unstructured pruning across models with different depths and widths, as well as on datasets of higher complexity, NEPENTHE has also been implemented for ResNet-50, ResNet-152, and MobileNetV2-0.75 models trained on CIFAR-10, as well as ResNet-18 trained on Ima-geNet (Deng et al., 2009). Results appear in Appendix D.2. In all the setups, we set $\zeta = 0.5$ for ResNet-18, $\zeta = 0.25$ for Swin-T, and $\zeta = 0.1$ for MobileNet-V2. Moreover, we set $\zeta = 0.25$ (respectively $\zeta = 0.15$) for the models trained on QNLI and RTE (respectively SST-2). All the hyperparameters, augmentation strategies, learning policies, and a study to choose $\zeta$ are provided in Appendix C.

## 4.2 Trend of Layer's Entropy and Model's Sharpness

First, we study the effect of pruning on the layer's entropy. Table 1 reports the entropy trend of the six layers having the lowest entropy for ResNet-18 trained on CIFAR-10. As expected from the derivation in Sec. 3.1, as the pruning progresses (and implicitly as $t$ grows), the entropy is naturally decreased, showcasing very small values after some pruning iterations. However, we also observe that as the entropy $\widehat{H}_1$ decreases, the top-1 accuracy begins to deteriorate. This occurs without proper pruning reallocation, unlike in NEPENTHE with equation 14. Indeed, in this case, our method not only preserves the model performance (even improves it), but we can also successfully remove three layers from

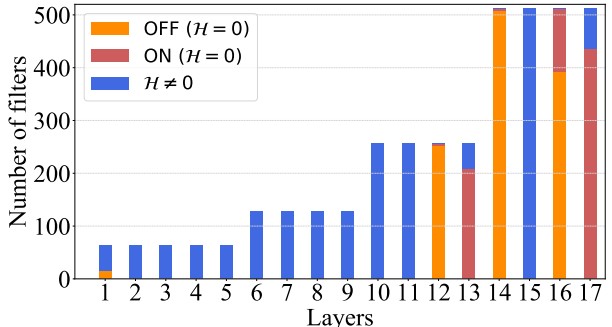

Figure 3: Filter states per layer for ResNet-18 trained on CIFAR-10 pruned by NEPENTHE.

Table 3: Trend for the model's Sharpness for Swin-T trained on CIFAR-10.

| Approach | Sharpness | top-1 | Rem. |
|---|---|---|---|
| Dense | 20.62 | 91.54 | 0/12 |
| IMP (iter #1) | 6.78 | 91.73 | 0/12 |
| IMP (iter #2) | 3.70 | 91.80 | 0/12 |
| IMP (iter #3) | 3.18 | 91.43 | 0/12 |
| IMP (iter #4) | 4.78 | 91.46 | 0/12 |
| IMP (iter #5) | 5.39 | 91.27 | 0/12 |
| IMP (iter #6) | 6.81 | 91.05 | 0/12 |
| IMP (iter #7) | 7.66 | 90.53 | 0/12 |
| NEPENTHE (iter #1) | 7.71 | 91.77 | 0/12 |
| NEPENTHE (iter #2) | 2.82 | 92.06 | 0/12 |
| NEPENTHE (iter #3) | 1.58 | 92.08 | 0/12 |
| NEPENTHE (iter #4) | 1.17 | 92.31 | 0/12 |
| NEPENTHE (iter #5) | 1.85 | 92.29 | 2/12 |
| NEPENTHE (iter #6) | 1.62 | 92.21 | 2/12 |
| NEPENTHE (iter #7) | 1.72 | 92.11 | 2/12 |

Table 4: Test performance (top-1) and the number of removed layers (Rem.) for all the considered NLP setups. The results achieved by our method are in italics.

| Dataset | Approach | BERT | | RoBERTa | |
|---|---|---|---|---|---|
| | | top-1 | Rem. | top-1 | Rem. |
| QNLI | Dense | 90.48 | 0/12 | 92.18 | 0/12 |
| | Smallest weights | 88.15 | 3/12 | 85.93 | 2/12 |
| | Smallest gradients | 88.44 | 3/12 | 86.84 | 2/12 |
| | IMP | 85.87 | 0/12 | **88.10** | 0/12 |
| | EGP | 87.90 | 4/12 | 84.66 | 2/12 |
| | *NEPENTHE* | **88.69** | **4/12** | *87.41* | **2/12** |
| RTE | Dense | 61.01 | 0/12 | 66.79 | 0/12 |
| | Smallest weights | 46.93 | 1/12 | 65.81 | 1/12 |
| | Smallest gradients | 55.23 | 1/12 | 62.20 | 1/12 |
| | IMP | 57.76 | 0/12 | 62.82 | 0/12 |
| | EGP | 57.73 | 1/12 | 52.71 | 1/12 |
| | *NEPENTHE* | **58.12** | **4/12** | **66.06** | **1/12** |
| SST-2 | Dense | 92.20 | 0/12 | 92.66 | 0/12 |
| | Smallest weights | 88.14 | 3/12 | 89.43 | 4/12 |
| | Smallest gradients | 87.25 | 3/12 | 89.43 | 4/12 |
| | IMP | 88.65 | 0/12 | 88.76 | 0/12 |
| | EGP | 85.09 | 3/12 | 86.47 | 4/12 |
| | *NEPENTHE* | **88.99** | **3/12** | **89.79** | **4/12** |

the model. Noticeably, $\widehat{H}_4$ and $\widehat{H}_5$ are also very low, while already starting from $\widehat{H}_6$, the entropy is very high. As opposed to IMP where in general the entropy lies in intermediate-range values, NEPENTHE tries to push all the encoded information toward layers having already high entropy, enabling effective layer removal with little (or in this case no) performance loss. This is also illustrated in Fig. 3, showing the distribution of the filter states per layer for the same setup. Our unstructured pruning approach effectively removes three layers by pushing all the neurons inside low-entropy layers to be either in the ON or in the OFF state. Besides, we also notice that in some layers (like 13 and 17), entire units reach zero entropy, indicating that structured sparsity can naturally emerge from an unstructured pruning process, as previously reported in related works (Han et al., 2015; Tartaglione et al., 2021).

Moreover, we analyze the layers pruned from a ResNet-18 trained on CIFAR-10 by other baselines in Appendix D.1. While with NEPENTHE, the layers with the lowest entropy are typically found near the deepest layers of the network, the layers near the output are often pruned first with EGP (Fig. 6c). This happens because the layers near the output usually capture highly task-specific or redundant features whose removal causes limited disturbance to earlier representations, allowing the network to retain most of its predictive capacity. In contrast, methods that prune layers based on the lowest sum of weights/gradients tend to remove layers near the input of the model first(Fig. 6a and Fig. 6b). These early layers have fewer parameters and smaller cumulative sums, making them appear less important. Once removed, however, the forward signal is broken and the model's accuracy drops to a random level.

Table 3 shows how pruning affects loss landscape sharpness for Swin-T on CIFAR-10 trained with IMP and NEPENTHE. Following Lee et al. (2025), we compute the maximum Hessian eigenvalues to measure the sharpness of the models. Lower sharpness values correspond to flatter loss surfaces. As in previous studies (Liebenwein et al., 2021), IMP is able to reduce the model's loss landscape sharpness at moderate or low sparsity, but the sharpness increases at higher sparsity. Unlike IMP, NEPENTHE can significantly reduce the model's loss landscape and improve generalization.

### 4.3 Main Results

**Image classification tasks.** Table 2 shows the test performance (top-1) and the number of removed layers (Rem.) for all the considered image classification setups. Removing layers with the lowest weight/gradient sums only works for Swin-T. On ResNet-18 and MobileNet-V2, these methods reduce accuracy to random levels after one layer removal. Since Hrank operates at the level of the neuron, even though it can help models maintain a good (or even better) performance after pruning, no layer can be removed with this method.

Table 5: Ablation study on ResNet-18 trained on CIFAR-10. Each component contributes to the effectiveness of NEPENTHE.

| Entropy-based budget | Don't care state | Neurons selection | top-1 | Rem. |
|:---:|:---:|:---:|:---:|:---:|
| | | | 91.66 | 0/17 |
| ✓ | | | 92.18 | 3/17 |
| ✓ | ✓ | | 92.33 | 3/17 |
| ✓ | ✓ | ✓ | 92.55 | 3/17 |
| Depth reduced model trained from initialization | | | 91.57 | 3/17 |

Table 6: MobileNet-V2 with ReLU6 trained on CIFAR-10 dataset.

| Method | top-1 | Rem. |
|:---:|:---:|:---:|
| Dense | 93.68 | 0/35 |
| NEPENTHE | 93.55 | 1/35 |
| NEPENTHE | 93.25 | 2/35 |
| NEPENTHE | 93.37 | 4/35 |
| NEPENTHE | 93.15 | 5/35 |
| NEPENTHE | 93.26 | 7/35 |
| NEPENTHE | 92.78 | 9/35 |

Thus, we report Hrank results only for CIFAR-10 to save computational resources. Moreover, although minimizing the group lasso penalty has little impact on the performance, its effectiveness in layer removal is not significant. Furthermore, IMP does not support the removal of any layers despite successfully maintaining performance. However, EGP enables the removal of some layers but at the expense of compromising generalizability. For example, on MobileNet-V2 trained on PACS, EGP removes three layers outright, leading to a significant drop in accuracy. In contrast, NEPENTHE produces models with a substantial number of removable layers with little (or no) performance loss compared to the dense baseline. For instance, on MobileNet-V2 trained on Tiny-ImageNet, NEPENTHE successfully removes 12 layers while even improving the top-1 accuracy by about 2%.

**NLP tasks.** The results for all NLP setups are presented in Table 4. Similarly to what was observed for image classification setups, we observe that while IMP does not significantly harm performance, it does not support whole-layer removal. In contrast, NEPENTHE produces models with a significant number of removable layers while maintaining a performance comparable to the dense models.

## 4.4 Combination of Unstructured Pruning and Structured Pruning Methods

Besides pure unstructured pruning, we also explore the potential of combining NEPENTHE with structured layer removal methods to extend its applicability. In particular, we integrate the group lasso regularization used in SSL with NEPENTHE. Fig. 4 shows the results of this combination. Although a slight drop in accuracy is observed when fewer layers are pruned compared with pure NEPENTHE, the joint approach enables the removal of a larger number of layers while maintaining competitive performance. This demonstrates NEPENTHE's flexibility and its potential to serve as a general framework that can be coupled with various structured sparsity or layer-selection techniques to achieve both entropy-driven and structure-aware depth reduction.

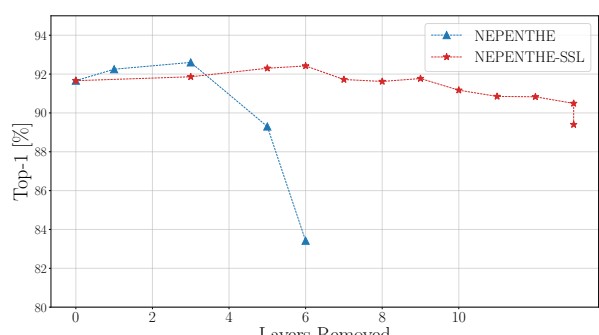

Figure 4: Test performance (top-1) and the number of removed layers (Rem.) for ResNet-18 trained on CIFAR-10.

## 4.5 Ablation Study

In this section, we conduct several studies. First, we perform a classical ablation to analyze the contribution of each term within NEPENTHE. Second, we verify the layer collapse phenomenon induced by unstructured pruning using some of the most common two-interval rectifiers. Then, we extend this validation to rectifiers with more intervals. Moreover, we test the phenomenon beyond magnitude-based pruning by applying it to Wanda. Finally, we demonstrate the practical benefits of our approach in terms of inference time, memory usage, and energy consumption.

Table 7: Test performance (top-1) and the number of removed layers (Rem.) for ResNet-18 trained on CIFAR-10 dataset and pruned by combining NEPENTHE and Wanda.

| Method | top-1 | Rem. |
|---|---|---|
| Dense | 91.66 | 0/17 |
| NEPENTHE Wanda | 92.67 | 1/17 |
| NEPENTHE Wanda | 89.80 | 3/17 |
| NEPENTHE Wanda | 89.37 | 5/17 |
| NEPENTHE Wanda | 88.31 | 6/17 |

Table 8: The accuracy and number of layers removable for Llama 3.1-8B (16FP) evaluated on MMLU - High school US history and pruned by NEPENTHE without fine-tuning (no ft).

| Method | Val top-1 | Test top-1 | Rem. |
|---|---|---|---|
| Dense | 40.91 | 33.82 | 0/32 |
| NEPENTHE (no ft) | 38.49 | 36.76 | 3/32 |
| NEPENTHE (no ft) | 36.36 | 29.90 | 5/32 |
| NEPENTHE (no ft) | 33.82 | 35.08 | 7/32 |
| NEPENTHE (no ft) | 31.82 | 28.92 | 9/32 |
| NEPENTHE (no ft) | 18.18 | 27.45 | 11/32 |

First, Table 5 provides an ablation study on the three key components identifiable within NEPENTHE: the entropy-based weighted pruned parameter budget (equation 13), the presence of the don't care state in the entropy formulation (equation 2), and the filtering mechanism of non-zero entropy neurons (equation 13). In table 5, we also report a depth-matched baseline trained from initialization. The results show that each component contributes to the overall effectiveness of NEPENTHE, jointly enhancing both pruning stability and the emergence of layer collapse.

Second, Table 9 shows the test performance of ResNet-18 on CIFAR-10 and pruned by NEPENTHE, for different rectifiers. The fact that unstructured pruning induces layer collapse is not dependent on any particular rectifier and can be effective with any, since our method removes three layers without performance loss for all the tested activations.

Table 9: Different activation functions on ResNet-18 trained on CIFAR-10.

| Activation | Method | top-1 | Rem. |
|---|---|---|---|
| ReLU | Dense | 91.66 | 0/17 |
| | NEPENTHE | 92.55 | 3/17 |
| SiLU | Dense | 91.66 | 0/17 |
| | NEPENTHE | 92.77 | 3/17 |
| PReLU | Dense | 91.25 | 0/17 |
| | NEPENTHE | 92.27 | 3/17 |
| LeakyReLU | Dense | 91.66 | 0/17 |
| | NEPENTHE | 92.49 | 3/17 |
| GELU | Dense | 91.89 | 0/17 |
| | NEPENTHE | 92.57 | 3/17 |

Moreover, we evaluate NEPENTHE on MobileNet-V2 with ReLU6 trained on CIFAR-10. Table 6 shows the results, indicating that NEPENTHE can remove layers with minimal performance loss. Note that ReLU6 is a rectifier divided into three intervals, suggesting that this layer collapse phenomenon also appears in models with rectifiers that have more states. We present the theory analysis for rectifiers with more than two intervals in Appendix B.

Furthermore, Table 7 presents the results of applying Wanda pruning within the NEPENTHE framework for ResNet-18 on CIFAR-10. In this setting, we replace magnitude pruning with Wanda as the underlying pruning criterion. With one layer removed, the model even achieves better performance, confirming that the layer collapse phenomenon induced by unstructured pruning is not limited to magnitude-based strategies but also arises under other popular methods such as Wanda.

Finally, Table 10 showcases the potential savings in terms of inference time, memory usage, and energy consumption on an NVIDIA A4500 GPU for a ResNet-18 trained on CIFAR-10 with NEPENTHE: the fewer layers the network has, the lower the inference time, the smaller the memory usage and energy consumption. Note that FLOPs do not necessarily decrease due to the fact that composing consecutive linear operators may not preserve unstructured sparsity; we clarify this point in Appendix D.3.

Table 10: Inference time [ms], Memory usage [MBs] and Energy consumption [mJ] of ResNet-18 on CIFAR-10 on a NVIDIA A4500.

| Rem. | Inference time [ms] | Mem.usage [MBs] | Energy [mJ] | top-1 |
|---|---|---|---|---|
| 0/17 | 3.32 | 230 | 498.7 | 91.66 |
| 1/17 | 3.27 | 202 | 490.2 | 92.25 |
| 3/17 | 2.96 | 170 | 444.0 | 92.55 |
| 5/17 | 2.60 | 60 | 389.7 | 89.30 |

### 4.6 Limitations

The work here presented shows two major limitations that will be tackled in future work.

The first limitation is related to the established theoretical framework. While our main objective is to shed light over the mechanism by which unstructured pruning drives rectifier-activated layers toward a low-entropy regime, the analysis is limited to ReLU activations. We derive in the supplementary material error bounds for other popular rectifiers; nevertheless this results in a limitation of the conducted analysis. Moreover, since $\widehat{\mathcal{H}}_l$ averages first-order neuron entropies, that represents nothing but a proxy for linearizability rather than the joint entropy of the layer's activation-state pattern.

The second limitation involves the iterative nature of NEPENTHE can limit its direct application to very large models, as repeated pruning and retraining cycles are computationally demanding. To keep the study tractable, our main experiments focus on moderate-scale architectures, with the corresponding training time analysis provided in Appendix D.4. Nevertheless, to illustrate the broader applicability of our approach, we apply NEPENTHE to a pre-trained Llama 3.1-8B model without any fine-tuning. As shown in Table 8, the method still performs reliably, indicating that it can scale to larger language models and remain effective in more complex scenarios. Future work will explore strategies to reduce iteration cost and further improve scalability.

## 5 Conclusion

In this work, we show that unstructured pruning can inherently induce structural effects in deep neural networks. Specifically, in rectifier-activated architectures, it naturally reduces neuron entropy, and when a layer's average entropy reaches zero, that layer becomes linearizable and can be safely removed without degrading representational capacity. Building on this observation, we introduced NEPENTHE, an entropy-guided unstructured pruning framework that reallocates pruning budgets toward low-entropy layers. This enables networks to collapse redundant layers while preserving or even improving accuracy. Across diverse architectures and datasets, NEPENTHE removes layers with minimal or no performance loss and yields flatter landscapes, better generalization, and efficiency gains.

Beyond empirical results, this study provides a new theoretical perspective: unstructured pruning can serve as an implicit depth-regularization mechanism, driving networks toward simpler and more linear representations. We hope that this study encourages further investigation into how apparently "unstructured" operations can yield structural behaviors in deep networks.

### Acknowledgments

This work received support from the European Union's HORIZON Research and Innovation Programme under grant agreement No. 101120657, as part of the ENFIELD project (European Lighthouse to Manifest Trustworthy and Green AI). Additionally, funding was provided by the French National Research Agency (ANR) under grant agreements ANR-22-PEFT-0003 and ANR-22-PEFT-0007, as part of the France 2030 initiative, specifically the NF-NAI and NF-FITNESS projects. This work was also supported by the French National Research Agency (ANR) in the framework of the IA Cluster project "Hi! PARIS Cluster 2030" under Grant ANR-23-IACL-005, and by the Hi! PARIS Center on Data Analytics and Artificial Intelligence. Zhu Liao acknowledges financial support from the China Scholarship Council(CSC).

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

# A   Approximation Error for Smooth Rectifiers

For smooth rectifiers (e.g., GELU/SiLU), $\psi(z)$ is not exactly the identity even when $z > 0$, and that collapsing such layers is not a mathematically lossless operation. In this section, we formalize the approximation error and show why this error is negligible under the training and pruning regime considered in this work.

Consider a neuron with pre-activation

$$z(x) = w^\top x + b, \tag{17}$$

and a smooth rectifier activation $\phi \in \{\text{GELU}, \text{SiLU}\}$. Removing the activation corresponds to replacing $\phi(z)$ by the identity map $z$. The induced neuron-level approximation error is

$$\varepsilon(x) := \phi(z(x)) - z(x). \tag{18}$$

Since layer removal is evaluated at the dataset level, the relevant quantity is the expected squared error

$$\mathcal{E} := \mathbb{E}_{x \sim \mathcal{D}}\big[\|\phi(z(x)) - z(x)\|^2\big]. \tag{19}$$

Both GELU and SiLU admit the decomposition

$$\phi(z) = z - r(z), \tag{20}$$

where $r(z)$ denotes the residual nonlinearity measuring the deviation from the identity map. For these smooth rectifiers, $r(z)$ exhibits strong tail decay in the positive regime:

$$|r(z)| \le C\exp(-\alpha z^2) \quad \text{for GELU}, \qquad |r(z)| \le C\exp(-\alpha z) \quad \text{for SiLU}, \tag{21}$$

for some constants $C, \alpha > 0$. As a consequence, $\phi'(z) \to 1$ and $\phi''(z) \to 0$ exponentially fast as $z \to +\infty$, implying that deviations from linearity are concentrated in a narrow neighborhood around $z \approx 0$.

In our setting, the distribution of pre-activations $z$ is jointly shaped by three mechanisms: (i) $\ell_2$ weight decay controls the operator norms and variance of the weights; (ii) magnitude-based unstructured pruning removes small-magnitude parameters, further reducing variance; (iii) entropy minimization enforces that neurons remain almost always in the same activation regime.

As shown in Sec. 3.2, the combined effect of these mechanisms is that pre-activations become increasingly concentrated and biased toward a single regime. In particular, in the almost-always-ON case, the distribution of $z$ can be well approximated by

$$z \sim \mathcal{N}(\mu_z, \sigma_z^2), \qquad \text{with } \mu_z \gg \sigma_z, \tag{22}$$

Using the decay properties of the residual nonlinearity $r(z)$ established above, the expected neuron-level approximation error can be bounded as

$$\mathbb{E}\big[\varepsilon(z)^2\big] = \mathbb{E}\big[r(z)^2\big] \;\le\; \int r(z)^2\, \mathcal{N}(z; \mu_z, \sigma_z^2)\, dz \;\le\; C\exp(-\alpha\mu_z^2), \tag{23}$$

for GELU (and analogously exponential decay in $\mu_z$ for SiLU). Therefore, once a neuron enters a low-entropy (almost-always-ON) regime, the approximation error induced by replacing $\phi(z)$ with the identity decays exponentially in the mean pre-activation and rapidly becomes negligible in practice.

Let $\delta_\ell$ denote the approximation error introduced by collapsing layer $\ell$. The propagated error satisfies

$$\|\delta_{\ell+1}\| \le \|W_{\ell+1}\|\,\|\delta_\ell\|. \tag{24}$$

Under $\ell_2$ regularization, the operator norms $\|W_\ell\|$ remain controlled and empirically decrease with pruning, which also consistent with the observed reduction in loss landscape sharpness (Table. 3). Consequently, over $k$ collapsed layers, the accumulated output error admits the bound

$$\|\delta_{\text{out}}\| \;\le\; \sum_{\ell=1}^{k} \Big(\prod_{j>\ell} \|W_j\|\Big) \varepsilon_\ell, \tag{25}$$

which remains small since each $\varepsilon_\ell$ is exponentially suppressed. This explains why approximation errors introduced by collapsing multiple layers do not accumulate catastrophically in deep networks.

# B  Analysis for Rectifiers with More Intervals

Unstructured pruning can still induce layer collapse in networks where rectifiers have more intervals. Let us take the rectifier ReLU6 and apply NEPENTHE as an example. If $z_{l,i}^{\boldsymbol{x}}$ is the output of the $i$-th neuron inside the $l$-th layer from a given input $\boldsymbol{x}$ of the dataset $\mathcal{D}$, the output $y_{l,i}^{\boldsymbol{x}}$ is:

$$y_{l,i}^{\boldsymbol{x}} = \min(\max(0, z_{l,i}^{\boldsymbol{x}}), 6), \tag{26}$$

There are two OFF states:

- $z_{l,i} \leq 0$. In this case, when employing a ReLU6, the output of the $i$-th neuron is always 0, we call this state OFF-. If a neuron has all its output in this state, this neuron can be simply pruned.

- $z_{l,i} \geq 6$. In this case, when employing a ReLU6, the output of the $i$-th neuron is always 6, we call this state OFF+. If a neuron has all its output in this state, this neuron can be replaced by a bias 6.

There is also one ON state:

- $0 \leq z_{l,i} \leq 6$. In this case, the output of the i-th neuron's rectifier is always the same as its input, we call this state ON. If a neuron has all its output in this state, this neuron can in principle be absorbed by the following layer as there is no non-linearity between them anymore.

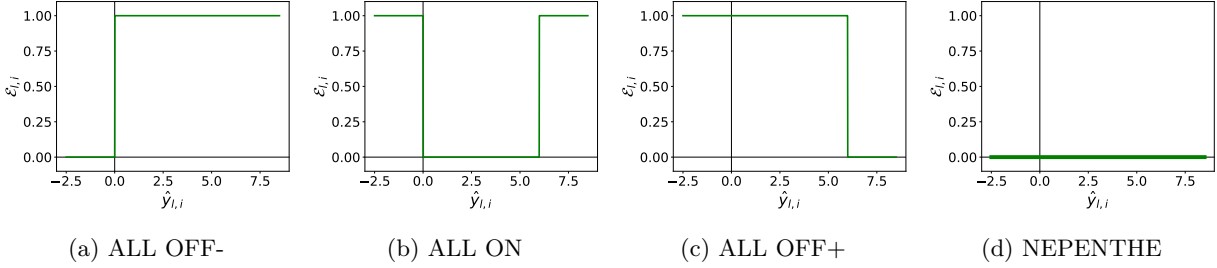

(a) ALL OFF-   (b) ALL ON   (c) ALL OFF+   (d) NEPENTHE

Figure 5: Error plot for the i-th neuron of the l-th ReLU6 activated layer as a function of the output average $\hat{y}_{l,i}$.

Fig. 5a shows how a neuron's error varies with its average output $\hat{y}_{l,i}$ when substituting the rectifier with a null function, which means pushing all the neurons to always OFF- state. Fig. 5b shows the error of pushing all the neurons to always ON state by substituting the rectifier $\psi_l$ by an identity function. Fig. 5c shows the error of pushing all the neurons to always OFF+ state by substituting the rectifier $\psi_l$ by a bias 6. Each of the above methods produces obvious errors in some cases. As shown in Fig. 5d, NEPENTHE can remove layers without introducing any error. It appears that the phenomenon that unstructured pruning induces layer collapse also appears in networks with rectifiers with more intervals rather than two.

# C  Details on the Learning Strategies Employed

The implementation details used in this paper are presented here.

Like in the He et al. (2022)'s setup, for the ResNet-18 network, a modified version of the `torchvision` model is used: the first convolutional layer is set with a filter of size $3 \times 3$ and the max-pooling layer that follows has been eliminated to adapt ResNet-18 for CIFAR-10.
CIFAR-10 is augmented with per-channel normalization, random horizontal flipping, and random shifting by up to four pixels in any direction. For the datasets of DomainBed, the images are augmented with per-channel normalization, random horizontal flipping, random cropping, and resizing to 224. The brightness, contrast, saturation, and hue are also randomly affected with a factor fixed to 0.4. Tiny ImageNet is augmented

Table 11: Iterations (Iter.), Test performance (top-1), and the number of removed layers (Rem.) for MobileNet-V2 trained on CIFAR-10 and pruned by NEPENTHE with different $\zeta$

| (a) $\zeta = 0.05$ | | | (b) $\zeta = 0.1$ | | | (c) $\zeta = 0.25$ | | |
|---|---|---|---|---|---|---|---|---|
| Iter. | top-1 | Rem. | Iter. | top-1 | Rem. | Iter. | top-1 | Rem. |
| 1 | 93.42 | 1/35 | 1 | 93.55 | 1/35 | 1 | 94.12 | 1/35 |
| 3 | 93.56 | 2/35 | 3 | 93.14 | 2/35 | 3 | 10 | 18/35 |
| 7 | 93.36 | 5/35 | 7 | 93.26 | 7/35 | 7 | - | - |
| 9 | 93.54 | 7/35 | 9 | 92.78 | 9/35 | 9 | - | - |

Table 12: Table of the different employed learning strategies.

| Model | Dataset | Epochs | Batch | Opt. | LR | $\beta_1$ | $\beta_2$ | $\epsilon$ |
|---|---|---|---|---|---|---|---|---|
| BERT | QNLI | 3 | 32 | AdamW | 2e-5 | 0.9 | 0.999 | 1e-8 |
| RoBERTa | QNLI | 3 | 32 | AdamW | 2e-5 | 0.9 | 0.999 | 1e-8 |
| BERT | RTE | 3 | 32 | AdamW | 2e-5 | 0.9 | 0.999 | 1e-8 |
| RoBERTa | RTE | 3 | 32 | AdamW | 2e-5 | 0.9 | 0.999 | 1e-8 |
| BERT | SST-2 | 3 | 32 | AdamW | 2e-5 | 0.9 | 0.999 | 1e-8 |
| RoBERTa | SST-2 | 3 | 32 | AdamW | 2e-5 | 0.9 | 0.999 | 1e-8 |

with per-channel normalization and random horizontal flipping. ImageNet is augmented with per-channel normalization, random horizontal flipping, random cropping, and resizing to 224. The sequence length of SST-2, QNLI from Williams et al, and RTE is set to 128.

All weights from ReLU-actived layers are set as prunable for ResNet-18. For Swin-T, BERT, and RoBERTa, all weights from GELU-activated layers are prunable. while for MobileNetv2 all weights from ReLU6-activated layers are considered in the pruning. Neither biases nor batch normalization parameters are pruned.

$\zeta$ is a crucial parameter. As Table 11 shows, in our approach, since we utilize iterative pruning, if the sparse ratio is set too low, more iterations will be required to get removable layers (because we remove fewer parameters at a time). On the other hand, if the sparse ratio is set too high, the model's performance will be destroyed quickly (because essential parameters are removed). Currently, we follow commonly used sparse ratios from unstructured pruning.

The training hyperparameters used in the Image classification experiments are presented in Table 13, hyper-parameters used in the NLP experiments are presented in Table 12. We have performed our experiments on an NVIDIA A4500 GPU. Our code is attached to this supplementary material and will be publicly available upon acceptance of the article.

# D  More Detailed Results

## D.1  Layer States for Models Trained on CIFAR-10

Fig. 6 shows the layers' states for ResNet-18 trained on CIFAR-10 with different methods.

## D.2  Experiments on More Architectures

Table 23 presents the results for ResNet-50, ResNet-152, and MobileNetV2-0.75 models trained on Cifar-10 dataset. Table 24 presents the results for ResNet-18 model trained on ImageNet dataset.

We also provide an ablation study on the three key components identifiable within NEPENTHE for Swin-T trained on Tiny-ImageNet. As shown in Table 14, each component contributes to the overall effectiveness of NEPENTHE.

Table 13: Table of the different employed learning strategies.

| Model | Dataset | Epochs | Batch | Opt. | Mom. | LR | Milestones | Drop Factor | Weight Decay |
|---|---|---|---|---|---|---|---|---|---|
| ResNet-18 | CIFAR-10 | 160 | 128 | SGD | 0.9 | 0.1 | [80, 120] | 0.1 | 1e-4 |
| ResNet-50 | CIFAR-10 | 160 | 128 | SGD | 0.9 | 0.1 | [80, 120] | 0.1 | 1e-4 |
| ResNet-152 | CIFAR-10 | 160 | 128 | SGD | 0.9 | 0.1 | [80, 120] | 0.1 | 1e-4 |
| Swin-T | CIFAR-10 | 160 | 128 | SGD | 0.9 | 0.001 | [80, 120] | 0.1 | 1e-4 |
| MobileNetv2 | CIFAR-10 | 160 | 128 | SGD | 0.9 | 0.1 | [80, 120] | 0.1 | 1e-4 |
| MobileNetV2-0.75 | CIFAR-10 | 160 | 128 | SGD | 0.9 | 0.1 | [80, 120] | 0.1 | 1e-4 |
| ResNet-18 | PACS | 30 | 16 | SGD | 0.9 | 0.001 | [24] | 0.1 | 5e-4 |
| Swin-T | PACS | 30 | 16 | SGD | 0.9 | 0.001 | [24] | 0.1 | 5e-4 |
| MobileNetv2 | PACS | 30 | 16 | SGD | 0.9 | 0.001 | [24] | 0.1 | 5e-4 |
| ResNet-18 | VLCS | 30 | 16 | SGD | 0.9 | 0.001 | [24] | 0.1 | 5e-4 |
| Swin-T | VLCS | 30 | 16 | SGD | 0.9 | 0.001 | [24] | 0.1 | 5e-4 |
| MobileNetv2 | VLCS | 30 | 16 | SGD | 0.9 | 0.001 | [24] | 0.1 | 5e-4 |
| ResNet-18 | SVIRO | 30 | 16 | SGD | 0.9 | 0.001 | [24] | 0.1 | 5e-4 |
| Swin-T | SVIRO | 30 | 16 | SGD | 0.9 | 0.001 | [24] | 0.1 | 5e-4 |
| MobileNetv2 | SVIRO | 30 | 16 | SGD | 0.9 | 0.001 | [24] | 0.1 | 5e-4 |
| ResNet-18 | Tiny ImageNet | 160 | 128 | SGD | 0.9 | 0.1 | [80, 120] | 0.1 | 1e-4 |
| Swin-T | Tiny ImageNet | 160 | 128 | SGD | 0.9 | 0.001 | [80, 120] | 0.1 | 1e-4 |
| MobileNetv2 | Tiny ImageNet | 160 | 128 | SGD | 0.9 | 0.1 | [80, 120] | 0.1 | 1e-4 |
| ResNet-18 | ImageNet | 90 | 128 | SGD | 0.9 | 0.1 | [30, 90] | 0.1 | 1e-4 |

Table 14: Ablation study on Swin-T trained on Tiny-ImageNet. Each component contributes to the effectiveness of NEPENTHE.

| Entropy-based budget | Don't care state | Neurons selection | top-1 | Rem. |
|---|---|---|---|---|
| | | | 75.60 | 0/12 |
| ✓ | | | 71.48 | 1/12 |
| ✓ | ✓ | | 71.54 | 1/12 |
| ✓ | ✓ | ✓ | 72.85 | 1/12 |

### D.3 Potential Computational Savings

Table 15, 16, 17, 18, 19, 20, 21, 22 present the potential savings in terms of FLOPs, latency, and memory across different tasks trained by NEPENTHE and on different hardware. It appears that in general, the fewer layers the networks have, the lower the inference time.

Moreover, for unstructured sparsity, the FLOPs of the composed operator do not necessarily decrease, since multiplying two sparse matrices may introduce sparsity fill-in and yield a denser effective operator. However, this concern does not apply to the computational setting considered in our work, and we clarify the distinction here.

Our statement that a linearized layer can be "absorbed by the following layer" is an *algebraic* statement about function composition:

$$W_{l+1}(W_l x) = (W_{l+1} W_l)x. \tag{27}$$

Importantly, NEPENTHE does not rely on performing runtime multiplication of two unstructured sparse matrices. Once a layer reaches (near-)zero entropy and is deemed linearizable, it is removed entirely from the computational graph, and the model is recompiled with reduced depth. This is a layer removal operation, not a sparse-sparse fusion executed at inference time.

We do not claim that the product $W_{l+1} W_l$ preserves the sparsity pattern of the original matrices. Indeed, sparsity fill-in is expected in general. However, our method does not require sparsity preservation in the merged operator, because the primary source of computational savings in our work is depth reduction, i.e., shortening the critical path of forward propagation, rather than exploiting sparse matrix kernels.

The reductions reported in our efficiency evaluation are based on actual measured inference time, memory usage, and energy consumption, rather than on idealized sparse FLOP counts. On modern GPUs, unstructured

Table 15: MFLOPs, Inference time [ms], Memory usage [MBs], and Energy consumption [mJ] of ResNet-18 on CIFAR-10 across different hardware platforms.

| Hardward | Rem. | MFLOPs | Inference time [ms] | Mem.usage [MBs] | top-1 |
|---|---|---|---|---|---|
| NVIDIA A4500 | 0/17 | 725.47 | 3.32 | 230 | 91.66 |
|  | 3/17 | 231.79 | 2.96 | 170 | 92.55 |
| RTX 2080 | 0/17 | 725.47 | 3.32 | 230 | 91.66 |
|  | 3/17 | 231.79 | 2.96 | 170 | 92.55 |
| RTX 4000 | 0/17 | 725.47 | 4.52 | 241 | 91.66 |
|  | 3/17 | 231.79 | 3.94 | 212 | 92.55 |
| Jetson Nano | 0/17 | 725.47 | 8.52 | 241 | 91.66 |
|  | 3/17 | 231.79 | 7.38 | 178 | 92.55 |

Table 16: MFLOPs, Inference time [ms], Memory usage [MBs], and Energy consumption [mJ] of ResNet-18 on Tiny-ImageNet across different hardware platforms.

| Hardward | Rem. | MFLOPs | Inference time [ms] | Mem.usage [MBs] | top-1 |
|---|---|---|---|---|---|
| NVIDIA A4500 | 0/17 | 7292.87 | 3.56 | 227 | 41.44 |
|  | 2/17 | 7618.77 | 3.24 | 225 | 41.42 |
| RTX 2080 | 0/17 | 7292.87 | 4.62 | 227 | 41.44 |
|  | 2/17 | 7618.77 | 4.21 | 225 | 41.42 |
| NVIDIA A4000 | 0/17 | 7292.87 | 4.73 | 228 | 41.44 |
|  | 2/17 | 7618.77 | 4.27 | 225 | 41.42 |
| Jetson Nano | 0/17 | 7292.87 | 12.06 | 228 | 41.44 |
|  | 2/17 | 7618.77 | 11.87 | 226 | 41.42 |

sparsity alone often yields limited speedups due to kernel launch overheads and memory access patterns. In contrast, reducing the number of layers directly reduces (i) kernel invocations, (ii) synchronization points, and (iii) activation storage and memory traffic. Therefore, even if the merged linear operator were dense, removing an entire layer can still lead to tangible performance gains, as empirically demonstrated.

### D.4 Trend of Layer's Entropy and Training Time

Tables 25, 26, 27, 28, 29, 30, 31, 32, 33, 34, 35, 36, 37, 38, 39, 41, 41, 42, 43, 44, 45, present the entropy trends in the six layers exhibiting the lowest entropy and the training time, for all the unstructured pruning and NEPENTHE setups. This illustrates that while unstructured pruning inherently reduces the entropy of certain layers, as detailed in Section 4.2, it lacks the capability to entirely eliminate any specific layer. In contrast, our methodology, NEPENTHE, aims to push all the encoded information from layers with low entropy to those with already high entropy. This strategy enables the removal of zero-entropy layers.

Table 17: MFLOPs, Inference time [ms], Memory usage [MBs], and Energy consumption [mJ] of ResNet-18 on PACS across different hardware platforms.

| Hardware | Rem. | MFLOPs | Inference time [ms] | Mem.usage [MBs] | top-1 |
|---|---|---|---|---|---|
| NVIDIA A4500 | 0/17 | 7292.67 | 3.58 | 248 | 94.70 |
| | 3/17 | 6736.15 | 3.04 | 181 | 90.10 |
| RTX 2080 | 0/17 | 7292.67 | 4.54 | 248 | 94.70 |
| | 3/17 | 6736.15 | 4.00 | 181 | 90.10 |
| NVIDIA A4000 | 0/17 | 7292.67 | 4.61 | 248 | 94.70 |
| | 3/17 | 6736.15 | 4.09 | 182 | 90.10 |
| Jetson Nano | 0/17 | 7292.67 | 12.00 | 248 | 94.70 |
| | 3/17 | 6736.15 | 11.04 | 181 | 90.10 |

Table 18: MFLOPs, Inference time [ms], Memory usage [MBs], and Energy consumption [mJ] of ResNet-18 on VLCS across different hardware platforms.

| Hardware | Rem. | MFLOPs | Inference time [ms] | Mem.usage [MBs] | top-1 |
|---|---|---|---|---|---|
| NVIDIA A4500 | 0/17 | 7292.67 | 3.56 | 285 | 80.89 |
| | 2/17 | 6760.27 | 3.05 | 258 | 78.38 |
| RTX 2080 | 0/17 | 7292.67 | 4.58 | 285 | 80.89 |
| | 2/17 | 6760.27 | 4.21 | 257 | 78.38 |
| NVIDIA A4000 | 0/17 | 7292.67 | 4.66 | 286 | 80.89 |
| | 2/17 | 6760.27 | 4.17 | 258 | 78.38 |
| Jetson Nano | 0/17 | 7292.67 | 12.00 | 285 | 80.89 |
| | 2/17 | 6760.27 | 7.75 | 258 | 78.38 |

Table 19: MFLOPs, Inference time [ms], Memory usage [MBs], and Energy consumption [mJ] of Swin-T on CIFAR-10 across different hardware platforms.

| Hardware | Rem. | MFLOPs | Inference time [ms] | Mem.usage [MBs] | top-1 |
|---|---|---|---|---|---|
| NVIDIA A4500 | 0/12 | 1041.34 | 7.79 | 245 | 91.54 |
| | 2/12 | 1020.11 | 7.71 | 79 | 92.29 |
| RTX 2080 | 0/12 | 1041.34 | 3.32 | 230 | 91.54 |
| | 2/12 | 1020.11 | 2.96 | 120 | 92.29 |
| RTX 4000 | 0/12 | 1041.34 | 4.52 | 241 | 91.54 |
| | 2/12 | 1020.11 | 3.94 | 212 | 92.29 |
| Jetson Nano | 0/12 | 1041.34 | 8.52 | 241 | 91.54 |
| | 2/12 | 1020.11 | 7.38 | 128 | 92.29 |

Table 20: MFLOPs, Inference time [ms], Memory usage [MBs], and Energy consumption [mJ] of Swin-T on Tiny-ImageNet across different hardware platforms.

| Hardward | Rem. | MFLOPs | Inference time [ms] | Mem.usage [MBs] | top-1 |
|---|---|---|---|---|---|
| NVIDIA A4500 | 0/12 | 7546.87 | 3.56 | 155 | 75.60 |
| | 1/12 | 7511.50 | 3.24 | 100 | 72.58 |
| RTX 2080 | 0/12 | 7546.87 | 4.62 | 245 | 75.60 |
| | 1/12 | 7511.50 | 4.21 | 80 | 72.58 |
| NVIDIA A4000 | 0/12 | 7546.87 | 4.73 | 155 | 75.60 |
| | 1/12 | 7511.50 | 4.27 | 100 | 72.58 |
| Jetson Nano | 0/12 | 7546.87 | 12.06 | 156 | 75.60 |
| | 1/12 | 7511.50 | 11.87 | 100 | 72.58 |

Table 21: MFLOPs, Inference time [ms], Memory usage [MBs], and Energy consumption [mJ] of Swin-T on PACS across different hardware platforms.

| Hardward | Rem. | MFLOPs | Inference time [ms] | Mem.usage [MBs] | top-1 |
|---|---|---|---|---|---|
| NVIDIA A4500 | 0/12 | 8989.52 | 9.47 | 212 | 97.10 |
| | 2/12 | 8177.89 | 9.27 | 115 | 95.10 |
| RTX 2080 | 0/12 | 8989.52 | 15.09 | 212 | 97.10 |
| | 2/12 | 8177.89 | 14.92 | 114 | 95.10 |
| NVIDIA A4000 | 0/12 | 8989.52 | 16.02 | 209 | 97.10 |
| | 2/12 | 8177.89 | 15.76 | 115 | 95.10 |
| Jetson Nano | 0/12 | 8989.52 | 38.71 | 212 | 97.10 |
| | 2/12 | 8177.89 | 38.93 | 115 | 95.10 |

Table 22: MFLOPs, Inference time [ms], Memory usage [MBs], and Energy consumption [mJ] of Swin-T on VLCS across different hardware platforms.

| Hardward | Rem. | MFLOPs | Inference time [ms] | Mem.usage [MBs] | top-1 |
|---|---|---|---|---|---|
| NVIDIA A4500 | 0/12 | 8989.48 | 9.48 | 194 | 80.89 |
| | 1/12 | 8582.50 | 9.34 | 118 | 78.38 |
| RTX 2080 | 0/12 | 8989.48 | 15.09 | 194 | 80.89 |
| | 1/12 | 8582.50 | 14.94 | 120 | 78.38 |
| NVIDIA A4000 | 0/12 | 8989.48 | 16.00 | 194 | 80.89 |
| | 1/12 | 8582.50 | 15.85 | 120 | 78.38 |
| Jetson Nano | 0/12 | 8989.48 | 39.61 | 195 | 80.89 |
| | 1/12 | 8582.50 | 38.93 | 118 | 78.38 |

Table 23: Test performance (top-1) and the number of removed layers (Rem.) for ResNet-50, ResNet-152, and MobileNetV2-0.75 models trained on CIFAR-10.

| Dataset | Approach | ResNet-50 | | ResNet-152 | | MobileNetV2-0.75 | |
|---|---|---|---|---|---|---|---|
| | | Top-1 | Rem. | Top-1 | Rem. | Top-1 | Rem. |
| CIFAR-10 | Dense | 87.37 | 0/49 | 85.61 | 0/151 | 85.17 | 0/35 |
| | NEPENTHE | 89.06 | 16/49 | 89.20 | 82/151 | 87.06 | 12/35 |

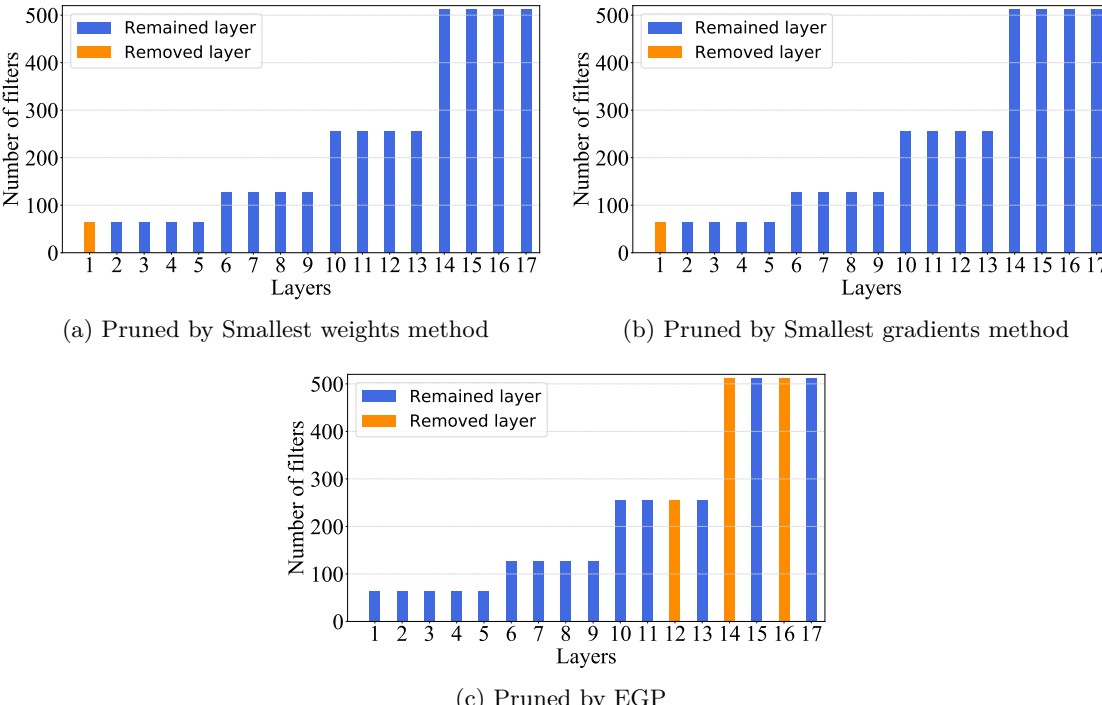

(a) Pruned by Smallest weights method  (b) Pruned by Smallest gradients method

(c) Pruned by EGP

Figure 6: Layer states for ResNet-18 trained on CIFAR-10 with different methods.

Table 24: Test performance (top-1) and the number of removed layers (Rem.) for Resnet-18 trained on Imagenet.

| Dataset | Approach | ResNet-18 | |
|---|---|---|---|
| | | top-1 | Rem. |
| | Dense model | 68.20 | 0/17 |
| | IMP (low prune) | 68.38 | 0/17 |
| | IMP (mid prune) | 67.88 | 0/17 |
| ImageNet | IMP (high prune) | 66.63 | 0/17 |
| | NEPENTHE (low prune) | 66.17 | 0/17 |
| | NEPENTHE (mid prune) | 62.74 | 1/17 |
| | NEPENTHE (high prune) | 62.15 | 3/17 |

Table 25: Test performance (top-1), bottom six layer's entropies $\widehat{\mathcal{H}}$ and the number of removed layers (Rem.) for ResNet-18 on CIFAR-10.

| Approach | $\widehat{\mathcal{H}}_1$ | $\widehat{\mathcal{H}}_2$ | $\widehat{\mathcal{H}}_3$ | $\widehat{\mathcal{H}}_4$ | $\widehat{\mathcal{H}}_5$ | $\widehat{\mathcal{H}}_6$ | top-1 | Rem. | Time |
|---|---|---|---|---|---|---|---|---|---|
| Dense model | 0.647 | 0.680 | 0.728 | 0.785 | 0.791 | 0.797 | 91.66 | 0/17 | 0h48 |
| IMP(iter #1) | 0.585 | 0.650 | 0.699 | 0.725 | 0.767 | 0.778 | 92.29 | 0/17 | 1h36 |
| IMP(iter #2) | 0.506 | 0.580 | 0.647 | 0.654 | 0.700 | 0.722 | 92.25 | 0/17 | 2h24 |
| IMP(iter #3) | 0.256 | 0.623 | 0.658 | 0.672 | 0.682 | 0.737 | 92.46 | 0/17 | 3h12 |
| IMP(iter #4) | 0.192 | 0.660 | 0.667 | 0.676 | 0.698 | 0.763 | 92.27 | 0/17 | 4h00 |
| IMP(iter #5) | 0.136 | 0.589 | 0.648 | 0.727 | 0.728 | 0.791 | 92.44 | 0/17 | 4h48 |
| IMP(iter #6) | 0.093 | 0.447 | 0.640 | 0.650 | 0.764 | 0.765 | 91.89 | 0/17 | 5h36 |
| IMP(iter #7) | 0.055 | 0.335 | 0.487 | 0.592 | 0.640 | 0.775 | 91.66 | 0/17 | 6h24 |
| NEPENTHE(iter #1) | 0 | 0.168 | 0.581 | 0.654 | 0.681 | 0.714 | 92.25 | 1/17 | 1h37 |
| NEPENTHE(iter #2) | 0 | 0.076 | 0.615 | 0.619 | 0.633 | 0.644 | 92.60 | 1/17 | 2h26 |
| NEPENTHE(iter #3) | 0 | 0 | 0 | 0.121 | 0.139 | 0.642 | 92.55 | 3/17 | 3h15 |
| NEPENTHE(iter #4) | 0 | 0 | 0 | 0.003 | 0.242 | 0.320 | 91.93 | 3/17 | 4h04 |
| NEPENTHE(iter #5) | 0 | 0 | 0 | 0 | 0 | 0.114 | 89.30 | 5/17 | 4h53 |
| NEPENTHE(iter #6) | 0 | 0 | 0 | 0 | 0 | 0.019 | 89.43 | 5/17 | 5h42 |
| NEPENTHE(iter #7) | 0 | 0 | 0 | 0 | 0 | 0 | 83.42 | 6/17 | 6h31 |

Table 26: Test performance (top-1), bottom six layer's entropies $\widehat{\mathcal{H}}$ and the number of removed layers (Rem.) for Swin-T on CIFAR-10.

| Approach | $\widehat{\mathcal{H}}_1$ | $\widehat{\mathcal{H}}_2$ | $\widehat{\mathcal{H}}_3$ | $\widehat{\mathcal{H}}_4$ | $\widehat{\mathcal{H}}_5$ | $\widehat{\mathcal{H}}_6$ | top-1 | Rem. | Time |
|---|---|---|---|---|---|---|---|---|---|
| Dense model | 0.03 | 0.054 | 0.382 | 0.394 | 0.394 | 0.44 | 91,54 | 0/12 | 1h53 |
| IMP(iter #1) | 0.03 | 0.055 | 0.383 | 0.397 | 0.398 | 0.444 | 91.73 | 0/12 | 3h46 |
| IMP(iter #2) | 0.031 | 0.057 | 0.382 | 0.392 | 0.399 | 0.443 | 91.80 | 0/12 | 5h39 |
| IMP(iter #3) | 0.034 | 0.063 | 0.375 | 0.383 | 0.393 | 0.438 | 91.43 | 0/12 | 7h32 |
| IMP(iter #4) | 0.036 | 0.072 | 0.365 | 0.379 | 0.382 | 0.426 | 91.46 | 0/12 | 9h25 |
| IMP(iter #5) | 0.041 | 0.080 | 0.349 | 0.361 | 0.369 | 0.409 | 91.27 | 0/12 | 11h18 |
| IMP(iter #6) | 0.048 | 0.096 | 0.334 | 0.343 | 0.350 | 0.386 | 91.05 | 0/12 | 13h11 |
| IMP(iter #7) | 0.055 | 0.113 | 0.31 | 0.325 | 0.327 | 0.355 | 90.53 | 0/12 | 15h04 |
| NEPENTHE(iter #1) | 0.001 | 0.215 | 0.385 | 0.397 | 0.407 | 0.443 | 91.77 | 0/12 | 3h48 |
| NEPENTHE(iter #2) | 0.001 | 0.219 | 0.387 | 0.399 | 0.409 | 0.445 | 92.06 | 0/12 | 5h45 |
| NEPENTHE(iter #3) | 0.001 | 0.254 | 0.380 | 0.395 | 0.405 | 0.440 | 92.08 | 0/12 | 7h38 |
| NEPENTHE(iter #4) | 0,001 | 0.001 | 0.377 | 0.388 | 0.404 | 0.433 | 92.31 | 0/12 | 9h33 |
| NEPENTHE(iter #5) | 0 | 0 | 0.363 | 0.373 | 0.406 | 0.423 | 92.29 | 2/12 | 11h28 |
| NEPENTHE(iter #6) | 0 | 0 | 0.344 | 0.359 | 0.407 | 0.412 | 92.21 | 2/12 | 13h23 |
| NEPENTHE(iter #7) | 0 | 0 | 0.287 | 0.317 | 0.405 | 0.405 | 92.11 | 2/12 | 15h18 |

Table 27: Test performance (top-1), bottom six layer's entropies $\widehat{\mathcal{H}}$ and the number of removed layers (Rem.) for MobileNetv2 on CIFAR-10.

| Approach | $\widehat{\mathcal{H}}_1$ | $\widehat{\mathcal{H}}_2$ | $\widehat{\mathcal{H}}_3$ | $\widehat{\mathcal{H}}_4$ | $\widehat{\mathcal{H}}_5$ | $\widehat{\mathcal{H}}_6$ | top-1 | Rem. | Time |
|---|---|---|---|---|---|---|---|---|---|
| Dense model | 0.386 | 0.474 | 0.486 | 0.504 | 0.528 | 0.544 | 93.68 | 0/35 | 0h43 |
| IMP(iter #1) | 0.186 | 0.206 | 0.233 | 0.241 | 0.249 | 0.260 | 94.07 | 0/35 | 1h26 |
| IMP(iter #2) | 0.116 | 0.117 | 0.153 | 0.167 | 0.167 | 0.168 | 93.67 | 0/35 | 2h09 |
| IMP(iter #3) | 0.082 | 0.084 | 0.111 | 0.119 | 0.119 | 0.125 | 93.83 | 0/35 | 2h52 |
| IMP(iter #4) | 0.063 | 0.066 | 0.090 | 0.094 | 0.095 | 0.101 | 93.32 | 0/35 | 3h35 |
| IMP(iter #5) | 0.056 | 0.057 | 0.074 | 0.075 | 0.086 | 0.088 | 93.26 | 0/35 | 4h18 |
| IMP(iter #6) | 0.050 | 0.050 | 0.065 | 0.067 | 0.071 | 0.076 | 93.47 | 0/35 | 5h01 |
| IMP(iter #7) | 0.046 | 0.047 | 0.059 | 0.060 | 0.061 | 0.064 | 93.50 | 0/35 | 5h44 |
| NEPENTHE(iter #1) | 0 | 0.198 | 0.229 | 0.232 | 0.244 | 0.248 | 93.55 | 1/35 | 1h27 |
| NEPENTHE(iter #2) | 0 | 0.109 | 0.127 | 0.138 | 0.139 | 0.149 | 93.42 | 1/35 | 2h11 |
| NEPENTHE(iter #3) | 0 | 0.082 | 0.085 | 0.103 | 0.106 | 0.107 | 93.14 | 1/35 | 2h55 |
| NEPENTHE(iter #4) | 0 | 0 | 0.063 | 0.065 | 0.074 | 0.076 | 93.25 | 2/35 | 3h39 |
| NEPENTHE(iter #5) | 0 | 0 | 0.064 | 0.067 | 0.049 | 0.051 | 93.37 | 4/35 | 4h23 |
| NEPENTHE(iter #6) | 0 | 0 | 0 | 0 | 0 | 0.042 | 93.15 | 5/35 | 5h07 |
| NEPENTHE(iter #7) | 0 | 0 | 0 | 0 | 0 | 0 | 93.26 | 7/35 | 5h51 |

Table 28: Test performance (top-1), bottom six layer's entropies $\widehat{\mathcal{H}}$ and the number of removed layers (Rem.) for ResNet-18 on Tiny ImageNet.

| Approach | $\widehat{\mathcal{H}}_1$ | $\widehat{\mathcal{H}}_2$ | $\widehat{\mathcal{H}}_3$ | $\widehat{\mathcal{H}}_4$ | $\widehat{\mathcal{H}}_5$ | $\widehat{\mathcal{H}}_6$ | top-1 | Rem. | Time |
|---|---|---|---|---|---|---|---|---|---|
| Dense model | 0.471 | 0.500 | 0.592 | 0.625 | 0.627 | 0.783 | 41,44 | 0/17 | 2h15 |
| IMP(iter #1) | 0.470 | 0.540 | 0.621 | 0.662 | 0.666 | 0.780 | 42.24 | 0/17 | 4h30 |
| IMP(iter #2) | 0.461 | 0.621 | 0.637 | 0.697 | 0.726 | 0.781 | 42.12 | 0/17 | 6h45 |
| IMP(iter #3) | 0.487 | 0.643 | 0.735 | 0.736 | 0.776 | 0.779 | 42.10 | 0/17 | 9h00 |
| IMP(iter #4) | 0.488 | 0.643 | 0.760 | 0.783 | 0.831 | 0.831 | 41.18 | 0/17 | 11h15 |
| IMP(iter #5) | 0.482 | 0.605 | 0.727 | 0.839 | 0.845 | 0.872 | 39.92 | 0/17 | 13h30 |
| IMP(iter #6) | 0.469 | 0.585 | 0.690 | 0.814 | 0.834 | 0.834 | 37.16 | 0/17 | 15h45 |
| IMP(iter #7) | 0.464 | 0.544 | 0.641 | 0.661 | 0.725 | 0.741 | 39.14 | 0/17 | 18h00 |
| NEPENTHE(iter #1) | 0 | 0 | 0.063 | 0.559 | 0.633 | 0.699 | 41.42 | 2/17 | 4h33 |
| NEPENTHE(iter #2) | 0 | 0 | 0 | 0 | 0 | 0.129 | 39.56 | 5/17 | 6h51 |
| NEPENTHE(iter #3) | 0 | 0 | 0 | 0 | 0 | 0.169 | 40.00 | 5/17 | 9h09 |
| NEPENTHE(iter #4) | 0 | 0 | 0 | 0 | 0 | 0.109 | 39.40 | 5/17 | 11h27 |
| NEPENTHE(iter #5) | 0 | 0 | 0 | 0 | 0 | 0.107 | 38.58 | 5/17 | 13h45 |
| NEPENTHE(iter #6) | 0 | 0 | 0 | 0 | 0 | 0.125 | 37.34 | 5/17 | 16h03 |
| NEPENTHE(iter #7) | 0 | 0 | 0 | 0 | 0 | 0.138 | 35.80 | 5/17 | 18h21 |

Table 29: Test performance (top-1), bottom six layer's entropies $\widehat{\mathcal{H}}$ and the number of removed layers (Rem.) for Swin-T on Tiny ImageNet.

| Approach | $\widehat{\mathcal{H}}_1$ | $\widehat{\mathcal{H}}_2$ | $\widehat{\mathcal{H}}_3$ | $\widehat{\mathcal{H}}_4$ | $\widehat{\mathcal{H}}_5$ | $\widehat{\mathcal{H}}_6$ | top-1 | Rem. | Time |
|---|---|---|---|---|---|---|---|---|---|
| Dense model | 0.067 | 0.133 | 0.38 | 0.388 | 0.395 | 0.411 | 75.60 | 0/12 | 5h41 |
| IMP(iter #1) | 0.069 | 0.131 | 0.370 | 0.373 | 0.384 | 0.399 | 75.86 | 0/12 | 11h22 |
| IMP(iter #2) | 0.073 | 0.133 | 0.355 | 0.356 | 0.367 | 0.380 | 75.26 | 0/12 | 17h03 |
| IMP(iter #3) | 0.080 | 0.143 | 0.335 | 0.336 | 0.346 | 0.357 | 73.60 | 0/12 | 22h44 |
| IMP(iter #4) | 0.089 | 0.156 | 0.313 | 0.314 | 0.319 | 0.330 | 72.32 | 0/12 | 28h25 |
| IMP(iter #5) | 0.096 | 0.169 | 0.291 | 0.291 | 0.295 | 0.309 | 70.90 | 0/12 | 34h06 |
| IMP(iter #6) | 0.102 | 0.184 | 0.268 | 0.269 | 0.275 | 0.294 | 69.80 | 0/12 | 39h47 |
| IMP(iter #7) | 0.104 | 0.193 | 0.249 | 0.255 | 0.266 | 0.289 | 67.56 | 0/12 | 45h28 |
| NEPENTHE(iter #1) | 0 | 0.139 | 0.370 | 0.377 | 0.392 | 0.394 | 72.58 | 1/12 | 11h27 |
| NEPENTHE(iter #2) | 0 | 0.143 | 0.150 | 0.183 | 0.195 | 0.381 | 71.02 | 1/12 | 17h13 |
| NEPENTHE(iter #3) | 0 | 0.143 | 0.158 | 0.183 | 0.192 | 0.269 | 70.76 | 1/12 | 22h59 |
| NEPENTHE(iter #4) | 0 | 0.133 | 0.137 | 0.165 | 0.178 | 0.187 | 70.12 | 1/12 | 28h45 |
| NEPENTHE(iter #5) | 0 | 0.128 | 0.132 | 0.172 | 0.173 | 0.180 | 69.68 | 1/12 | 34h36 |
| NEPENTHE(iter #6) | 0 | 0.124 | 0.129 | 0.164 | 0.174 | 0.176 | 70.06 | 1/12 | 39h17 |
| NEPENTHE(iter #7) | 0 | 0.123 | 0.128 | 0.160 | 0.170 | 0.180 | 69.42 | 1/12 | 46h03 |

Table 30: Test performance (top-1), bottom six layer's entropies $\widehat{\mathcal{H}}$ and the number of removed layers (Rem.) for MobileNetv2 on Tiny ImageNet.

| Approach | $\widehat{\mathcal{H}}_1$ | $\widehat{\mathcal{H}}_2$ | $\widehat{\mathcal{H}}_3$ | $\widehat{\mathcal{H}}_4$ | $\widehat{\mathcal{H}}_5$ | $\widehat{\mathcal{H}}_6$ | top-1 | Rem. | Time |
|---|---|---|---|---|---|---|---|---|---|
| Dense model | 0.076 | 0.112 | 0.131 | 0.133 | 0.153 | 0.154 | 45.860 | 0/35 | 1h47 |
| IMP(iter #1) | 0.036 | 0.039 | 0.039 | 0.045 | 0.048 | 0.054 | 45.84 | 0/35 | 3h34 |
| IMP(iter #2) | 0.025 | 0.026 | 0.031 | 0.035 | 0.037 | 0.037 | 47.10 | 0/35 | 5h21 |
| IMP(iter #3) | 0.018 | 0.021 | 0.030 | 0.030 | 0.031 | 0.031 | 47.740 | 0/35 | 7h08 |
| IMP(iter #4) | 0.016 | 0.017 | 0.028 | 0.028 | 0.030 | 0.030 | 46.800 | 0/35 | 8h55 |
| IMP(iter #5) | 0.013 | 0.016 | 0.025 | 0.025 | 0.028 | 0.029 | 47.560 | 0/35 | 10h42 |
| IMP(iter #6) | 0.008 | 0.011 | 0.022 | 0.023 | 0.028 | 0.028 | 47.580 | 0/35 | 12h29 |
| IMP(iter #7) | 0.007 | 0.01 | 0.022 | 0.023 | 0.028 | 0.029 | 47.440 | 0/35 | 14h16 |
| NEPENTHE(iter #1) | 0.001 | 0.004 | 0.095 | 0.096 | 0.098 | 0.110 | 46.70 | 0/35 | 3h37 |
| NEPENTHE(iter #2) | 0 | 0.003 | 0.058 | 0.064 | 0.071 | 0.073 | 47.22 | 1/35 | 5h27 |
| NEPENTHE(iter #3) | 0 | 0 | 0 | 0 | 0 | 0 | 47.26 | 6/35 | 7h17 |
| NEPENTHE(iter #4) | 0 | 0 | 0 | 0 | 0 | 0 | 47.82 | 9/35 | 9h07 |
| NEPENTHE(iter #5) | 0 | 0 | 0 | 0 | 0 | 0 | 47.92 | 12/35 | 10h57 |
| NEPENTHE(iter #6) | 0 | 0 | 0 | 0 | 0 | 0 | 0.50 | 34/35 | 12h37 |

Table 31: Test performance (top-1), bottom six layer's entropies $\widehat{\mathcal{H}}$ and the number of removed layers (Rem.) for ResNet-18 on PACS.

| Approach | $\widehat{\mathcal{H}}_1$ | $\widehat{\mathcal{H}}_2$ | $\widehat{\mathcal{H}}_3$ | $\widehat{\mathcal{H}}_4$ | $\widehat{\mathcal{H}}_5$ | $\widehat{\mathcal{H}}_6$ | top-1 | Rem. | Time |
|---|---|---|---|---|---|---|---|---|---|
| Dense model | 0.332 | 0.439 | 0.602 | 0.667 | 0.686 | 0.687 | 94.70 | 0/17 | 0h46 |
| IMP(iter #1) | 0.331 | 0.423 | 0.608 | 0.669 | 0.688 | 0.688 | 95.40 | 0/17 | 1h31 |
| IMP(iter #2) | 0.319 | 0.429 | 0.602 | 0.668 | 0.670 | 0.683 | 95.30 | 0/17 | 2h17 |
| IMP(iter #3) | 0.324 | 0.419 | 0.607 | 0.631 | 0.682 | 0.682 | 94.60 | 0/17 | 3h03 |
| IMP(iter #4) | 0.318 | 0.441 | 0.613 | 0.613 | 0.661 | 0.688 | 95.10 | 0/17 | 3h49 |
| IMP(iter #5) | 0.300 | 0.452 | 0.587 | 0.621 | 0.636 | 0.694 | 94.00 | 0/17 | 4h35 |
| IMP(iter #6) | 0.285 | 0.458 | 0.533 | 0.643 | 0.647 | 0.694 | 92.30 | 0/17 | 5h21 |
| IMP(iter #7) | 0.280 | 0.418 | 0.479 | 0.584 | 0.646 | 0.657 | 90.80 | 0/17 | 6h07 |
| NEPENTHE(iter #1) | 0.129 | 0.430 | 0.482 | 0.634 | 0.668 | 0.669 | 94.20 | 0/17 | 1h32 |
| NEPENTHE(iter #2) | 0 | 0.041 | 0.091 | 0.482 | 0.596 | 0.596 | 92.40 | 1/17 | 2h19 |
| NEPENTHE(iter #3) | 0 | 0.030 | 0.066 | 0.527 | 0.559 | 0.599 | 93.00 | 1/17 | 3h06 |
| NEPENTHE(iter #4) | 0 | 0 | 0.033 | 0.067 | 0.422 | 0.565 | 90.40 | 2/17 | 3h53 |
| NEPENTHE(iter #5) | 0 | 0 | 0.032 | 0.061 | 0.084 | 0.217 | 89.50 | 2/17 | 4h40 |
| NEPENTHE(iter #6) | 0 | 0 | 0 | 0.001 | 0.002 | 0.028 | 90.10 | 3/17 | 5h27 |
| NEPENTHE(iter #7) | 0 | 0 | 0 | 0.002 | 0.002 | 0.040 | 86.30 | 3/17 | 6h14 |

Table 32: Test performance (top-1), bottom six layer's entropies $\widehat{\mathcal{H}}$ and the number of removed layers (Rem.) for Swin-T on PACS.

| Approach | $\widehat{\mathcal{H}}_1$ | $\widehat{\mathcal{H}}_2$ | $\widehat{\mathcal{H}}_3$ | $\widehat{\mathcal{H}}_4$ | $\widehat{\mathcal{H}}_5$ | $\widehat{\mathcal{H}}_6$ | top-1 | Rem. | Time |
|---|---|---|---|---|---|---|---|---|---|
| Dense model | 0.057 | 0.184 | 0.344 | 0.362 | 0.380 | 0.381 | 97.10 | 0/12 | 0h57 |
| IMP(iter #1) | 0.060 | 0.204 | 0.349 | 0.362 | 0.382 | 0.389 | 96.60 | 0/12 | 1h54 |
| IMP(iter #2) | 0.067 | 0.214 | 0.357 | 0.370 | 0.383 | 0.388 | 96.20 | 0/12 | 2h51 |
| IMP(iter #3) | 0.081 | 0.245 | 0.362 | 0.366 | 0.375 | 0.378 | 96.00 | 0/12 | 3h48 |
| IMP(iter #4) | 0.095 | 0.269 | 0.352 | 0.353 | 0.355 | 0.372 | 96.60 | 0/12 | 4h45 |
| IMP(iter #5) | 0.110 | 0.303 | 0.314 | 0.335 | 0.339 | 0.341 | 95.00 | 0/12 | 5h42 |
| IMP(iter #6) | 0.113 | 0.278 | 0.306 | 0.318 | 0.321 | 0.329 | 94.60 | 0/12 | 6h39 |
| IMP(iter #7) | 0.101 | 0.232 | 0.269 | 0.284 | 0.293 | 0.298 | 93.90 | 0/12 | 7h36 |
| NEPENTHE(iter #1) | 0.086 | 0.240 | 0.344 | 0.376 | 0.376 | 0.403 | 96.90 | 0/12 | 1h55 |
| NEPENTHE(iter #2) | 0.001 | 0.369 | 0.372 | 0.383 | 0.398 | 0.416 | 96.50 | 0/12 | 2h53 |
| NEPENTHE(iter #3) | 0.001 | 0.368 | 0.383 | 0.385 | 0.390 | 0.406 | 95.90 | 0/12 | 3h51 |
| NEPENTHE(iter #4) | 0.001 | 0.360 | 0.369 | 0.369 | 0.392 | 0.394 | 96.30 | 0/12 | 4h49 |
| NEPENTHE(iter #5) | 0 | 0 | 0.001 | 0.335 | 0.359 | 0.366 | 95.10 | 2/12 | 5h47 |
| NEPENTHE(iter #6) | 0 | 0 | 0.107 | 0.298 | 0.348 | 0.349 | 94.60 | 2/12 | 6h45 |
| NEPENTHE(iter #7) | 0 | 0 | 0.001 | 0.001 | 0.161 | 0.232 | 93.30 | 2/12 | 7h43 |

Table 33: Test performance (top-1), bottom six layer's entropies $\widehat{\mathcal{H}}$ and the number of removed layers (Rem.) for MobileNetv2 on PACS.

| Approach | $\widehat{\mathcal{H}}_1$ | $\widehat{\mathcal{H}}_2$ | $\widehat{\mathcal{H}}_3$ | $\widehat{\mathcal{H}}_4$ | $\widehat{\mathcal{H}}_5$ | $\widehat{\mathcal{H}}_6$ | top-1 | Rem. | Time |
|---|---|---|---|---|---|---|---|---|---|
| Dense model | 0.207 | 0.291 | 0.324 | 0.372 | 0.463 | 0.471 | 93.20 | 0/35 | 0h34 |
| IMP(iter #1) | 0.218 | 0.263 | 0.284 | 0.390 | 0.457 | 0.467 | 95.70 | 0/35 | 1h09 |
| IMP(iter #2) | 0.216 | 0.235 | 0.257 | 0.373 | 0.453 | 0.462 | 95.50 | 0/35 | 1h44 |
| IMP(iter #3) | 0.224 | 0.244 | 0.252 | 0.403 | 0.464 | 0.478 | 95.40 | 0/35 | 2h19 |
| IMP(iter #4) | 0.222 | 0.229 | 0.241 | 0.386 | 0.459 | 0.476 | 95.70 | 0/35 | 2h54 |
| IMP(iter #5) | 0.212 | 0.223 | 0.233 | 0.397 | 0.464 | 0.47 | 95.60 | 0/35 | 3h29 |
| IMP(iter #6) | 0.196 | 0.212 | 0.237 | 0.405 | 0.470 | 0.485 | 96.20 | 0/35 | 4h04 |
| IMP(iter #7) | 0.170 | 0.207 | 0.234 | 0.412 | 0.468 | 0.472 | 95.40 | 0/35 | 4h39 |
| NEPENTHE(iter #1) | 0.119 | 0.139 | 0.151 | 0.192 | 0.200 | 0.225 | 93.30 | 0/35 | 1h10 |
| NEPENTHE(iter #2) | 0.093 | 0.128 | 0.129 | 0.130 | 0.135 | 0.165 | 93.20 | 0/35 | 1h46 |
| NEPENTHE(iter #3) | 0.077 | 0.093 | 0.112 | 0.125 | 0.140 | 0.141 | 92.50 | 0/35 | 2h22 |
| NEPENTHE(iter #4) | 0 | 0.076 | 0.083 | 0.105 | 0.106 | 0.116 | 92.20 | 1/35 | 2h58 |
| NEPENTHE(iter #5) | 0 | 0.054 | 0.068 | 0.096 | 0.097 | 0.115 | 89.70 | 1/35 | 3h34 |
| NEPENTHE(iter #6) | 0 | 0.014 | 0.016 | 0.036 | 0.050 | 0.051 | 89.00 | 1/35 | 4h10 |
| NEPENTHE(iter #7) | 0 | 0.004 | 0.008 | 0.023 | 0.027 | 0.034 | 88.70 | 1/35 | 4h46 |

Table 34: Test performance (top-1), bottom six layer's entropies $\widehat{\mathcal{H}}$ and the number of removed layers (Rem.) for ResNet-18 on VLCS.

| Approach | $\widehat{\mathcal{H}}_1$ | $\widehat{\mathcal{H}}_2$ | $\widehat{\mathcal{H}}_3$ | $\widehat{\mathcal{H}}_4$ | $\widehat{\mathcal{H}}_5$ | $\widehat{\mathcal{H}}_6$ | top-1 | Rem. | Time |
|---|---|---|---|---|---|---|---|---|---|
| Dense model | 0.382 | 0.457 | 0.647 | 0.676 | 0.681 | 0.698 | 80.89 | 0/17 | 3h49 |
| IMP(iter #1) | 0.387 | 0.471 | 0.647 | 0.679 | 0.681 | 0.703 | 82.76 | 0/17 | 8h37 |
| IMP(iter #2) | 0.392 | 0.476 | 0.644 | 0.654 | 0.69 | 0.703 | 82.01 | 0/17 | 13h06 |
| IMP(iter #3) | 0.378 | 0.474 | 0.620 | 0.658 | 0.707 | 0.707 | 82.01 | 0/17 | 17h15 |
| IMP(iter #4) | 0.391 | 0.491 | 0.595 | 0.672 | 0.711 | 0.726 | 80.15 | 0/17 | 22h24 |
| IMP(iter #5) | 0.372 | 0.479 | 0.571 | 0.665 | 0.716 | 0.739 | 79.31 | 0/17 | 27h13 |
| IMP(iter #6) | 0.383 | 0.519 | 0.531 | 0.699 | 0.721 | 0.750 | 78.84 | 0/17 | 31h22 |
| IMP(iter #7) | 0.357 | 0.409 | 0.502 | 0.64 | 0.707 | 0.712 | 74.09 | 0/17 | 35h11 |
| NEPENTHE(iter #1) | 0,001 | 0.453 | 0.497 | 0.651 | 0.676 | 0.680 | 78.99 | 0/17 | 8h44 |
| NEPENTHE(iter #2) | 0 | 0 | 0,001 | 0.508 | 0.516 | 0.619 | 78.38 | 2/17 | 13h20 |
| NEPENTHE(iter #3) | 0 | 0 | 0 | 0 | 0.518 | 0.553 | 76.98 | 4/17 | 17h36 |
| NEPENTHE(iter #4) | 0 | 0 | 0 | 0 | 0.516 | 0.574 | 78.66 | 4/17 | 22h52 |
| NEPENTHE(iter #5) | 0 | 0 | 0 | 0 | 0 | 0 | 76.05 | 6/17 | 27h48 |
| NEPENTHE(iter #6) | 0 | 0 | 0 | 0 | 0 | 0 | 74.28 | 6/17 | 32h04 |
| NEPENTHE(iter #7) | 0 | 0 | 0 | 0 | 0 | 0 | 74.37 | 6/17 | 36h01 |

Table 35: Test performance (top-1), bottom six layer's entropies $\widehat{\mathcal{H}}$ and the number of removed layers (Rem.) for Swin-T on VLCS.

| Approach | $\widehat{\mathcal{H}}_1$ | $\widehat{\mathcal{H}}_2$ | $\widehat{\mathcal{H}}_3$ | $\widehat{\mathcal{H}}_4$ | $\widehat{\mathcal{H}}_5$ | $\widehat{\mathcal{H}}_6$ | top-1 | Rem. | Time |
|---|---|---|---|---|---|---|---|---|---|
| Dense model | 0.070 | 0.175 | 0.373 | 0.385 | 0.414 | 0.427 | 86.58 | 0/12 | 2h22 |
| IMP(iter #1) | 0.078 | 0.179 | 0.391 | 0.402 | 0.421 | 0.433 | 84.72 | 0/12 | 4h45 |
| IMP(iter #2) | 0.093 | 0.195 | 0.387 | 0.403 | 0.416 | 0.419 | 85.65 | 0/12 | 8h08 |
| IMP(iter #3) | 0.102 | 0.224 | 0.388 | 0.411 | 0.424 | 0.424 | 84.34 | 0/12 | 11h31 |
| IMP(iter #4) | 0.122 | 0.236 | 0.395 | 0.402 | 0.415 | 0.418 | 84.06 | 0/12 | 14h54 |
| IMP(iter #5) | 0.140 | 0.261 | 0.369 | 0.394 | 0.404 | 0.412 | 82.01 | 0/12 | 18h56 |
| IMP(iter #6) | 0.141 | 0.292 | 0.331 | 0.387 | 0.390 | 0.393 | 81.36 | 0/12 | 22h38 |
| IMP(iter #7) | 0.139 | 0.277 | 0.304 | 0.370 | 0.373 | 0.374 | 80.06 | 0/12 | 26h20 |
| NEPENTHE(iter #1) | 0.121 | 0.187 | 0.400 | 0.402 | 0.430 | 0.437 | 85.46 | 0/12 | 4h57 |
| NEPENTHE(iter #2) | 0.132 | 0.225 | 0.403 | 0.406 | 0.428 | 0.438 | 85.09 | 0/12 | 8h26 |
| NEPENTHE(iter #3) | 0 | 0.411 | 0.413 | 0.432 | 0.437 | 0.457 | 85.27 | 1/12 | 11h55 |
| NEPENTHE(iter #4) | 0 | 0.409 | 0.420 | 0.441 | 0.446 | 0.463 | 83.88 | 1/12 | 15h18 |
| NEPENTHE(iter #5) | 0 | 0.406 | 0.409 | 0.428 | 0.434 | 0.469 | 81.73 | 1/12 | 19h28 |
| NEPENTHE(iter #6) | 0 | 0.318 | 0.383 | 0.398 | 0.413 | 0.469 | 81.55 | 1/12 | 23h20 |
| NEPENTHE(iter #7) | 0 | 0.001 | 0.304 | 0.369 | 0.374 | 0.475 | 79.22 | 1/12 | 27h08 |

Table 36: Test performance (top-1), bottom six layer's entropies $\widehat{\mathcal{H}}$ and the number of removed layers (Rem.) for MobileNetv2 on VLCS.

| Approach | $\widehat{\mathcal{H}}_1$ | $\widehat{\mathcal{H}}_2$ | $\widehat{\mathcal{H}}_3$ | $\widehat{\mathcal{H}}_4$ | $\widehat{\mathcal{H}}_5$ | $\widehat{\mathcal{H}}_6$ | top-1 | Rem. | Time |
|---|---|---|---|---|---|---|---|---|---|
| Dense model | 0.257 | 0.353 | 0.46 | 0.513 | 0.514 | 0.572 | 81.83 | 0/35 | 3h00 |
| IMP(iter #1) | 0.270 | 0.292 | 0.490 | 0.503 | 0.533 | 0.573 | 80.80 | 0/35 | 6h01 |
| IMP(iter #2) | 0.261 | 0.263 | 0.507 | 0.524 | 0.528 | 0.562 | 81.64 | 0/35 | 9h01 |
| IMP(iter #3) | 0.269 | 0.271 | 0.496 | 0.501 | 0.536 | 0.573 | 80.43 | 0/35 | 12h02 |
| IMP(iter #4) | 0.258 | 0.258 | 0.491 | 0.521 | 0.550 | 0.579 | 79.59 | 0/35 | 15h03 |
| IMP(iter #5) | 0.264 | 0.271 | 0.474 | 0.540 | 0.545 | 0.589 | 79.96 | 0/35 | 18h03 |
| IMP(iter #6) | 0.268 | 0.277 | 0.468 | 0.547 | 0.549 | 0.585 | 80.80 | 0/35 | 21h04 |
| IMP(iter #7) | 0.273 | 0.279 | 0.470 | 0.524 | 0.554 | 0.592 | 80.43 | 0/35 | 24h05 |
| NEPENTHE(iter #1) | 0.184 | 0.249 | 0.342 | 0.505 | 0.534 | 0.581 | 81.08 | 0/35 | 6h04 |
| NEPENTHE(iter #2) | 0.001 | 0.077 | 0.251 | 0.345 | 0.417 | 0.500 | 80.52 | 0/35 | 9h07 |
| NEPENTHE(iter #3) | 0 | 0.002 | 0.005 | 0.260 | 0.354 | 0.488 | 78.84 | 1/35 | 12h11 |
| NEPENTHE(iter #4) | 0 | 0.001 | 0.040 | 0.261 | 0.363 | 0.527 | 77.91 | 1/35 | 15h15 |
| NEPENTHE(iter #5) | 0 | 0 | 0.001 | 0.274 | 0.366 | 0.523 | 80.06 | 2/35 | 18h18 |
| NEPENTHE(iter #6) | 0 | 0 | 0.001 | 0.26 | 0.351 | 0.485 | 79.31 | 2/35 | 21h22 |

Table 37: Test performance (top-1), bottom six layer's entropies $\widehat{\mathcal{H}}$ and the number of removed layers (Rem.) for ResNet-18 on SVIRO.

| Approach | $\widehat{\mathcal{H}}_1$ | $\widehat{\mathcal{H}}_2$ | $\widehat{\mathcal{H}}_3$ | $\widehat{\mathcal{H}}_4$ | $\widehat{\mathcal{H}}_5$ | $\widehat{\mathcal{H}}_6$ | top-1 | Rem. | Time |
|---|---|---|---|---|---|---|---|---|---|
| Dense model | 0.009 | 0.446 | 0.631 | 0.685 | 0.693 | 0.707 | 99,93 | 0/17 | 7h03 |
| IMP(iter #1) | 0.009 | 0.456 | 0.629 | 0.679 | 0.684 | 0.705 | 99.98 | 0/17 | 14h06 |
| IMP(iter #2) | 0.008 | 0.446 | 0.624 | 0.646 | 0.68 | 0.700 | 99.98 | 0/17 | 21h09 |
| IMP(iter #3) | 0.009 | 0.454 | 0.638 | 0.655 | 0.676 | 0.689 | 100 | 0/17 | 28h12 |
| IMP(iter #4) | 0.007 | 0.478 | 0.641 | 0.658 | 0.677 | 0.687 | 99.95 | 0/17 | 35h15 |
| IMP(iter #5) | 0.010 | 0.507 | 0.658 | 0.659 | 0.698 | 0.702 | 99.96 | 0/17 | 42h18 |
| IMP(iter #6) | 0.006 | 0.530 | 0.577 | 0.676 | 0.688 | 0.691 | 100 | 0/17 | 49h21 |
| IMP(iter #7) | 0.003 | 0.488 | 0.495 | 0.518 | 0.629 | 0.657 | 99.95 | 0/17 | 56h26 |
| NEPENTHE(iter #1) | 0,001 | 0.512 | 0.551 | 0.649 | 0.686 | 0.702 | 99.98 | 0/17 | 14h19 |
| NEPENTHE(iter #2) | 0 | 0 | 0,001 | 0.031 | 0.453 | 0.460 | 99.93 | 2/17 | 21h35 |
| NEPENTHE(iter #3) | 0 | 0 | 0.012 | 0.397 | 0.440 | 0.598 | 99.91 | 2/17 | 28h51 |
| NEPENTHE(iter #4) | 0 | 0 | 0 | 0.006 | 0.013 | 0.371 | 99.86 | 3/17 | 36h07 |
| NEPENTHE(iter #5) | 0 | 0 | 0 | 0.001 | 0.005 | 0.054 | 99.84 | 3/17 | 43h23 |
| NEPENTHE(iter #6) | 0 | 0 | 0 | 0 | 0 | 0 | 99.61 | 8/17 | 50h44 |
| NEPENTHE(iter #7) | 0 | 0 | 0 | 0 | 0 | 0 | 98.75 | 8/17 | 57h57 |

Table 38: Test performance (top-1), bottom six layer's entropies $\widehat{\mathcal{H}}$ and the number of removed layers (Rem.) for Swin-T on SVIRO.

| Approach | $\widehat{\mathcal{H}}_1$ | $\widehat{\mathcal{H}}_2$ | $\widehat{\mathcal{H}}_3$ | $\widehat{\mathcal{H}}_4$ | $\widehat{\mathcal{H}}_5$ | $\widehat{\mathcal{H}}_6$ | top-1 | Rem. | Time |
|---|---|---|---|---|---|---|---|---|---|
| Dense model | 0.061 | 0.205 | 0.280 | 0.290 | 0.321 | 0.325 | 99,95 | 0/12 | 5h38 |
| IMP(iter #1) | 0.046 | 0.232 | 0.284 | 0.285 | 0.291 | 0.303 | 99.84 | 0/12 | 12h17 |
| IMP(iter #2) | 0.026 | 0.125 | 0.273 | 0.280 | 0.283 | 0.289 | 99.77 | 0/12 | 18h56 |
| IMP(iter #3) | 0.022 | 0.062 | 0.216 | 0.233 | 0.238 | 0.275 | 99.84 | 0/12 | 25h55 |
| IMP(iter #4) | 0.027 | 0.071 | 0.163 | 0.183 | 0.187 | 0.187 | 99.68 | 0/12 | 32h14 |
| IMP(iter #5) | 0.034 | 0.095 | 0.101 | 0.115 | 0.143 | 0.149 | 99.68 | 0/12 | 39h33 |
| IMP(iter #6) | 0.036 | 0.047 | 0.090 | 0.125 | 0.127 | 0.129 | 99.79 | 0/12 | 46h12 |
| IMP(iter #7) | 0.026 | 0.041 | 0.074 | 0.124 | 0.127 | 0.137 | 99.75 | 0/12 | 52h01 |
| NEPENTHE(iter #1) | 0.001 | 0.269 | 0.321 | 0.326 | 0.343 | 0.348 | 99.93 | 0/12 | 12h28 |
| NEPENTHE(iter #2) | 0 | 0.338 | 0.347 | 0.357 | 0.362 | 0.367 | 99.82 | 1/12 | 19h18 |
| NEPENTHE(iter #3) | 0 | 0.156 | 0.282 | 0.309 | 0.376 | 0.381 | 99.79 | 1/12 | 26h28 |
| NEPENTHE(iter #4) | 0 | 0.001 | 0.092 | 0.235 | 0.267 | 0.356 | 99.68 | 1/12 | 32h58 |
| NEPENTHE(iter #5) | 0 | 0 | 0.001 | 0.001 | 0.001 | 0.339 | 99.77 | 2/12 | 40h28 |
| NEPENTHE(iter #6) | 0 | 0 | 0 | 0 | 0 | 0.162 | 99.75 | 5/12 | 47h18 |
| NEPENTHE(iter #7) | 0 | 0 | 0 | 0 | 0 | 0.001 | 99.70 | 5/12 | 53h18 |

Table 39: Test performance (top-1), bottom six layer's entropies $\widehat{\mathcal{H}}$ and the number of removed layers (Rem.) for MobileNetv2 on SVIRO.

| Approach | $\widehat{\mathcal{H}}_1$ | $\widehat{\mathcal{H}}_2$ | $\widehat{\mathcal{H}}_3$ | $\widehat{\mathcal{H}}_4$ | $\widehat{\mathcal{H}}_5$ | $\widehat{\mathcal{H}}_6$ | top-1 | Rem. | Time |
|---|---|---|---|---|---|---|---|---|---|
| Dense model | 0.187 | 0.241 | 0.337 | 0.341 | 0.484 | 0.508 | 99.98 | 0/35 | 5h35 |
| IMP(iter #1) | 0.165 | 0.218 | 0.248 | 0.366 | 0.478 | 0.515 | 99.98 | 0/35 | 12h11 |
| IMP(iter #2) | 0.152 | 0.191 | 0.232 | 0.402 | 0.471 | 0.560 | 100 | 0/35 | 18h47 |
| IMP(iter #3) | 0.161 | 0.180 | 0.290 | 0.411 | 0.483 | 0.521 | 99.98 | 0/35 | 24h23 |
| IMP(iter #4) | 0.169 | 0.215 | 0.296 | 0.419 | 0.430 | 0.483 | 99.96 | 0/35 | 30h19 |
| IMP(iter #5) | 0.162 | 0.173 | 0.306 | 0.372 | 0.417 | 0.435 | 99.93 | 0/35 | 36h35 |
| IMP(iter #6) | 0.155 | 0.196 | 0.333 | 0.337 | 0.362 | 0.417 | 99.93 | 0/35 | 42h31 |
| IMP(iter #7) | 0.146 | 0.185 | 0.291 | 0.317 | 0.322 | 0.381 | 99.95 | 0/35 | 48h47 |
| NEPENTHE(iter #1) | 0.001 | 0.168 | 0.220 | 0.385 | 0.459 | 0.508 | 100 | 0/35 | 12h22 |
| NEPENTHE(iter #2) | 0.001 | 0.001 | 0.163 | 0.196 | 0.321 | 0.367 | 99.98 | 0/35 | 19h09 |
| NEPENTHE(iter #3) | 0 | 0 | 0.002 | 0.059 | 0.218 | 0.375 | 99.98 | 2/35 | 24h56 |
| NEPENTHE(iter #4) | 0 | 0 | 0.004 | 0.020 | 0.268 | 0.392 | 99.95 | 2/35 | 30h54 |
| NEPENTHE(iter #5) | 0 | 0 | 0.001 | 0.045 | 0.262 | 0.388 | 99.97 | 2/35 | 37h19 |
| NEPENTHE(iter #6) | 0 | 0 | 0 | 0 | 0.020 | 0.147 | 35.55 | 4/35 | 49h53 |

Table 40: Test performance (top-1), bottom six layer's entropies $\widehat{\mathcal{H}}$ and the number of removed layers (Rem.) for BERT on QNLI.

| Approach | $\widehat{\mathcal{H}}_1$ | $\widehat{\mathcal{H}}_2$ | $\widehat{\mathcal{H}}_3$ | $\widehat{\mathcal{H}}_4$ | $\widehat{\mathcal{H}}_5$ | $\widehat{\mathcal{H}}_6$ | top-1 | Rem. | Time |
|---|---|---|---|---|---|---|---|---|---|
| Dense model | 0.173 | 0.191 | 0.212 | 0.219 | 0.223 | 0.227 | 90.48 | 0/12 | 0h34 |
| IMP(iter #1) | 0.179 | 0.198 | 0.224 | 0.229 | 0.238 | 0.248 | 89.60 | 0/12 | 1h11 |
| IMP(iter #2) | 0.202 | 0.209 | 0.213 | 0.220 | 0.224 | 0.258 | 89.91 | 0/12 | 1h47 |
| IMP(iter #3) | 0.215 | 0.217 | 0.234 | 0.243 | 0.250 | 0.269 | 89.80 | 0/12 | 2h23 |
| IMP(iter #4) | 0.231 | 0.235 | 0.251 | 0.252 | 0.269 | 0.300 | 89.75 | 0/12 | 2h59 |
| IMP(iter #5) | 0.238 | 0.241 | 0.248 | 0.256 | 0.284 | 0.306 | 89.38 | 0/12 | 3h35 |
| IMP(iter #6) | 0.265 | 0.268 | 0.269 | 0.280 | 0.294 | 0.335 | 89.15 | 0/12 | 4h11 |
| IMP(iter #7) | 0.273 | 0.274 | 0.280 | 0.283 | 0.291 | 0.356 | 88.41 | 0/12 | 4h47 |
| NEPENTHE(iter #1) | 0 | 0 | 0.218 | 0.234 | 0.240 | 0.244 | 89.58 | 2/12 | 1h23 |
| NEPENTHE(iter #2) | 0 | 0 | 0.215 | 0.246 | 0.251 | 0.264 | 89.46 | 2/12 | 2h11 |
| NEPENTHE(iter #3) | 0 | 0 | 0 | 0.227 | 0.239 | 0.246 | 88.89 | 3/12 | 2h59 |
| NEPENTHE(iter #4) | 0 | 0 | 0 | 0.239 | 0.248 | 0.252 | 88.94 | 3/12 | 3h47 |
| NEPENTHE(iter #5) | 0 | 0 | 0 | 0.192 | 0.263 | 0.279 | 89.38 | 3/12 | 4h35 |
| NEPENTHE(iter #6) | 0 | 0 | 0 | 0 | 0.280 | 0.299 | 88.21 | 3/12 | 5h23 |
| NEPENTHE(iter #7) | 0 | 0 | 0 | 0 | 0.251 | 0.314 | 88.69 | 4/12 | 6h11 |

Table 41: Test performance (top-1), bottom six layer's entropies $\widehat{\mathcal{H}}$ and the number of removed layers (Rem.) for RoBERTa on QNLI.

| Approach | $\widehat{\mathcal{H}}_1$ | $\widehat{\mathcal{H}}_2$ | $\widehat{\mathcal{H}}_3$ | $\widehat{\mathcal{H}}_4$ | $\widehat{\mathcal{H}}_5$ | $\widehat{\mathcal{H}}_6$ | top-1 | Rem. | Time |
|---|---|---|---|---|---|---|---|---|---|
| Dense model | 0.190 | 0.201 | 0.210 | 0.235 | 0.275 | 0.303 | 92.18 | 0/12 | 0h35 |
| IMP(iter #1) | 0.198 | 0.214 | 0.215 | 0.260 | 0.271 | 0.276 | 92.07 | 0/12 | 1h10 |
| IMP(iter #2) | 0.205 | 0.212 | 0.255 | 0.263 | 0.300 | 0.325 | 92.07 | 0/12 | 1h45 |
| IMP(iter #3) | 0.216 | 0.223 | 0.265 | 0.266 | 0.325 | 0.334 | 91.56 | 0/12 | 2h21 |
| IMP(iter #4) | 0.227 | 0.234 | 0.265 | 0.278 | 0.343 | 0.383 | 91.65 | 0/12 | 2h56 |
| IMP(iter #5) | 0.227 | 0.255 | 0.268 | 0.321 | 0.379 | 0.380 | 91.51 | 0/12 | 3h31 |
| IMP(iter #6) | 0.239 | 0.259 | 0.261 | 0.313 | 0.368 | 0.386 | 90.19 | 0/12 | 4h06 |
| IMP(iter #7) | 0.248 | 0.254 | 0.300 | 0.346 | 0.393 | 0.420 | 90.08 | 0/12 | 4h41 |
| NEPENTHE(iter #1) | 0.001 | 0.001 | 0.230 | 0.274 | 0.426 | 0.475 | 88.36 | 0/12 | 1h22 |
| NEPENTHE(iter #2) | 0 | 0 | 0.001 | 0.028 | 0.031 | 0.487 | 87.41 | 2/12 | 2h09 |
| NEPENTHE(iter #3) | 0 | 0 | 0.001 | 0.001 | 0.003 | 0.322 | 86.43 | 2/12 | 2h56 |
| NEPENTHE(iter #4) | 0 | 0 | 0 | 0.001 | 0.001 | 0.023 | 87.26 | 3/12 | 3h43 |
| NEPENTHE(iter #5) | 0 | 0 | 0 | 0.000 | 0.001 | 0.017 | 86.03 | 3/12 | 4h30 |
| NEPENTHE(iter #6) | 0 | 0 | 0 | 0.001 | 0.001 | 0.006 | 85.21 | 3/12 | 5h17 |
| NEPENTHE(iter #7) | 0 | 0 | 0 | 0 | 0 | 0 | 84.37 | 6/12 | 6h04 |

Table 42: Test performance (top-1), bottom six layer's entropies $\widehat{\mathcal{H}}$ and the number of removed layers (Rem.) for BERT on RTE.

| Approach | $\widehat{\mathcal{H}}_1$ | $\widehat{\mathcal{H}}_2$ | $\widehat{\mathcal{H}}_3$ | $\widehat{\mathcal{H}}_4$ | $\widehat{\mathcal{H}}_5$ | $\widehat{\mathcal{H}}_6$ | top-1 | Rem. | Time |
|---|---|---|---|---|---|---|---|---|---|
| Dense model | 0.211 | 0.216 | 0.233 | 0.242 | 0.252 | 0.261 | 61.01 | 0/12 | 0h02 |
| IMP(iter #1) | 0.224 | 0.225 | 0.237 | 0.250 | 0.260 | 0.276 | 55.60 | 0/12 | 0h03 |
| IMP(iter #2) | 0.243 | 0.244 | 0.245 | 0.274 | 0.288 | 0.294 | 62.09 | 0/12 | 0h04 |
| IMP(iter #3) | 0.258 | 0.263 | 0.279 | 0.286 | 0.309 | 0.344 | 59.21 | 0/12 | 0h05 |
| IMP(iter #4) | 0.275 | 0.282 | 0.321 | 0.326 | 0.342 | 0.386 | 58.84 | 0/12 | 0h06 |
| IMP(iter #5) | 0.287 | 0.293 | 0.328 | 0.366 | 0.369 | 0.397 | 58.48 | 0/12 | 0h07 |
| IMP(iter #6) | 0.303 | 0.313 | 0.342 | 0.388 | 0.403 | 0.409 | 59.21 | 0/12 | 0h08 |
| IMP(iter #7) | 0.335 | 0.343 | 0.370 | 0.374 | 0.432 | 0.451 | 57.76 | 0/12 | 0h09 |
| NEPENTHE(iter #1) | 0.204 | 0.223 | 0.230 | 0.233 | 0.244 | 0.260 | 57.04 | 0/12 | 0h03 |
| NEPENTHE(iter #2) | 0.001 | 0.153 | 0.173 | 0.181 | 0.264 | 0.280 | 52.71 | 0/12 | 0h04 |
| NEPENTHE(iter #3) | 0 | 0.001 | 0.201 | 0.283 | 0.298 | 0.311 | 54.87 | 1/12 | 0h05 |
| NEPENTHE(iter #4) | 0 | 0 | 0 | 0.246 | 0.258 | 0.266 | 53.43 | 3/12 | 0h06 |
| NEPENTHE(iter #5) | 0 | 0 | 0 | 0.007 | 0.287 | 0.296 | 56.32 | 3/12 | 0h07 |
| NEPENTHE(iter #6) | 0 | 0 | 0 | 0.022 | 0.296 | 0.309 | 54.87 | 3/12 | 0h08 |
| NEPENTHE(iter #7) | 0 | 0 | 0 | 0.008 | 0.327 | 0.339 | 53.79 | 3/12 | 0h09 |

Table 43: Test performance (top-1), bottom six layer's entropies $\widehat{\mathcal{H}}$ and the number of removed layers (Rem.) for RoBERTa on RTE.

| Approach | $\widehat{\mathcal{H}}_1$ | $\widehat{\mathcal{H}}_2$ | $\widehat{\mathcal{H}}_3$ | $\widehat{\mathcal{H}}_4$ | $\widehat{\mathcal{H}}_5$ | $\widehat{\mathcal{H}}_6$ | top-1 | Rem. | Time |
|---|---|---|---|---|---|---|---|---|---|
| Dense model | 0.236 | 0.242 | 0.263 | 0.292 | 0.302 | 0.303 | 66,787 | 0/12 | 0h02 |
| IMP(iter #1) | 0.241 | 0.243 | 0.282 | 0.296 | 0.314 | 0.321 | 72.56 | 0/12 | 0h03 |
| IMP(iter #2) | 0.253 | 0.256 | 0.296 | 0.341 | 0.352 | 0.385 | 72.92 | 0/12 | 0h04 |
| IMP(iter #3) | 0.249 | 0.293 | 0.304 | 0.376 | 0.401 | 0.409 | 66.06 | 0/12 | 0h05 |
| IMP(iter #4) | 0.254 | 0.291 | 0.326 | 0.387 | 0.434 | 0.435 | 67.51 | 0/12 | 0h06 |
| IMP(iter #5) | 0.262 | 0.291 | 0.363 | 0.418 | 0.459 | 0.484 | 63.54 | 0/12 | 0h07 |
| IMP(iter #6) | 0.285 | 0.296 | 0.385 | 0.426 | 0.465 | 0.497 | 64.98 | 0/12 | 0h08 |
| IMP(iter #7) | 0.297 | 0.305 | 0.405 | 0.445 | 0.492 | 0.502 | 60.29 | 0/12 | 0h09 |
| NEPENTHE(iter #1) | 0.001 | 0.252 | 0.263 | 0.298 | 0.310 | 0.325 | 64.26 | 0/12 | 0h03 |
| NEPENTHE(iter #2) | 0.001 | 0.271 | 0.295 | 0.322 | 0.347 | 0.351 | 68.23 | 0/12 | 0h04 |
| NEPENTHE(iter #3) | 0 | 0.001 | 0.265 | 0.295 | 0.363 | 0.380 | 66.06 | 1/12 | 0h05 |
| NEPENTHE(iter #4) | 0 | 0 | 0.326 | 0.333 | 0.344 | 0.349 | 54.15 | 2/12 | 0h06 |
| NEPENTHE(iter #5) | 0 | 0 | 0.263 | 0.359 | 0.392 | 0.399 | 59.93 | 2/12 | 0h07 |
| NEPENTHE(iter #6) | 0 | 0 | 0.001 | 0.001 | 0.325 | 0.387 | 54.15 | 2/12 | 0h08 |
| NEPENTHE(iter #7) | 0 | 0 | 0.001 | 0.001 | 0.001 | 0.390 | 59.21 | 2/12 | 0h09 |

Table 44: Test performance (top-1), bottom six layer's entropies $\widehat{\mathcal{H}}$ and the number of removed layers (Rem.) for BERT on SST-2.

| Approach | $\widehat{\mathcal{H}}_1$ | $\widehat{\mathcal{H}}_2$ | $\widehat{\mathcal{H}}_3$ | $\widehat{\mathcal{H}}_4$ | $\widehat{\mathcal{H}}_5$ | $\widehat{\mathcal{H}}_6$ | top-1 | Rem. | Time |
|---|---|---|---|---|---|---|---|---|---|
| Dense model | 0.114 | 0.122 | 0.130 | 0.172 | 0.178 | 0.205 | 92,20 | 0/12 | 0h21 |
| IMP(iter #1) | 0.119 | 0.126 | 0.139 | 0.190 | 0.213 | 0.226 | 91.63 | 0/12 | 0h41 |
| IMP(iter #2) | 0.138 | 0.141 | 0.161 | 0.241 | 0.264 | 0.267 | 90.83 | 0/12 | 1h01 |
| IMP(iter #3) | 0.133 | 0.144 | 0.185 | 0.239 | 0.280 | 0.307 | 90.48 | 0/12 | 1h22 |
| IMP(iter #4) | 0.142 | 0.167 | 0.205 | 0.251 | 0.303 | 0.332 | 90.37 | 0/12 | 1h43 |
| IMP(iter #5) | 0.158 | 0.181 | 0.219 | 0.279 | 0.312 | 0.376 | 89.91 | 0/12 | 2h04 |
| IMP(iter #6) | 0.178 | 0.215 | 0.241 | 0.322 | 0.361 | 0.363 | 88.53 | 0/12 | 2h25 |
| IMP(iter #7) | 0.187 | 0.226 | 0.270 | 0.365 | 0.370 | 0.466 | 87.96 | 0/12 | 2h46 |
| NEPENTHE(iter #1) | 0.001 | 0.132 | 0.137 | 0.182 | 0.224 | 0.229 | 91.17 | 0/12 | 0h48 |
| NEPENTHE(iter #2) | 0.001 | 0.001 | 0.191 | 0.209 | 0.238 | 0.246 | 89.68 | 0/12 | 1h16 |
| NEPENTHE(iter #3) | 0 | 0 | 0 | 0.193 | 0.205 | 0.224 | 89.00 | 3/12 | 1h44 |
| NEPENTHE(iter #4) | 0 | 0 | 0 | 0.001 | 0.204 | 0.243 | 88.88 | 3/12 | 2h12 |
| NEPENTHE(iter #5) | 0 | 0 | 0 | 0.001 | 0.228 | 0.254 | 87.39 | 3/12 | 3h01 |
| NEPENTHE(iter #6) | 0 | 0 | 0 | 0.001 | 0.246 | 0.252 | 88.76 | 3/12 | 3h29 |
| NEPENTHE(iter #7) | 0 | 0 | 0 | 0.001 | 0.245 | 0.253 | 87.73 | 3/12 | 3h57 |

Table 45: Test performance (top-1), bottom six layer's entropies $\widehat{\mathcal{H}}$ and the number of removed layers (Rem.) for RoBERTa on SST-2.

| Approach | $\widehat{\mathcal{H}}_1$ | $\widehat{\mathcal{H}}_2$ | $\widehat{\mathcal{H}}_3$ | $\widehat{\mathcal{H}}_4$ | $\widehat{\mathcal{H}}_5$ | $\widehat{\mathcal{H}}_6$ | top-1 | Rem. | Time |
|---|---|---|---|---|---|---|---|---|---|
| Dense model | 0.131 | 0.145 | 0.152 | 0.187 | 0.192 | 0.278 | 92.66 | 0/12 | 0h21 |
| IMP(iter #1) | 0.129 | 0.151 | 0.153 | 0.189 | 0.196 | 0.276 | 92.32 | 0/12 | 0h41 |
| IMP(iter #2) | 0.136 | 0.155 | 0.156 | 0.186 | 0.226 | 0.266 | 92.43 | 0/12 | 1h01 |
| IMP(iter #3) | 0.159 | 0.173 | 0.179 | 0.195 | 0.234 | 0.346 | 93.23 | 0/12 | 1h22 |
| IMP(iter #4) | 0.178 | 0.180 | 0.215 | 0.227 | 0.304 | 0.374 | 92.66 | 0/12 | 1h43 |
| IMP(iter #5) | 0.178 | 0.199 | 0.253 | 0.293 | 0.319 | 0.419 | 92.20 | 0/12 | 2h04 |
| IMP(iter #6) | 0.177 | 0.208 | 0.226 | 0.280 | 0.323 | 0.416 | 91.17 | 0/12 | 2h25 |
| IMP(iter #7) | 0.174 | 0.200 | 0.220 | 0.285 | 0.344 | 0.428 | 90.83 | 0/12 | 2h46 |
| NEPENTHE(iter #1) | 0.001 | 0.128 | 0.160 | 0.161 | 0.202 | 0.255 | 92.32 | 0/12 | 0h48 |
| NEPENTHE(iter #2) | 0.001 | 0.001 | 0.166 | 0.205 | 0.216 | 0.251 | 92.09 | 0/12 | 1h16 |
| NEPENTHE(iter #3) | 0 | 0.001 | 0.162 | 0.197 | 0.283 | 0.322 | 90.83 | 1/12 | 1h44 |
| NEPENTHE(iter #4) | 0 | 0 | 0.191 | 0.198 | 0.257 | 0.267 | 90.60 | 2/12 | 2h12 |
| NEPENTHE(iter #5) | 0 | 0 | 0.191 | 0.212 | 0.281 | 0.345 | 90.48 | 2/12 | 3h01 |
| NEPENTHE(iter #6) | 0 | 0 | 0.001 | 0.210 | 0.293 | 0.378 | 90.71 | 2/12 | 3h29 |
| NEPENTHE(iter #7) | 0 | 0 | 0.001 | 0.215 | 0.272 | 0.275 | 89.68 | 2/12 | 3h57 |

### D.5 Practical Sanity Checks on Gaussian Assumptions

Our derivation is based on the assumption that the weights of a layer and the inputs for its corresponding rectifier activation are Gaussian distributed. We validate here this assumption empirically. Fig. 7, 9, 11, 13, 15 show the weights distribution for each rectifier activated layer for all the models employed in our experiments. Moreover, Fig. 8, 10, 12, 14, 16 show the input distribution for each rectifier layer for each model used in our experiments. Inputs from CIFAR-10 are used for ResNet-18, Swin-T, and MobileNet-V2 models, while inputs from QNLI are employed for BERT and RoBERTa models. It appears that the weights and the inputs of the corresponding layers are following a Gaussian distribution.

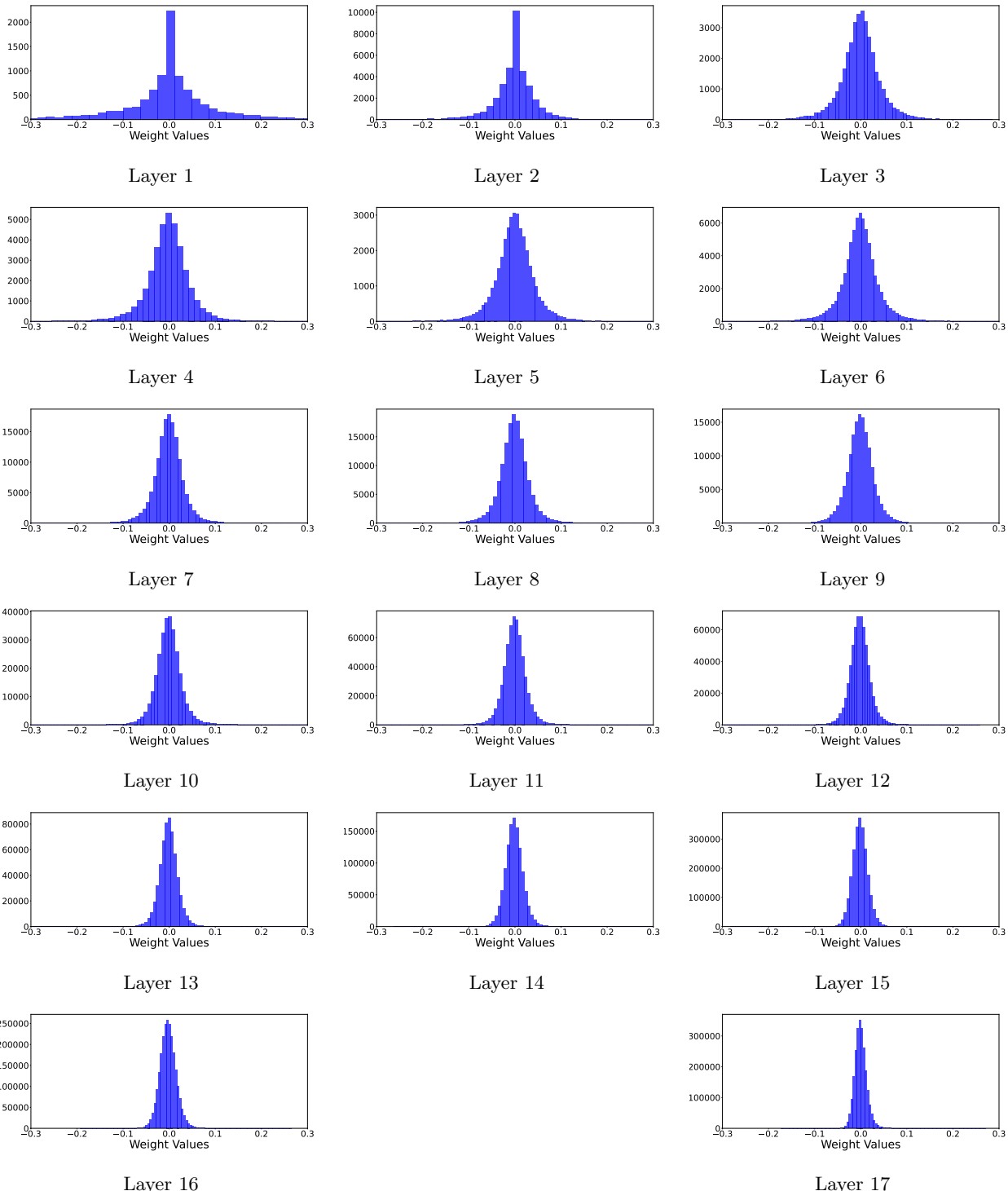

Figure 7: Weights distribution of each layer in ResNet-18.

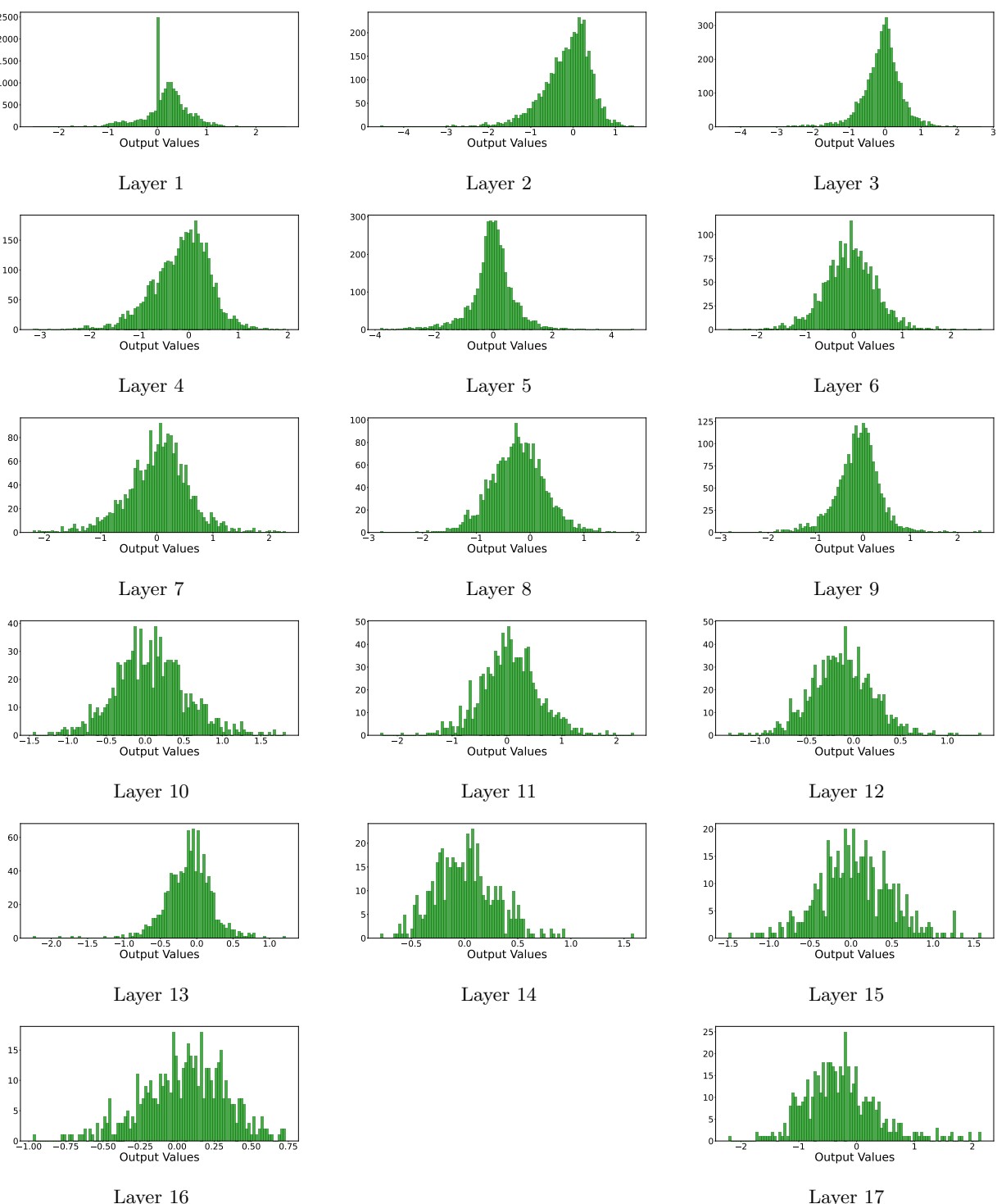

Figure 8: Input distribution of each layer in ResNet-18.

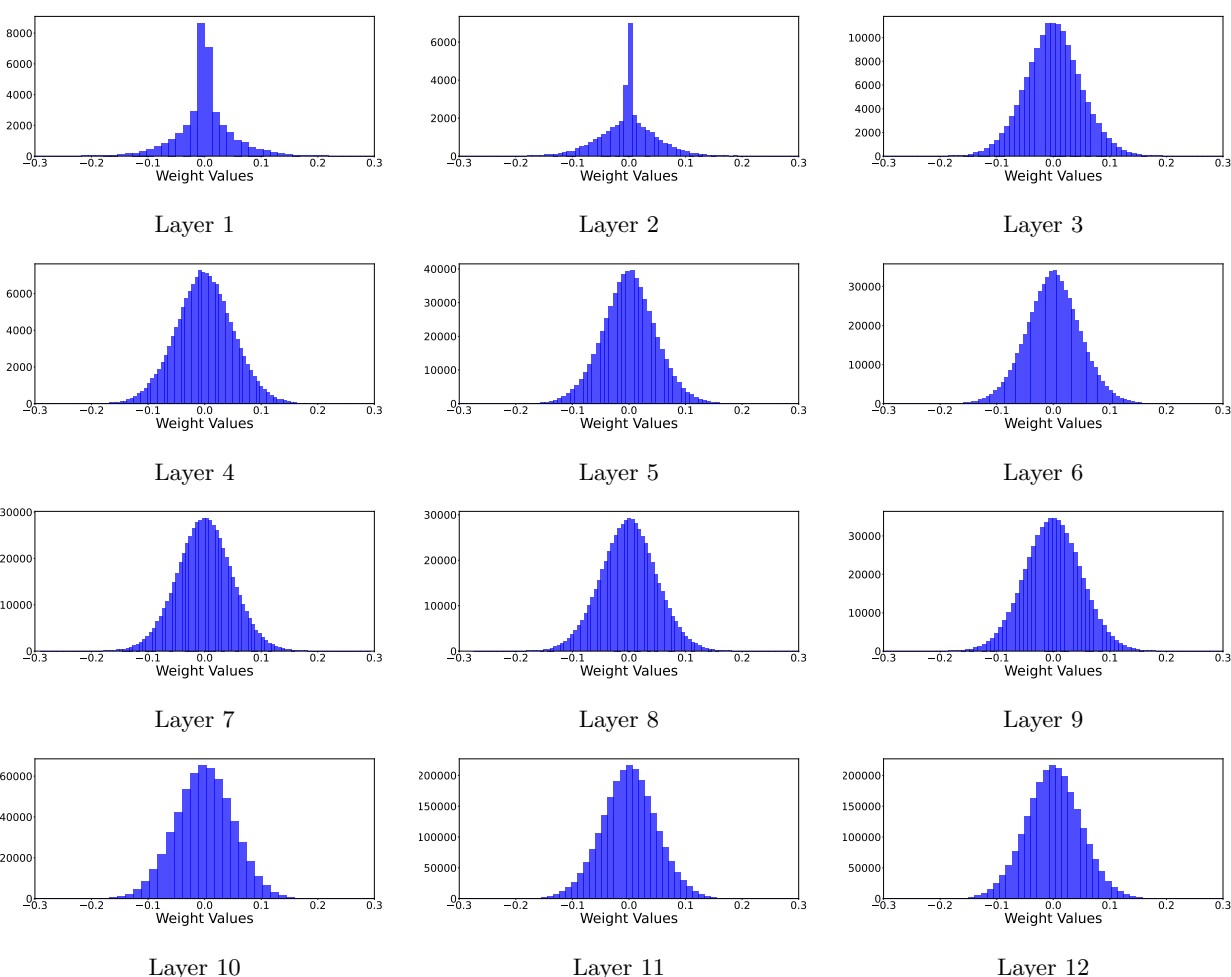

Figure 9: Weights distribution of each layer in Swin-T.

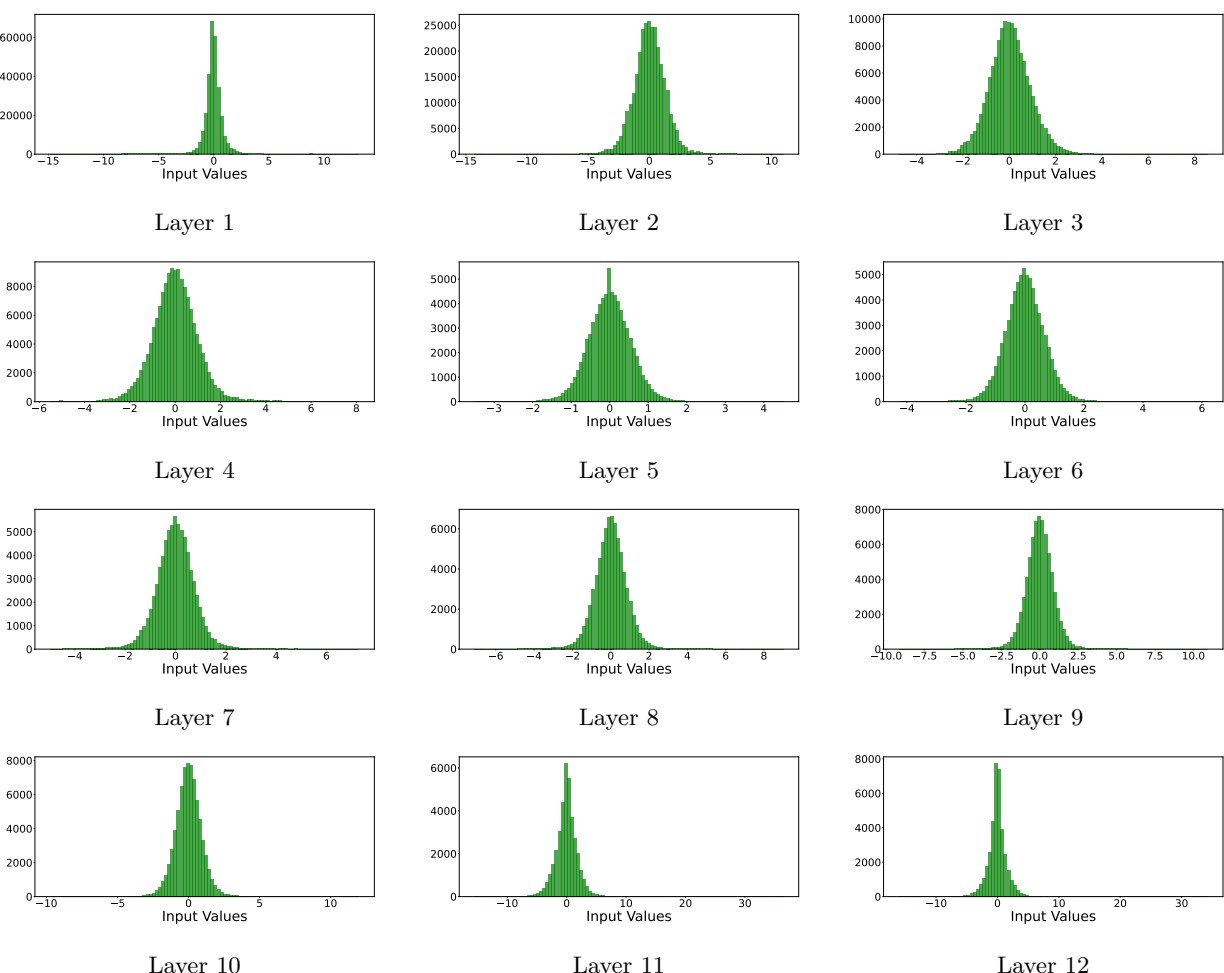

Figure 10: Input distribution of each layer in Swin-T.

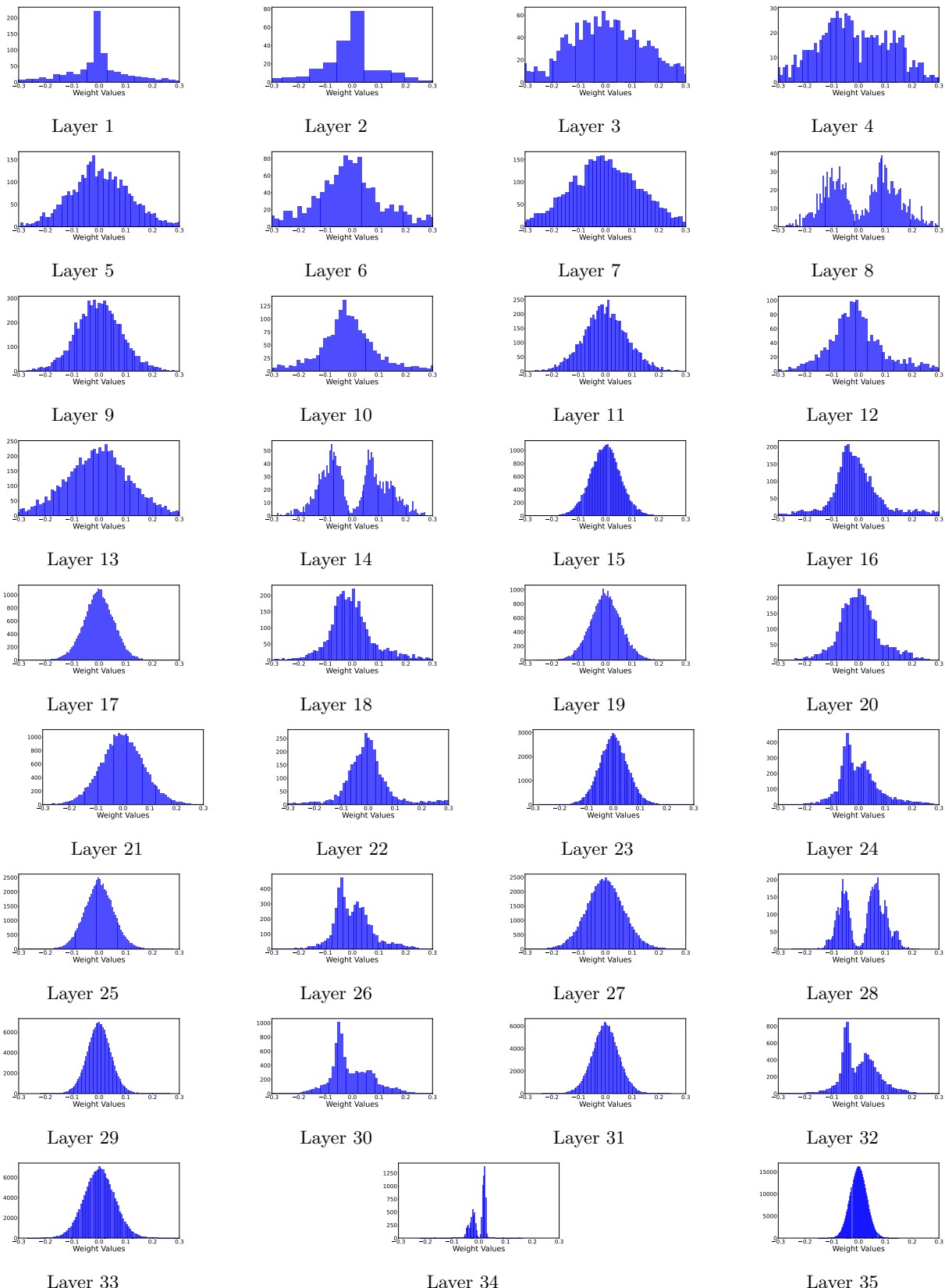

Figure 11: Weights distribution of each layer in MobileNet.

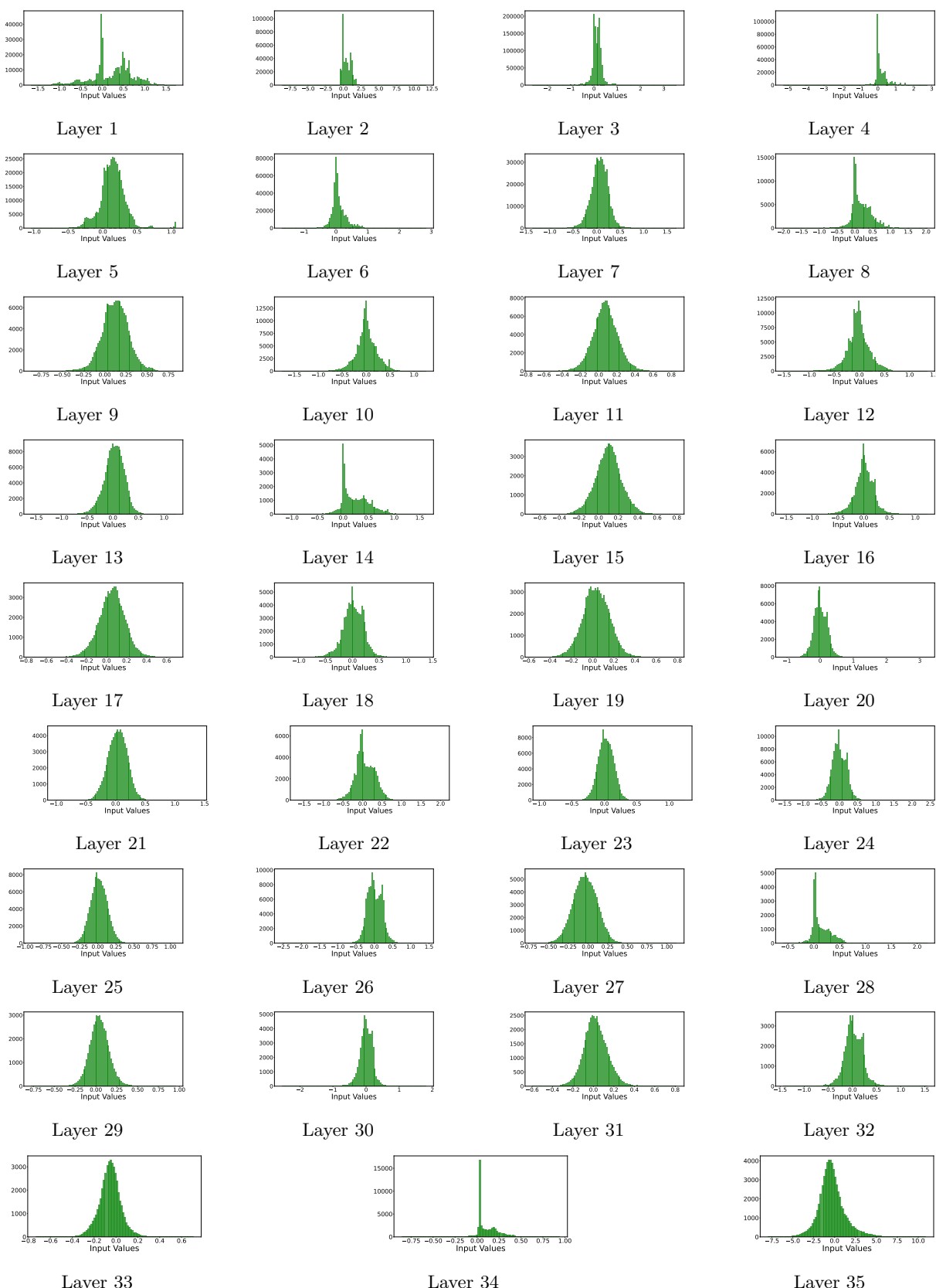

Figure 12: Input distribution of each layer in MobileNet.

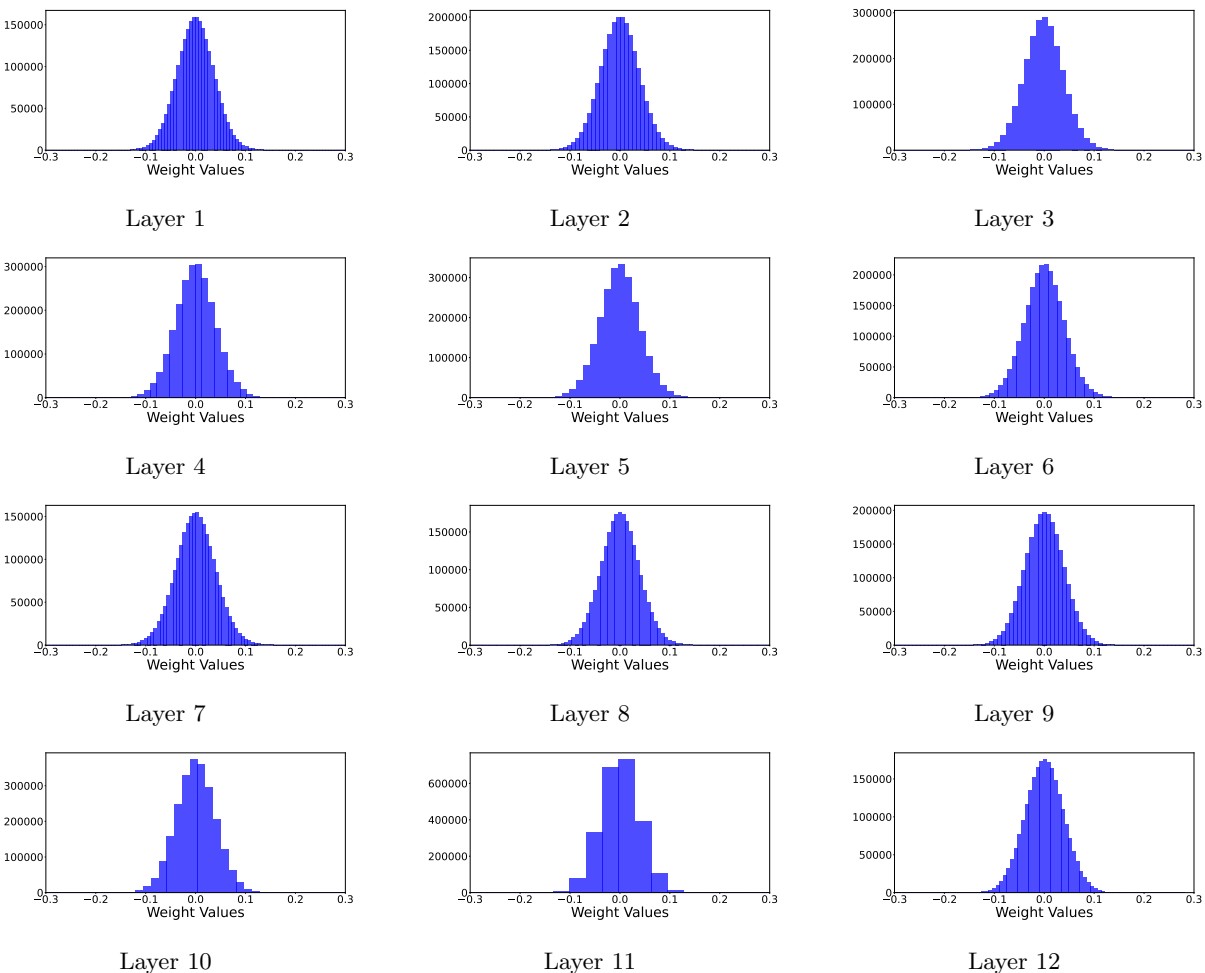

Figure 13: Weights distribution of each layer in BERT.

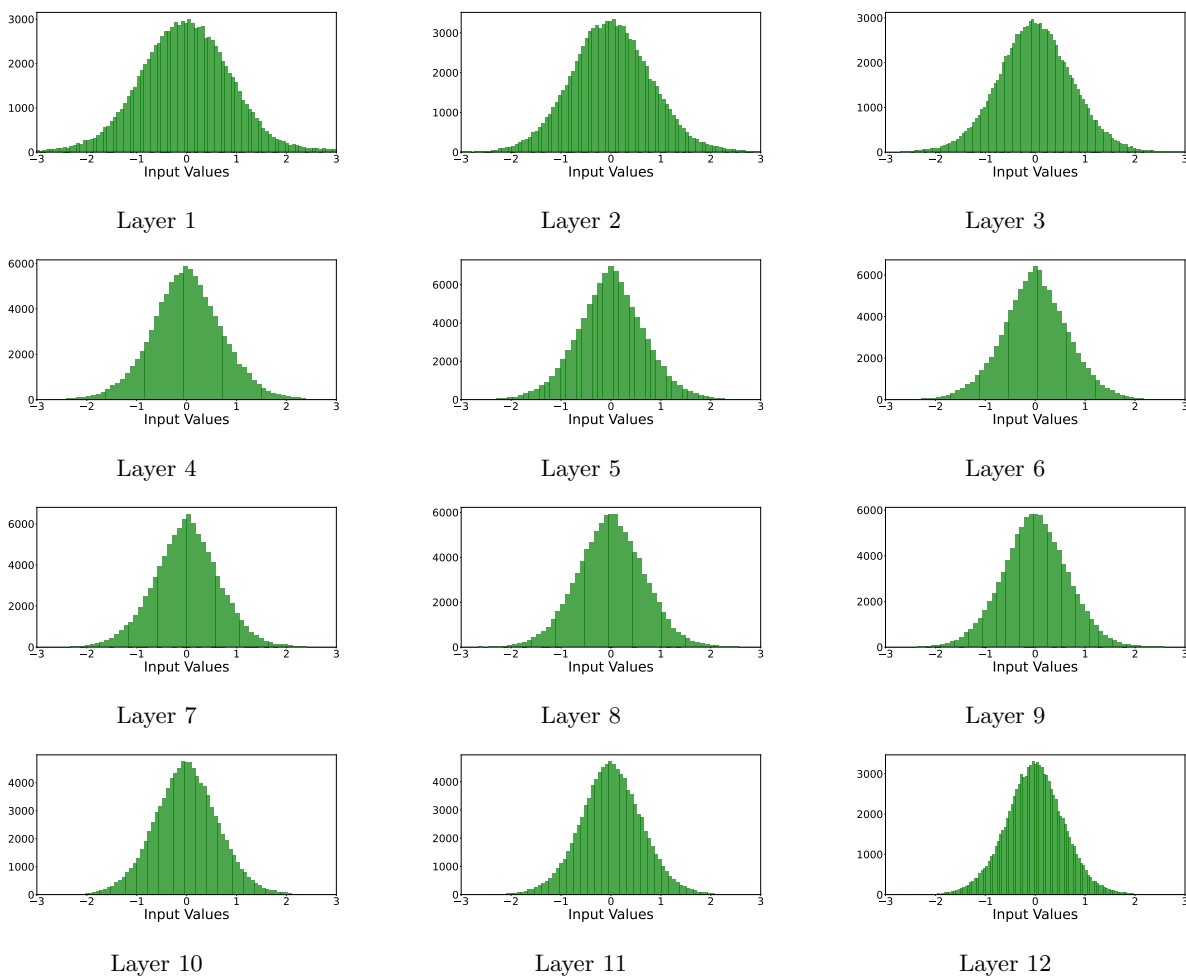

Figure 14: Input distribution of each layer in BERT.

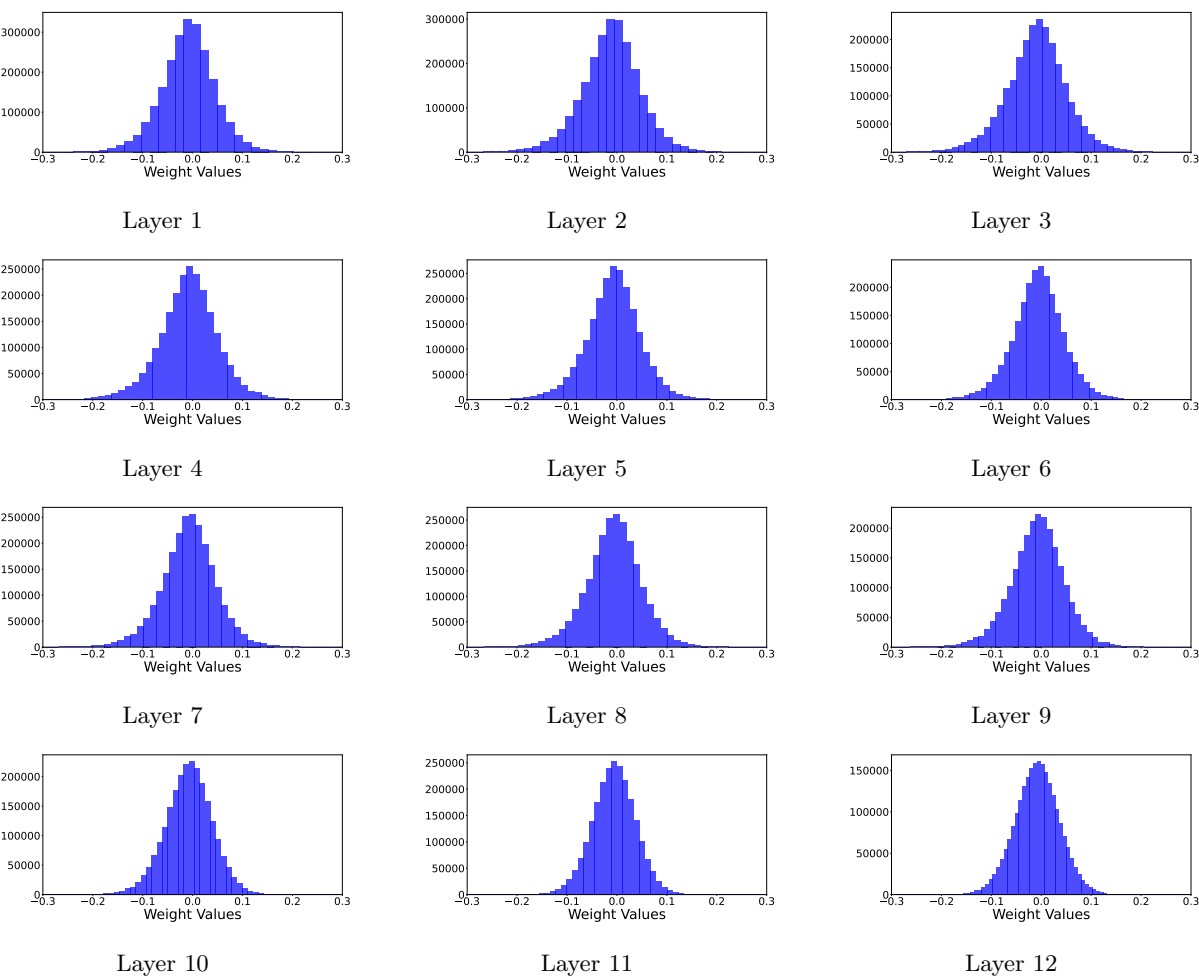

Figure 15: Weights distribution of each layer in RoBERTa.

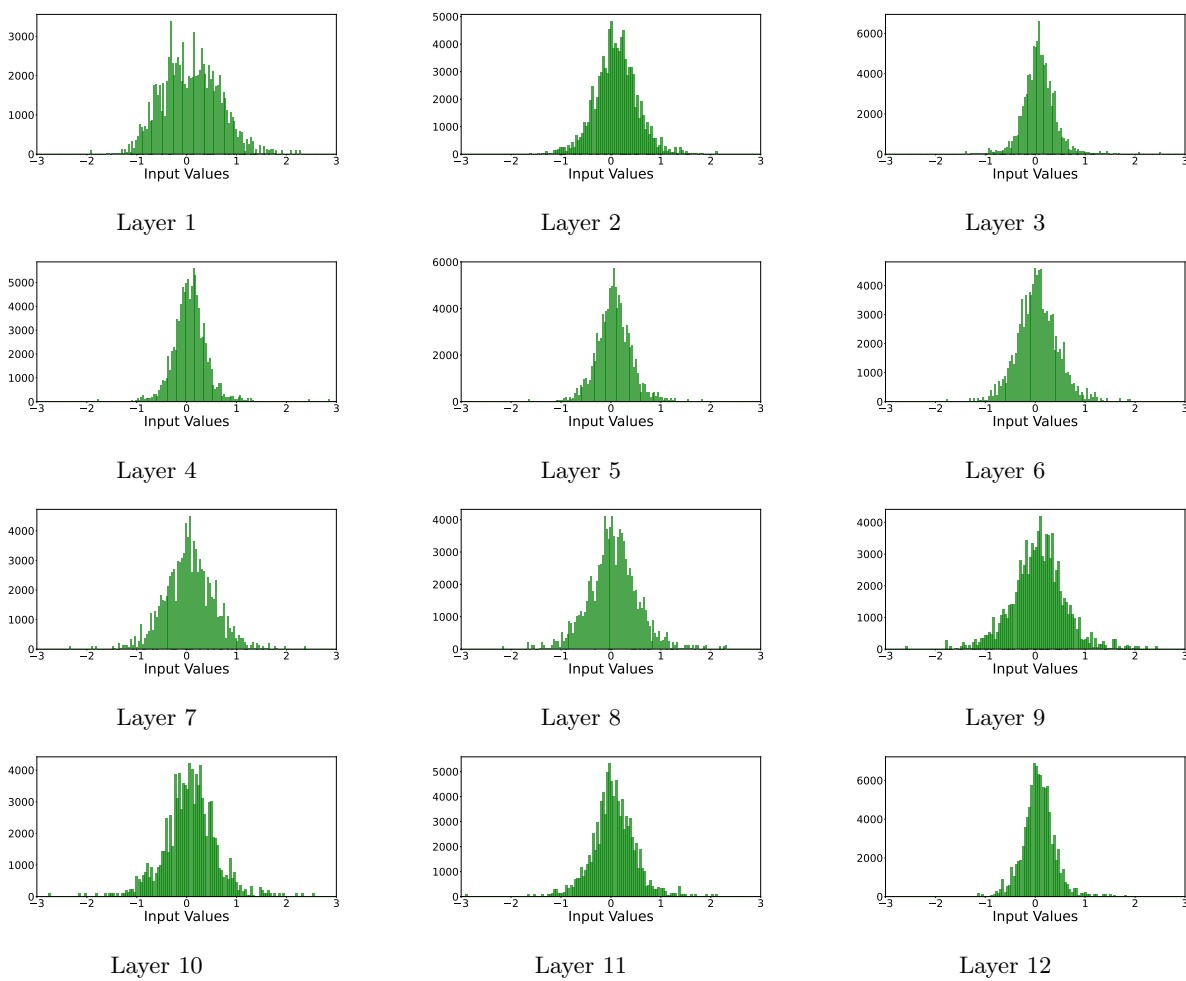

Figure 16: Input distribution of each layer in RoBERTa.

