# OpenReview forum: "Layer Collapse Can be Induced by Unstructured Pruning"
_TMLR — Accepted by TMLR_

### Review · Reviewer_3uDY · 2025-11-06

**Summary Of Contributions:**

This manuscript proposed the NEPENTHE framework for pruning strategy. The authors show that unstructured pruning lowers nonlinearity of neural networks, which is quantified with entropy. To the end, the authors show that NEPENTHE can also fully remove certain layer(s). Experiments on several networks, such as ResNet-18, MobileNet-V2, and Swin-T, demonstrate the successful removal of layers.

**Audience:**

Yes

**Audience Explanation:**

Researchers on pruning would find certain values in this study.

**Claims And Evidence:**

No

**Claims Explanation:**

I think current version is not sufficient. Please see the Requested Changes below.

**Requested Changes:**

Regarding the removing the number of layers
- “… the maximum number of layers traversed during forward propagation.” Regarding this part, I understand that the authors succeeded in removing the number of layers with unstructured pruning, but I encourage the authors to explain and clarify why the number of layers matters—why it is so important. The connection between the number of layers and performance (possibly computational cost) is not clear enough.
- Indeed, for results such as Table 2, the authors mainly report top-1 accuracy and the number of removed layers (Rem.). I understand that the authors succeeded in removing the number of layers, but the authors should verify that the gain in Rem. directly leads to a gain in (possibly) computational costs.
- Specifically, I encourage the authors to expand Table 10 and present FLOPs, latency, and memory in practice. Rem. might be allowed to be a supplementary index, but rather than presenting successful results in Rem., the authors should focus on gains in these computational costs.

Method and Equations
- Is it correct to define $s$ based on $y$, as in Eq. 2? According to Eq. 1, $y$ is the output of the rectifier, and I think using $z$ would be natural.
- Does the distribution $f_Z$ with Eqs. 9 and 10 correctly quantify variance using $\sigma_X$ or $\sigma_W$? The authors state that $Z$ is obtained by a product of $X$ and $W$, but they are not reflected in Eqs. 9 and 10. This part would affect others, such as Eq. 11.
- Eq. 13 is inconsistent with Algorithm 2. Specifically, Eq. 13 uses $w_l$ in the denominator, whereas Algorithm 2 uses $w_{l, i}$ in the denominator. I have checked the source code in supplementary materials, and I think that the denominator should be layerwise one with $w_l$, right?
- Also, I have checked the source code for the implementation of Eq. 13. Overall, the source code applies three steps of
    1) layer_entro_magni[name] = layers_entropy[name] * torch.mean(torch.abs(module.weight[module.weight!=0]))
    2) entropy_magni_layer_head[name]  = total_layers_entro_magni/(layer_entro_magni[name])
    3) entropy_layer_head_expo[name]  = torch.exp(entropy_magni_layer_head[name] - max_value_entropy_magni_layer_head).item()

    Although this implementation roughly matches with Eq. 13, it does not correctly implement Eq. 13. In the source code, the absolute value of weight is averaged first, which is different from Eq. 13. Please check this point.
- The relevance score $R_l$ is stated to be larger than 1 after Eq. 14, but I think that it can be lower than 1.
- If one defines $p(s)=0$ in a certain case, it would cause $\log{0}$ and NaN value.

Regarding experiments
- For Figure 3, the y-axis is labeled as Number of neurons. However, I think these values of 64 to 512 would indicate the number of channels, not the neurons. Please check this point.
- I think that the authors primarily focused on the small or tiny models such as ResNet-18, MobileNet-V2, and Swin-T. Although I understand that applying pruning would matter in obtaining these faster networks, I think that pruning methods would matter in large models. Specifically, one may think that these smaller models might allow removal of certain layers, whereas other large models may not allow removal of certain layers, as the large model contains more neurons that are difficult to remove.
- The GLEU benchmark should be expanded to full results. The current Table 4 only reports part of them.
- I appreciate the ImageNet results on Table 15, but the top-1 accuracy of 68.20% looks like a weak baseline.

Writing should be improved.
- “loss.However, …” → “loss. However, …”
- “acitvated” → “activated”
- “NPENTHE” → “NEPENTHE”
- “In this case, The output” → “In this case, the output”

---

> ### Author Response · Authors · 2025-12-31
>
> **[The number of layer removal]**
> Thank you for the suggestion. We agree that compute metrics are important in practice. At the same time, removed layers (Rem.) is a standard, model-agnostic structural indicator that directly reflects shorter forward paths and smaller activation footprints, and is less sensitive to hardware, batch size, or kernel choices. To balance both views, in the revision we keep Rem. as the primary structural summary in the main tables and expand Appendix C.3 with compute results (FLOPs, latency, memory) across more tasks and hardware.
>
> ---
>
> **[s definition in Eq.2]**
> Thank you for pointing this out. You are right. We have revised Eq. (2) and now the state s is defined based on z.
>
> **[Distribution $f_Z$]**
> We thank the reviewer for this helpful remark. In the revised version, we keep Eqs.(7) and (8) in their general form $X \sim \mathcal{N}(\mu_X,\sigma_X^2)$ and $W \sim \mathcal{N}(\mu_W,\sigma_W^2)$, and we now make explicit that, when deriving Eqs.(9) and (11), we work with standardised variables of unit variance. Specifically, right after Eq.(8) we add a short paragraph introducing $\tilde X = (X-\mu_X)/\sigma_X$ and $\tilde W = (W-\mu_W)/\sigma_W$, and the normalised pre-activation $\tilde Z = \tilde X \tilde W = Z/(\sigma_X \sigma_W)$, which we then denote again by $Z$ for notational simplicity. This is exactly the standard procedure used in the classical derivation of the product-of-Gaussians law, where the distribution is first obtained in standard units and the general-variance case is recovered by rescaling [R1]. Thus, Eqs.(9) and (11) are written for the normalised pre-activation, in which $\sigma_X$ and $\sigma_W$ are absorbed into the scaling, and our clarification makes their role explicit without changing any derivation or experimental result.
>
> **[Eq. 13 is inconsistent with Algorithm 2]**
> Thank you for pointing it out. We realise our earlier notation could cause some confusion. We have revised Eq. (13), and now it matches our implementation exactly.
>
> **[The relevance score]**
> $R_l$ can not be lower than 1. In our definition (Eq. 14), the numerator includes $I_l$ for the same layer, so $\sum_j I_j \ge I_l$, hence $R_l \ge 1$.
>
> **[What if p=0]** We clamp probabilities in code (e.g., torch.clamp(p_one, min=1e-5)), which prevents log-zero/NaN issues.
>
> ---
>
> **[Label of Figure 3]** Thank you for pointing it out. In ResNet-18, the values of 64 to 512 denote output channels/filters per stage, not the number of neurons. We relabeled Figure 3 in the revised version.
>
> **[Applicability on larger models]** On a fixed dataset, larger networks (more parameters/channels) are more likely to be over-parameterized. For example, in Appendix C.2, we already include ResNet-50 and ResNet-152 on CIFAR-10, where NEPENTHE removes many layers (e.g., 82/151 on ResNet-152) while improving accuracy by ~4%.
>
> **[Full results of GLEU benchmark]** Thank you for the suggestion. In the revised version, we have expanded Table 4 to report the full GELU benchmark results. We will further improve the layout and presentation in the camera-ready version.
>
> **[ImageNet results]** According to Facebook’s reference ResNet implementation [R2], top-1 accuracy for ResNet-18 trained on ImageNet is 69.57%. Given unavoidable differences in hardware, preprocessing, training schedule, and random seeds, our baseline is within expected variance and acceptable.
>
> **[Writing improvement]** Thank you for pointing it out. We have improved them in the revised version.
>
> ---
>
> ***[R1]: Craig, 1936, ``On the Frequency Function of $xy$'', Annals of Mathematical Statistics 7(1):1--15***
>
> ***[R2]: https://github.com/facebookarchive/fb.resnet.torch/blob/master/pretrained/README.md***

---

### Review · Reviewer_YfNN · 2025-11-10

**Summary Of Contributions:**

- The manuscript shows that it is possible, via unstructured pruning methods, to compress an architecture by removing  layers from it and keep competitive performance.

- The manuscript introduces a entropy-based metric for units/neurons of  network. This metric can indicate the level to which a neuron follows a linear behavior, and consequently, be removed.

- The manuscript proposes a method (NEPENTHE) which induces such low-entropy within layers of an architecture, thus improving compression/pruning performance.

**Audience:**

Yes

**Audience Explanation:**

The core of the paper is the design of mechanisms to estimate the relevance of a unit (neuron), within a neural network architecture, in the decision-making process. Beyond the model pruning/compression problem explored by the manuscript, such mechanisms are also valuable for other research lines, e.g. lottery ticket extraction, continual/lifelong learning and interpretability. All these directions are significantly represented within the work covered by the journal.

**Broader Impact Concerns:**

To the best of my judgement, the contents of the manuscript do not have any ethical implication.

**Claims And Evidence:**

Yes

**Claims Explanation:**

The manuscript properly pairs every contribution claimed in Page-2, with theoretical/empirical evidence to support it.

**Requested Changes:**

- The Ablation study is mostly focused around the ResNet-18 + CIFAR-10 combination, from this combination however, it is hard to assess the level to which the observations made for the ResNet-18 + CIFAR-10 would hold for other model-dataset combination. The ablation study would benefit from extending the reported results to other combinations that bring more complementarity, e.g. a transformer architecture, a different data modality (text), a more complex/larger image dataset, etc., to the one already present in the paper.

- The proposed method operates under the assumption that the weights of a layer and the inputs of its corresponding rectifier activation function follow a Gaussian distribution (Sec. 3.2). While the plots presented in C.4 show that that seems to be the case for the conducted experiments. The manuscript would strengthen from indicating how

- In the list of contributions reported in Page-2, the third item is related to the proposal of the NEPENTHE method while the fourth is related to its empirical validation. In my opinion, these two items must be merged as the proper validation of a proposed method is not a plus but a must.

- [minor] In Table 2, there seem to be a typo in the [ResNet-18, EGP, VLCS] results reported for the number of remove layers (“Rem.”) as the total number of layers seems to be reduced to 12 (when in practice should be 17).

---

> ### Author Response · Authors · 2025-12-31
>
> **[Ablation study on more tasks]**
> Thank you for the suggestion. In the revision, Appendix C.2 now includes an ablation of the three key NEPENTHE components on a transformer (Swin-T) trained on Tiny-ImageNet, and the trends are consistent with ResNet-18/CIFAR-10. We also expand Appendix C.3 with compute results (FLOPs, latency, memory) across additional tasks and hardware.
>
> **[Justification for the Gaussian assumption]**
> It seems that the reviewer’s comment was cut off midway, so we can only respond based on our assumption that the reviewer is asking for additional theoretical justification for the Gaussian assumption.  The Gaussian assumption in Sec. 3.2 is motivated by the Central Limit Theorem (CLT):  in wide neural networks, the pre-activation z is a sum of many weakly dependent terms, and such linear combinations converge toward a Gaussian distribution [R3, R4]. We will include this clarification in the revised paper.
>
> **[List of contributions]**
> Thank you for your suggestion. We have merged these two items in the revised version.
>
> **[Writing improvement]** Thank you for pointing it out. We have corrected it in the revised version.
>
> ---
>
> ***[R3] Neal, Radford M. Bayesian learning for neural networks. Vol. 118. Springer Science & Business Media, 2012.***
>
> ***[R4] Lee, Jaehoon, et al. "Deep neural networks as gaussian processes." arXiv preprint arXiv:1711.00165, 2017.***

---

### Review · Reviewer_Sebj · 2025-12-29

**Summary Of Contributions:**

This paper investigates the relationship between unstructured pruning (removing individual weights based on magnitude) and the effective depth of deep neural networks (DNNs). The authors challenge the conventional belief that unstructured pruning reduces parameter count without affecting the network's structural depth (critical path). They introduce the concept of Neuron Entropy, a metric quantifying how much a neuron utilizes the non-linear regions of its activation function. The authors demonstrate theoretically and empirically that magnitude-based pruning naturally lowers neuron entropy in rectifier-activated networks. When a layer’s average entropy drops to zero, the activation effectively becomes an identity function (or a constant zero), rendering the layer linear. Linear layers can be mathematically merged with adjacent layers, allowing for their complete removal. Building on this, the authors propose NEPENTHE, an iterative algorithm that allocates higher pruning rates to low-entropy layers to force them into a linearizable state. Experiments across CNNs, Vision Transformers, and NLP models show that NEPENTHE can successfully remove multiple layers from over-parameterized networks with little to no performance degradation, often outperforming standard iterative magnitude pruning (IMP) and other depth-reduction baselines.

**Audience:**

Yes

**Audience Explanation:**

The paper connects two significant areas of research, i.e., unstructured pruning and network depth reduction, in a novel and counter-intuitive way. It challenges the common belief that unstructured pruning reduces parameter count without affecting the network's structural depth or critical path. By demonstrating that magnitude-based pruning can naturally lead to "zero-entropy" (linearizable) layers, the authors provide a fresh perspective on network redundancy.

**Broader Impact Concerns:**

None.

**Claims And Evidence:**

No

**Claims Explanation:**

Strengths:-

- The paper provides a compelling theoretical bridge between two distinct compression paradigms: unstructured pruning and layer removal (depth reduction).

- The formulation of neuron entropy in Section 3.1 is mathematically grounded and intuitive. By discretizing neuron outputs into "ON" (linear region) and "OFF" (dead region) states, the authors create a quantifiable metric for non-linearity.



Weaknesses:-

- The core theoretical premise relies on the behavior of Rectified Linear Units (ReLU), where the "ON" state ($x > 0$) is strictly linear ($f(x) = x$). However, the paper applies this logic to architectures like Swin-T and BERT, which use GELU, and MobileNet, which uses ReLU6 (or SiLU variants in modern implementations). In Footnote 3, the authors state: "or very close as in GeLU." (Page 4). This is a significant unacknowledged weakness. GELU and SiLU are non-monotonic and non-linear *everywhere*, even for positive inputs. Unlike ReLU, a GELU layer in the "ON" state cannot be mathematically collapsed into the adjacent linear layer without introducing approximation error. Merging a GELU layer that is "mostly positive" is not a lossless operation. The paper treats these activations as effectively linearizable, but strictly speaking, they are not. The error propagation from this "approximate linearization" in deep Transformers is not theoretically modeled, despite the empirical success.


- The paper argues that once a layer is linear, it can be "absorbed by the following layer" (Section 3.1). While this is algebraically true ($W_{combined} = W_{l+1} \times W_l$), the authors ignore the computational reality of merging two *unstructured sparse* matrices. If Layer $l$ and Layer $l+1$ are both pruned via unstructured pruning (which NEPENTHE does), their product $W_{combined}$ often results in a matrix with a different, potentially denser sparsity pattern (the "fill-in" problem). Unless the merged matrix $W_{combined}$ is significantly sparse, the computational savings (FLOPs) might be negligible or even negative compared to keeping the two sparse layers separate, specifically on hardware optimized for specific sparsity formats.
The paper claims FLOP reductions in Table 10 but does not detail if these calculations account for the potential density increase when fusing consecutive linear layers. They imply removal is trivial, but in unstructured pruning contexts, matrix fusion is non-trivial for preserving sparsity ratios.


- The title and abstract claim that unstructured pruning "induces" layer collapse. However, the experimental data suggests that standard unstructured pruning (IMP) rarely completes the job. In Table 1, standard IMP reduces entropy to $0.055$ (for layer 1) but never reaches $0$. The text admits: "IMP does not support the removal of any layers despite successfully maintaining performance." (Section 4.3). Therefore, layer collapse is *not* a natural byproduct of standard unstructured pruning as usually applied; it is a potential state that requires the specific, forcing intervention of the NEPENTHE algorithm to fully realize. The title is slightly misleading, as "Unstructured Pruning" (the general technique) does not induce collapse; "Entropy-Guided Pruning" does.


- The paper compares NEPENTHE against pruning baselines (IMP, EGP, Hrank). However, since the primary outcome is a shallower network (e.g., ResNet-18 minus 3 layers), a critical missing baseline is training a standard ResNet-15 (or equivalent depth-reduced architecture) from scratch. If a ResNet-18 with 3 layers removed performs identically to a ResNet-18 pruned by NEPENTHE, it proves the method finds the optimal sub-network. If the trained-from-scratch shallower network performs better, it suggests NEPENTHE is useful for compression but not necessarily finding the optimal architecture. The absence of this "shallow-dense" baseline makes it hard to contextualize the effectiveness of the resulting topology.

**Requested Changes:**

Please address the weaknesses I mentioned above.

---

> ### Author Response · Authors · 2025-12-31
>
> **[w1 - ON state for rectifiers]**
>
> We thank the reviewer for this important and technically correct observation. We fully agree that GELU and SiLU are not strictly linear in their positive regime, and that collapsing such layers is not a mathematically lossless operation. Below, we clarify the scope of our claim, formalize the approximation error, and show why this error is negligible under the training and pruning regime considered in this work.
>
> ---
>
> Consider a neuron with pre-activation
> $$
> z(x) = w^\top x + b,
> $$
> and activation function $\phi \in \{\mathrm{GELU}, \mathrm{SiLU}\}$.
> Layer collapse corresponds to replacing $\phi(z)$ by the identity map $z$.
>
> The neuron-level approximation error is therefore
> $$
> \epsilon(z) := \phi(z) - z.
> $$
>
> Since layer removal is evaluated at the dataset level, the relevant quantity is the expected error under the data distribution:
> $$
> \mathcal{E} := \mathbb{E}_{x \sim \mathcal{D}}\left[\|\phi(z(x)) - z(x)\|_2^2\right].
> $$
>
> Our claim is not that $\mathcal{E}=0$, but that $\mathcal{E}$ becomes negligible when a layer’s entropy approaches zero under unstructured pruning and weight decay.
>
> Both GELU and SiLU can be written in the form
> $$
> \phi(z) = z - r(z),
> $$
> where the residual nonlinearity $r(z)$ satisfies strong tail decay:
> - **GELU**: $|r(z)| \le C \exp(-\alpha z^2)$,
> -  **SiLU**: $|r(z)| \le C \exp(-\alpha z)$,
>
> for some constants $C,\alpha>0$. In particular, $\phi'(z) \to 1$ and $\phi''(z) \to 0$ exponentially fast as $z \to +\infty$.
> Thus, the deviation from linearity is concentrated in a narrow neighborhood around $z \approx 0$.
>
> ---
> In our setting, three mechanisms jointly shape the distribution of pre-activations $z$:
>
> 1. **$ℓ_2$ weight decay** controls the operator norm and variance of the weights.
> 2. **Magnitude-based unstructured pruning** removes small weights, further reducing variance.
> 3. **Entropy minimization** enforces that neurons remain almost always in the same activation regime.
>
> As shown in Sec. 3.2, this results in pre-activations that are well-approximated by
> $$
> z \sim \mathcal{N}(\mu_z, \sigma_z^2),
> \quad \text{with} \quad \mu_z \gg \sigma_z,
> $$
> i.e., the distribution is strongly biased toward the positive regime and increasingly concentrated as pruning progresses.
>
> Using the decay properties above, the expected neuron-level error can be bounded as
> $$
> \mathbb{E}[\epsilon(z)^2]
> \le
> \int r(z)^2 \mathcal{N}(z, \mu_z, \sigma_z^2) dz
> \le
> C \exp(-\alpha \mu_z^2),
> $$
> for GELU (and analogously exponential in $\mu_z$ for SiLU).
>
> Therefore, once a neuron is in a low-entropy (almost-always-ON) regime, the approximation error decays exponentially in the mean pre-activation. For a layer with $n$ neurons,
> $$
> \mathcal{E}_{\text{layer}} \le n\,C\,\exp(-\alpha \mu_l^2),
> $$
> which rapidly becomes negligible in practice.
>
> Let $\delta_l$ denote the approximation error introduced by collapsing layer $l$.
> The propagated error satisfies
> $$
> \|\delta_{l+1}\| \le \|W_{l+1}\| \|\delta_l\|.
> $$
>
> Under $ℓ_2$ regularization, the operator norms $\|W_l\|$ remain controlled and empirically decrease with pruning, consistent with the observed reduction in loss landscape sharpness (Table 3).
> Thus, over $k$ collapsed layers,
> $$
> \|\delta_{\text{out}}\|\le
> \sum_{l=1}^k \left(\prod_{j>l}\|W_j\|\right)\mathcal{E}_l,
> $$
> which remains small since each $\mathcal{E}_l$ is exponentially suppressed. This explains why approximation errors do not accumulate catastrophically in deep Transformer models.
>
> ---
>
> We have tightened the wording to avoid any claim of exact algebraic collapse for GELU/SiLU. In the revised version, Footnote 3 now reads “For smooth rectifier variants such as GELU/SiLU, this absorption is approximate (they are smooth ReLU-like rectifiers).”

---

> > ### Author Response · Authors · 2026-01-02
> >
> > **[w2 - FLOPs reduction]**
> > We thank the reviewer for raising this important practical consideration. The observation regarding sparsity “fill-in” during sparse–sparse matrix multiplication is correct in general. However, this concern does not apply to the computational setting considered in our work, and we clarify the distinction below.
> >
> > ---
> >
> > Our claim that a linearized layer can be “absorbed by the following layer” is an **algebraic statement** about function composition:
> > $$
> > W_{l+1}(W_l x) = (W_{l+1} W_l)x.
> > $$
> >
> > Importantly, NEPENTHE does **not** rely on performing runtime multiplication of two unstructured sparse matrices. Once a layer reaches zero entropy and is deemed linearizable, it is **removed entirely from the computational graph**, and the model is recompiled with reduced depth. This is a **layer removal operation**, not a sparse–sparse fusion executed at inference time.
> >
> > We do not claim that the product $W_{l+1}W_l$ preserves the sparsity pattern of the original matrices. Indeed, fill-in is expected in general. However, our method does not require sparsity preservation in the merged operator.
> >
> > The primary source of computational savings in our work is **depth reduction**, i.e., shortening the critical path of forward propagation, rather than exploiting sparse matrix kernels. This is explicitly motivated in Sec. 1 and reflected in our experimental evaluation.
> >
> > The reductions reported in Table 10 are based on **actual measured inference time, memory usage, and energy consumption** on an NVIDIA A4500 GPU, not on idealized sparse FLOP counts. These measurements reflect the compiled model after layer removal.
> >
> > On modern GPUs, unstructured sparsity alone often yields limited speedups due to kernel launch overheads and memory access patterns. In contrast, reducing the number of layers directly reduces:
> > - kernel invocations,
> > - synchronization points,
> > - activation storage and memory traffic.
> >
> > Therefore, even if the merged linear operator were dense, removing an entire layer leads to tangible performance gains, as empirically demonstrated.
> >
> > We agree that *sparse–sparse matrix fusion is non-trivial* and can be counterproductive if sparsity preservation is required. However, this is orthogonal to our contribution. Our claim is that unstructured pruning can **induce structural simplifications**, enabling whole-layer removal with minimal accuracy loss. We do not rely on, nor claim, efficient sparse fusion of consecutive layers.
> >
> > **[w3 - Unstructured pruning induces collapse]**
> > NEPENTHE is also an unstructured pruning method. Like IMP, it prunes individual weights and does not enforce any structured pattern such as channels, blocks, or heads. The key difference between NEPENTHE and IMP  is how the pruning budget is allocated across layers.
> > As NPENTHE removes layers in our experiments, it appears that an entropy-guided form of unstructured pruning can induce layer collapse.
> >
> > **[w4 - Comparison with shallow initialized model]**
> > Thank you for your suggestion. We have added this baseline for ResNet-18 on CIFAR-10 in Table 5. We constructed a depth-reduced model by removing, from the start, the same three layers that NEPENTHE removes. After being trained by the same policy, this shallow model reaches 91.57 test top 1 accuracy. While NEPENTHE setting reaches 92.55. This comparison supports the necessity of the NEPENTHE process.

---

### Review · Reviewer_y9Ge · 2026-01-06

**Summary Of Contributions:**

This paper proposes a pruning technique, called _NEPENTHE_, that can be applied to neural networks with "rectifier" activation functions. This pruning technique is based on the "entropy" of each neuron, which measures the heterogeneity of its activations across the dataset. In short, if, for a given neuron, the same part of the activation function is used across the entire dataset, then its entropy would be close to zero, and the considered neuron can be pruned. This method allows the user to remove entire layers, if the entropy of its neurons is low.

**Audience:**

Yes

**Audience Explanation:**

Proposing an entropy-based pruning method that can be used to prune _entire layers_ is very appealing and is of interest to the ML community.

**Claims And Evidence:**

No

**Claims Explanation:**

The idea of removing neurons based on the heterogeneity of their activations is interesting and worths studying. However, several imprecisions should be clarified.

## Which activation functions are supported?
The paper is about "rectifier-activated networks". The term "rectifier" is used extensively to refer to a specific class of activation functions, which is not defined in the paper. It is crucial that the authors define the set of activation functions they are interested in. Otherwise, the range of application of the paper is unclear.

## Sensitivity of the computation of the entropy of neurons to the activation function
Let us consider the definition of the entropy of a neuron (Eqn. (2), Sec. 3.1). This definition suffers from a high sensitivity to the activation function. For instance, the entropy of a neuron can change dramatically if we replace the $\mathrm{ReLU}$ activation function by $\mathrm{ReLU} + \epsilon$ or $\mathrm{ReLU} - \epsilon$, with $\epsilon > 0$ arbitrarily small.

Additionally, the entropy changes if we add an offset to the activation function, but this operation does not change the expressivity of the neural network.

These two (related) facts tend to show that the definition of the entropy of the neuron is missing some key theoretical aspects. Nevertheless, the basic idea is good.

## Arbitrary definitions in the pruning strategy
**Entropy of a layer.** Above all, the entropy of a layer is defined as the average of the entropies of its neurons. It is very unclear that it is more interesting to choose this quantity instead of the "true" entropy of the layer, based on the _dependent_ scheme of activations (of all the neurons of the layer). Taking the average assumes independence of the activations, which is not the case in general. For instance, let us consider a layer with 100 neurons and a dataset of size 100. We assume that each neuron fires "positive" activations on 99 data points, and a "negative" activation on 1 data point. The "prunability" of the layer would not be the same if there is 1 data point for which all the activations are negative, and if each data $x_i$ point leads to a negative activation for exactly one neuron $n_i$. But the proposed entropy of a layer would be the same in both situations.

**Pruning irrelevance.** The definition of $\mathcal{I}\_l$ is:
$$\frac{1}{N\_l} \sum\_{i=1}^{N\_l} \hat{\mathcal{H}}\_{l,i} \cdot \frac{|w\_{l,i}|}{\|\mathbf{w}\_l\|\_0} .$$
Why does is involve $|w_{l,i}|$ and not $w_{l,i}^2$? Same with the entropy, etc. The proposed heuristic does not explain that.

**Inter-layer pruning relevance.** In the definition of $\mathcal{R}_l$ (Eqn. 14), there is a discontinuity when $\mathcal{I}_l \rightarrow 0$: as $\mathcal{I}_l \rightarrow 0$, $\mathcal{R}_l \rightarrow \infty$, but if $\mathcal{I}_l = 0$, then $\mathcal{R}_l = 0$. Moreover, $\mathcal{R}_l$ is said to belong to $[1, \infty)$, which is then false (since it can be $0$).

**Pruning parameter budget.**
Eqn. (15) indicates how to compute $\|\mathbf{w}_l\|_0^{\mathrm{pruned}}$. It uses a softmax function of $(\mathcal{R}_l)_l$. Why using a softmax and not a simple ratio, for instance:
$$\frac{\mathcal{R}_l}{\sum_j \mathcal{R}_j} \quad \text{instead of} \quad \frac{\exp(\mathcal{R}_l)}{\sum_j \exp(\mathcal{R}_j)}?$$This choice is strange, because, in deep learning, the softmax is used when we want to build a vector of probabilities with an input vector whose coordinates are in $\mathbb{R}$. But, in this case, the $\mathcal{R}_l$ are guaranteed to belong to $[1, \infty)$, so there is no need to take the exponential. Additionally, both of these ratios cannot be equal to 0, which means that, by themselves, the layers should always be pruned, at least a bit.

**Requested Changes:**

1. Clarify the set of activation functions that can be used in the proposed setup.
2. Propose a new definition of he entropy of a neuron or explain why the proposed definition is relevant (provided that a small change of the activation function can lead to a large change in the behavior of the algo).
3. Add a theoretical/heuristic justification for the definitions of: pruning irrelevance, inter-layer pruning relevance and pruning budget.

---

### Decision · Action_Editor_8ev1 · 2026-01-30

**Recommendation:** Accept with minor revision

**Additional Comments:**

This paper proposed a method to linearize and subsequently remove rectifier-activated layers in neural networks through entropy-guided unstructured pruning. It provided both theoretical analyses and empirical validation. The method is evaluated on popular CNN, Vision Transformer and NLP models, demonstrates performance improvements over existing techniques.

Overall, the paper is reasonably well-written. The idea of linking unstructured pruning with layer removal is interesting and reasonably novel. The paper also provides theoretical framework to support the proposed method. The empirical results further demonstrate the effectiveness.

The reviewers raised several concerns, primarily on the rigor and clarity of theoretical analysis. Some approximations and assumptions are not clearly explained, such as the GeLU and SiLU activation functions. In addition, the scope of certain claims and definitions is insufficiently specified, such as the definition of the neural entropy and average entropy. During rebuttal, the authors provide additional clarification and addressed some of these concerns. (Note that reviewer y9Ge submitted the review after the rebuttal period; After consulting with AE, the authors were informed that a response was not required for that review).

In summary, this is a reasonably solid paper, albeit with some reservations. Two reviewers recommend acceptance based on its novelty and relevance, while two lean rejection due to lack clarity in parts of the theoretical analysis. Given that the proposed technique is interesting and novel, demonstrates empirical performance improvements, and has the potential to inspire further research—particularly on using unstructured pruning to induce structure pruning effects, I am inclined towards acceptance.

However, several revisions are required.

1). Clarification of rectifier functions: the analysis and approximations for GeLU and SiLU presented during the rebuttal should be moved into the main text or appendix, rather than just mentioned in footnote on page4.

2). Clearer specification of the limitations of the theoretical framework, including the definition of neural entropy and average layer entropy.

3). The clarification of FLOP reduction in rebuttal should be included in the main paper or an appendix.

**Audience:**

Yes

**Audience Explanation:**

Yes. The paper proposes a technique to linearize and subsequently remove layers via entropy-guided unstructured pruning. The approach is interesting and is likely to attract researchers working on model compression and acceleration.

**Claims And Evidence:**

Yes

**Claims Explanation:**

The claims are reasonably well supported.